# Monitoring shipping emissions in the German Bight using MAX-DOAS measurements

**André Seyler[1], Folkard Wittrock[1], Lisa Kattner[1,2], Barbara Mathieu-Üffing[1,2], Enno Peters[1], Andreas Richter[1], Stefan Schmolke[2], and John P. Burrows[1]**

[1]Institute of Environmental Physics, University of Bremen, Germany
[2]Federal Maritime and Hydrographic Agency (BSH), Hamburg, Germany

*Correspondence to:* André Seyler (aseyler@iup.physik.uni-bremen.de)

**Abstract.** A three-year time series of ground-based MAX-DOAS measurements of $NO_2$ and $SO_2$ on the island Neuwerk has been analyzed for contributions from shipping emissions. The island is located in the German Bight, close to the main shipping lane (in a distance of 6–7 km) into the river Elbe towards the harbor of Hamburg. Measurements of individual ship plumes as well as of background pollution are possible from this location. A simple approach using the column amounts of the oxygen molecule dimer or collision complex, $O_4$, for the determination of the horizontal light path length has been applied to retrieve path-averaged volume mixing ratios. An excellent agreement between mixing ratios determined from $NO_2$ retrievals in the UV and visible parts of the spectrum has been found, showing the validity of the approach. Obtained mixing ratios of $NO_2$ and $SO_2$ are compared to co-located in-situ measurements showing good correlation on average but also a systematic underestimation by the MAX-DOAS $O_4$-scaling approach. Comparing data before and after the introduction of stricter fuel sulfur content limits (from 1 % to 0.1 %) on 1 January 2015 in the North Sea emission control area (ECA), a significant reduction in $SO_2$ levels has been observed. For situations with wind from the open North Sea, where ships are the only local source of air pollution, the average mixing ratio of $SO_2$ decreased by a factor of eight, while for $NO_2$ in the whole time series from 2013 till 2016 no significant change in emissions has been observed. More than 2000 individual ship emission plumes have been identified in the data and analyzed for the emission ratio of $SO_2$ to $NO_2$, yielding an average ratio of 0.3 for the years 2013/2014, decreasing significantly presumably due to lower fuel sulfur content in 2015/2016. By sorting measurements according to the prevailing wind direction and selecting two angular reference sectors representative for wind from open North Sea and coast excluding data with mixed air mass origin, relative contributions of ships and land-based sources to air pollution levels in the German Bight have been estimated to be around 40 % : 60 % for $NO_2$ as well as $SO_2$ in 2013/2014, dropping to 14 % : 86 % for $SO_2$ in 2015/2016.

# 1 Introduction

## 1.1 Shipping – a fast growing sector

Shipping has always been an important mode of transportation throughout the course of history. In contrast to the past, nowadays ships are almost exclusively carrying freight with the exception of a small number of cruise ships and ferries. Globalization of markets has lead to an enormous increase in world trade and shipping traffic in the last decades, with growth rates being typically about twice that of the world gross domestic product (GDP) (Bollmann et al., 2010).

Shipping is generally the most energy efficient transportation mode, having the lowest greenhouse gas emissions per tonne per kilometer (3–60 gCO$_2$/t/km), followed by rail (10–120 gCO$_2$/t/km), road (80–180 gCO$_2$/t/km) and air transport (435–1800 gCO$_2$/t/km), which is by far the least efficient (Bollmann et al., 2010; IEA/OECD, 2009). At the same time, with a volume of 9.84 billion tons in 2014 it accounts for four fifths of the worldwide total merchandise trade volume (UNCTAD, 2015), as compared to for example the total air cargo transport volume of 51.3 million tons

in 2014 (International Air Transport Association (IATA), 2015). As a result, shipping accounts for a significant part of the emissions from the transportation sector (Eyring et al., 2005b).

Despite growth rates now being lower compared to those prior to the 2008 economic crisis, seaborne trade is growing faster than the rest of the transportation sector, with an annual growth rate of 3–4 % in the years 2010 to 2014, compared to 2.0–2.6 % for the global merchandise volume (UNCTAD, 2014, 2015). The number of ships larger than 100 gross tonnage increased from around 31 000 in 1950 over 52 000 in 1970 to 89 000 in 2001 (Eyring et al., 2005a) and is estimated to increase to about 150 000 in 2050 (Eyring et al., 2005b). At the same time, total fuel consumption and emissions increased as well (Corbett and Koehler, 2003; Eyring et al., 2005a, b, 2010b). Eyring et al. (2005b) predicted that future development of shipping emissions will depend more on the usage of new technologies and imposed regulations than on the economic growth rates.

## 1.2 Ship emission chemistry

The most important pollutants emitted by ships are carbon dioxide ($CO_2$), carbon monoxide (CO), nitrogen oxides ($NO_x$ = NO + $NO_2$), sulfur dioxide ($SO_2$), black carbon (BC), volatile organic compounds (VOC) and particulate matter (PM) (Eyring et al., 2010a). This study focuses on $NO_2$ and $SO_2$, because both are emitted in considerable amounts and both absorb light in the uv-visible spectral range and therefore can readily be measured by Differential Optical Absorption Spectroscopy (DOAS), which is explained in Sect. 3.1. In 2001, shipping emissions accounted for 15 % of all anthropogenic $NO_x$ and provided 8 % of all anthropogenic $SO_2$ emissions (Eyring et al., 2010a).

$NO_x$ is predominantly formed thermally from atmospheric molecular nitrogen (N2) and oxygen ($O_2$) during high temperature combustion processes in ship engines in an endothermic chain reaction called the Zeldovich mechanism. The emitted $NO_x$ comprises mainly NO, with less than 25 % of $NO_x$ being emitted as $NO_2$ (Alföldy et al., 2013). Zhang et al. (2016) measured emission factors for gaseous and particulate pollutants on-board three Chinese vessels and found that more than 80 % of the $NO_x$ was emitted as NO and that emission factors were significantly different during different operation modes.

In the ambient atmosphere, NO is rapidly converted to $NO_2$ by reaction with ozone ($O_3$) leading to a life time of only a few minutes. During daytime $NO_2$ is photolyzed by UV radiation ($\lambda < 420$ nm) releasing NO and ground state oxygen radicals ($O(^3P)$). In a three-body-collision reaction involving $N_2$ or $O_2$ the oxygen radical reacts with an oxygen molecule to reform ozone (Singh, 1987). When daylight is available, these reactions form a "null-cycle" and transformation between NO and $NO_2$ is very fast, leading to a dynamic equilibrium. This is also known as the Leighton pho-

tostationary state. Owing to the lack of photolysis, NO reacts rapidly with $O_3$ to form $NO_2$ during the night. In addition, the nitrate radical (NO3) is formed by reaction of $NO_2$ with $O_3$. An equilibrium of $NO_2$ with NO3 forming $N_2O_5$, the acid anhydride of nitric acid $HNO_3$, results (Wayne, 2006; Seinfeld and Pandis, 2006).

During the day OH reacts with $NO_2$ in a three body reaction to form $HNO_3$. An important sink for $NO_2$ in the troposphere is wet deposition of the resulting $HNO_3$. The mean tropospheric lifetime of $NO_x$ varies between a few hours in summer and a few days in winter (Singh, 1987), depending on altitude. Inside ship plumes, Chen et al. (2005) found a substantially reduced lifetime of $NO_x$ of about 1.8 h compared to approximately 6.5 h in the background marine boundary layer (around noon). This is attributed to enhanced levels of OH radicals in the plume.

Unlike for $NO_x$, ship emissions of $SO_2$ are directly linked to the fuel sulfur content. Around 86 % of the fuel sulfur content is emitted as $SO_2$ (Balzani Lööv et al., 2014). Alföldy et al. (2013) found a linear relationship between $SO_2$ and sulfate particle emission and that only around 4.8 % of the total sulfur content is either directly emitted as or immediately transformed into particles after the emission. An important sink for $SO_2$ is wet deposition after oxidation by OH radicals to the extremely hygroscopic sulfur trioxide ($SO_3$) reacting rapidly with liquid water to form sulfuric acid ($H_2SO_4$) (Brasseur, 1999). Another important sink is dry deposition, leading to a lifetime of approximately one day in the boundary layer, which can be even shorter in the presence of clouds (Seinfeld and Pandis, 2006).

## 1.3 Influence on air quality and climate

Sulfate aerosols influence climate directly by scattering and absorption of solar radiation and indirectly by increasing cloud condensation, changing cloud reflectivity and lifetime (Lawrence and Crutzen, 1999; Lauer et al., 2007; Eyring et al., 2010b). In the presence of volatile organic compounds (VOC), nitrogen oxides are important precursors for the formation of tropospheric ozone and therefore photochemical smog. The release of both $NO_2$ and $SO_2$ leads to an increase in acidification of 3–10 % in coastal regions, contributing significantly to acid rain formation damaging eco-systems (Endresen et al., 2003; Jonson et al., 2000). The deposition of reactive nitrogen compounds causes eutrophication of ecosystems and decreases biodiversity (Galloway et al., 2003).

Around 70 % of shipping emissions occur within 400 km of land (Corbett et al., 1999), contributing substantially to air pollution in coastal areas (Eyring et al., 2010b). Ship emissions were found to provide a dominant source of air pollution in harbor cities (Eyring et al., 2010a). In addition to that, transport of tropospheric ozone and aerosol precursors over several hundreds of kilometers also affect air quality, human health and vegetation further inland, far away from their emission point (Corbett et al., 2007; Eyring et al., 2010a, b).

$NO_2$ and $SO_2$ can cause a variety of respiratory problems. Tropospheric ozone is harmful to animals and plants, causing various health problems. The EU legislation for O3 exposure to humans has set a target limit of $120 \, \mu g \, m^{-3}$ ($\sim 60 \, ppbv$) for an maximum daily 8 hour mean but allows exceedences on 25 days averaged over 3 years (EU, 2008, 2016). As mentioned above, both $NO_2$ and $SO_2$ play a role in the formation of particles. Fine particles are associated with various health impacts like lung cancer, heart attacks, asthma and allergies (Corbett et al., 2007; Pandya et al., 2002; WHO, 2006).

## 1.4 Attempts to decrease shipping emissions by stricter regulations

International ship traffic is subject to regulations of the International Maritime Organization (IMO). Shipping emissions are regulated by the International Convention for the Prevention of Pollution from Ships (MARPOL 73/78) Annex VI (DNV, 2008). This Annex was added in 1997 and entered into force in 2005. A revision with more stringent emission limits was adopted in 2008 and went into force 2010. With this, limits on sulfur content in heavy fuel oils globally are set and local Sulfur Emission Control Areas (SECA), later revised to general Emission Control Areas (ECA), along the North American coast and in the Baltic and North Sea (including the English Channel) are established with more stringent restrictions and controls. MARPOL introduced a global fuel sulfur limit of 4.5 %, which was reduced to 3.5 % in 2012 and will be further reduced in 2020 (or 2025 depending on a review in 2018) to 0.5 %. In the established ECAs, from 2010 on the limit was set to 1.5 % and was further reduced in 2010 to 1.0 %. Carrying out airborne in-situ measurements in several flight campaigns in the English Channel, North and Baltic Sea, Beecken et al. (2014) measured a 85 % compliance in 2011 and 2012 with the 1 % fuel sulfur limit. In the Gulf of Finland and Neva Bay area, Beecken et al. (2015) found a 90 % compliance in 2011 and 97 % compliance in 2012 with the 1 % fuel sulfur limit from ground-based, ship-based and helicopter-based in-situ measurements.

Recently, from 1 January 2015 on, the allowed fuel sulfur content in SECAs was further reduced to 0.1 %. Using in-situ measurements in Wedel at the bank of the river Elbe, a few kilometers downstream from Hamburg, Germany, Kattner et al. (2015) showed that in late 2014 more than 99 % of the measured ships complied with the 1 % sulfur limit and in early 2015 95.4 % of the measured ships complied with the new 0.1 % sulfur limit. By analyzing one and a half years of $SO_2$ measurements at the English Channel, Yang et al. (2016) found a three-fold reduction in $SO_2$ from 2014 to 2015. They estimated the lifetime of $SO_2$ in the marine boundary layer to be around half a day. Lack et al. (2011) measured a substantial drop of $SO_2$ emissions by 91 % when the investigated container ships entered the Californian ECA and switched from heavy fuel oil (HFO) with 3.15 % fuel sulfur content to marine gas oil (MGO) with 0.07 % fuel sulfur

content. These estimates were obtained performing airborne in-situ measurements.

MARPOL Annex VI also establishes limits dependent on engine power for the emission of $NO_x$ from engines built after 2000 (Tier I), 2011 (Tier II) and 2016 (Tier III), but due to the slow penetration to the full shipping fleet, the impact on $NO_x$ emissions is not yet clear. Since 2010, a $NO_x$ emission control area exists around the North American coast and in the Caribbean, while for North and Baltic Sea the establishment of such a NECA is planned and was recently agreed on, but the future enforcement date is still unclear. The European Union also established a sulfur content limit of 0.1 % for inland waterway vessels and ships at berth in Community ports, which is in force since 1 January 2010 (EU, 2005).

The impact of shipping emissions on the North Sea for different regulation scenarios was investigated in a model study by the Helmholtz-Zentrum Geesthacht (HZG) within the scope of the Clean North Sea Shipping project. For current emissions, a relative contribution of shipping emissions to air pollution in coastal regions of up to 25 % in summer and 15 % in winter for $NO_2$ and 30 % in summer and 12 % in winter for $SO_2$ was found (Aulinger et al., 2016). For the year 2030, the contribution of the continuously growing shipping sector to the $NO_2$ concentrations is predicted to decrease. The extent of reduction depends on the date on which the stricter Tier III regulations enter into force and on the fraction of the fleet complying to these regulations (i. e. the age of the fleet), with up to 80 % reduction if all ships comply (in the improbable case of a new ships only fleet). For $SO_2$, the established fuel sulfur content limit of 0.1 % (ECA) and 0.5 % (globally) will lead to significant reductions, a further decrease is expected if the fraction of LNG powered ships grows (Matthias et al., 2016).

## 1.5 DOAS measurements of shipping emissions – previous studies

Optical remote sensing using the Differential Optical Absorption Spectroscopy (DOAS) technique to measure shipping emissions has been conducted before. For example, Berg et al. (2012) performed airborne (from airplane and helicopter) DOAS measurements of $NO_2$ and $SO_2$ in ship plumes by measuring sea scattered light. Masieri et al. (2009) and Premuda et al. (2011) measured flow rate emissions (mass per second) of $NO_2$ and $SO_2$ for single ships with ground based MAX-DOAS measurements across the Giudecca Channel in the Venice lagoon. McLaren et al. (2012) measured nocturnal $NO_2$ to $SO_2$ ratios in ship plumes in the Strait of Georgia with the active long path DOAS technique. Balzani Lööv et al. (2014) tested and compared optical remote sensing methods (DOAS, LIDAR, UV camera) and in-situ (sniffer) methods for the measurement of shipping emissions in the framework of the SIRENAS-R campaign in the harbour of Rotterdam in 2009. Prata (2014) showed that a UV ($SO_2$) imaging camera can be used to measure $SO_2$ in

ship plumes at the Kongsfjord at Ny Ålesund, Svalbard and the harbor of Rotterdam.

The global pathways of the ships can be seen in long time averaged $NO_2$ measurements from various satellite instruments: from GOME over the Indian Ocean (Beirle et al., 2004), from SCIAMACHY on board ENVISAT over the Indian Ocean and the Red Sea (Richter et al., 2004), in even more detail with a lot more visible ship tracks from GOME-2 on board MetOp-A (Richter et al., 2011). The higher resolution of OMI yielded ship tracks in the Baltic Sea (Ialongo et al., 2014) and in all European seas (Vinken et al., 2014).

## 1.6 The MeSMarT project

The current study is part of the project MeSMarT (Measurements of Shipping emissions in the Marine Troposphere), which is a cooperation between the University of Bremen (Institute of Environmental Physics, IUP) and the Federal Maritime and Hydrographic Agency (Bundesamt für Seeschifffahrt und Hydrographie, BSH), supported by the Helmholtz Zentrum Geesthacht (HZG). It aims to monitor background concentration as well as elevated signals of gases and particles related to ship emissions with various methods to cover a wide range of relevant pollutants and their spatial and seasonal distribution to estimate the influence of ship emissions on the chemistry of the atmospheric boundary layer (for further information visit: http://www.mesmart.de/).

## 1.7 Aims of this study

The objectives of this study are to assess whether measurements of individual ship plumes are feasible with a ground-based MAX-DOAS instrument, to compare MAX-DOAS with co-located in-situ measurements, to estimate the contribution of ships and land-based sources to air pollution in a North Sea coastal region, to survey the effect of fuel sulfur content regulations on $SO_2$ concentrations in the marine boundary layer and to analyze the $SO_2$ to $NO_2$ ratio in plumes to gain information about plume chemistry and the sulfur content in shipping fuels.

In the following, first the measurement site is described, followed by a presentation of the wind statistics and data availability. After this, the Differential Optical Absorption Spectroscopy (DOAS), the MAX-DOAS instrumentation and measurement geometry as well as the DOAS data analysis approach used are briefly described. In the next section, selected results from this study are presented: the measured differential slant column densities (DSCD), the retrieved path-averaged volume mixing ratios, the comparison to in-situ measurements, the diurnal and weekly variability, the contribution estimates for ships as well as land-based pollution sources and the analysis of $SO_2$ to $NO_2$ ratios in ship plumes. Finally, a summary is given and conclusions are drawn.

## 2 Measurement site

The measurements presented within this study were taken on Neuwerk, a small island in the North Sea (German Bight) with the size of about $3 \, km^2$ and 33 inhabitants. It is located in the Wadden Sea northwest of Cuxhaven at the mouth of the river Elbe, roughly 8–9km off the Coast, as can be seen from the map in Fig. 1 A).

The North Sea has one of the highest ship densities in the world (Matthias et al., 2016). The majority of ships that arrive in the port of Hamburg sail through the German Bight and the river Elbe and therefore pass Neuwerk. Hamburg is among the largest ports worldwide, together with Rotterdam and Antwerp one of the three largest ports in Europe, having a 4–5 % increase in container volume in the last years (UNCTAD, 2014, 2015). Hamburg also experiences a large increase in the number of cruise ships, having 176 ship calls in 2014 compared to 25 in the year 2005 (Statistische Ämter des Bundes und der Länder (Statistikamt Nord), 2015).

Neuwerk is relatively close to the main shipping lane from the North Sea into the river Elbe. On this highly frequented waterway, nearly all ships to and from the port of Hamburg and the Kiel canal (connection to the Baltic Sea) pass the island at a distance of 6–7 km, as shown in Fig. 1 B). Still close, but further away to the west are the shipping lanes to the Weser river to the ports of Bremerhaven and Bremen and to Wilhelmshaven (JadeWeserPort).

Neuwerk is surrounded by the Hamburg Wadden Sea National Park and there are no significant sources of air pollution on the island itself, making it a very suitable station for measurements of shipping emissions.

The ship emission measurements presented in this study were carried out with a MAX-DOAS instrument (see Sect. 3.2) which measures in multiple azimuthal viewing directions, as shown in Fig. 1 B), pointing directly towards the shipping lane while the different viewing azimuth angles cover a large part of the region.

Several measurement devices, including the two-channel MAX-DOAS instrument (for UV and visible spectral range), an Airpointer in-situ measurement device (measuring $CO_2$, $NO_x$, $SO_2$ and $O_3$), a high volume filter sampler and passive samplers as well as a weather station and an AIS (Automatic Identification System) signal receiver, are positioned on the main platform of a radar tower at a height of about 30m (see Fig. 2).

Additional wind data is available from measurements by the Hamburg Port Authority (HPA) on Neuwerk and the neighboring island Scharhörn. The seasonal distribution of wind directions on Neuwerk is shown in Fig. 3.

In spring and summer, on a high percentage of days the wind blows from the open North Sea, where shipping emissions are the only significant source of local air pollution. Consequently, the site provides an optimal opportunity for measurements of ship emission plumes. In winter, southerly directions prevail, bringing potentially polluted air masses

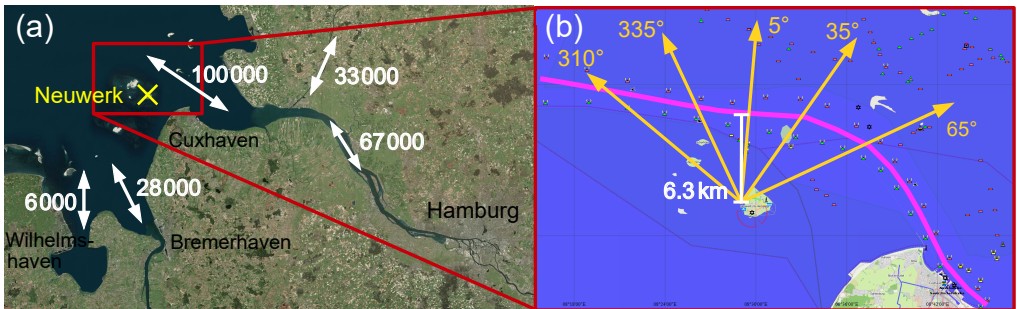

**Figure 1. (a)** Location of the measurement site Neuwerk in the German Bight, close to the mouth of the river Elbe. Number of ship movements (data from 2011/2012) is given by the white numbers. Data source: German Federal Waterways and Shipping Administration (WSV, 2013, 2014) Map source: http://www.bing.com/maps/ (01.04.2014)
**(b)** Azimuthal viewing directions of the MAX-DOAS instrument towards the main shipping lane (highlighted by the magenta line), passing the island in the north in a distance of 6–7 km. Map source: http://www.freie-tonne.de (16.07.2013)

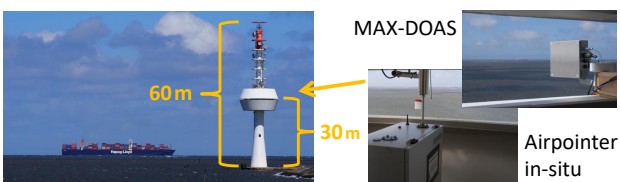

**Figure 2.** Radar tower Neuwerk with MAX-DOAS and in-situ measurement device

from the land and blowing the ship emission plumes away from the measurement site. In addition, as the MAX-DOAS technique requires daylight and because of the short days and the low sun resulting in less UV light reaching the surface, measurements are in general sparse in winter months, especially for $SO_2$, which has its strong absorption features in the UVB. This effect can be seen in winter gaps in Fig. 4, which presents the data availability for more than two years of measurements on Neuwerk.

## 3 Measurement techniques, instruments and data analysis

### 3.1 Differential Optical Absorption Spectroscopy (DOAS)

The principle of optical absorption spectroscopy is the attenuation of light intensity while passing through an absorbing medium, described by the well-known Lambert-Beer-law (also known as Beer-Lamber-Bouguer law). For the general case of electromagnetic radiation passing through an anisotropic medium having a number density $n$ and a temperature and pressure dependent absorption cross section $\sigma$ of an absorbing species along the light path $s$, the measured intensity at wavelength $\lambda$ is given by

$$I(s,\lambda) = I_0(\lambda) \cdot \exp\left\{ -\int_0^s n(s') \cdot \sigma\left(\lambda, T(s'), p(s')\right) \cdot \mathrm{d}s' \right\}$$

where the intensity of radiation entering the medium is $I_0$. For measurements in the atmosphere, this simple model has to be extended by considering multiple trace gases having different absorption cross sections and light scattering on air molecules (Rayleigh scattering), aerosol particles or water droplets (Mie scattering) as well as inelastic scattering by air and trace gas molecules (Raman scattering). The latter is responsible for the Ring effect (Grainger and Ring, 1962), another important extinction process, which can be described by a pseudo cross-section.

The key and original idea of the Differential Optical Absorption Spectroscopy (DOAS) is to separate the optical depth and the absorption cross-sections $\sigma_i(\lambda)$ into a slowly varying function $\sigma_{i,0}(\lambda)$ accounting for elastic scattering and broadband absorption structures and described by a low-order polynomial and a rapidly varying part $\sigma_i'(\lambda)$, the *differential* cross-section, considering the narrow-band absorption structures (Platt and Perner, 1980; Platt and Stutz, 2008). The absorption cross-sections are measured in the laboratory. Neglecting the temperature and pressure dependence of the absorption cross section, polynomial and differential cross sections are fitted to the measured optical thickness $\ln\left(\frac{I}{I_0}\right)$ in the linearized so-called DOAS equation:

$$\ln\left(\frac{I(\lambda)}{I_0(\lambda)}\right) = -\sum_{i=1}^{N} \mathrm{SCD}_i \cdot \sigma_i'(\lambda) - \sum_p c_p \cdot \lambda^p + \mathrm{Residual}(\lambda)$$

The retrieved quantities are the coefficients of the polynomial $c_p$ and the slant column density of the trace gas which is the integrated number density along the light path: $\mathrm{SCD}_i = \int n_i(s)\mathrm{d}s.$

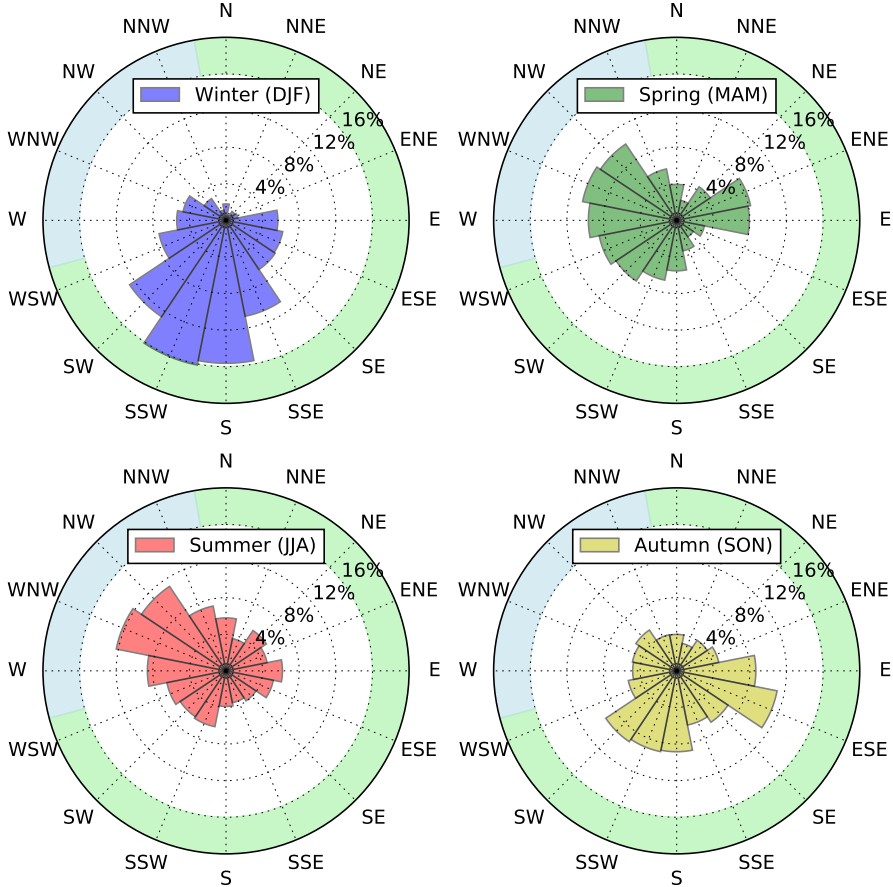

**Figure 3.** Seasonal wind direction distribution for Neuwerk (Data from 4 July 2013 to 27 June 2016). The colored sectors show directions with wind from the coast (green) and from the open North Sea (blue).

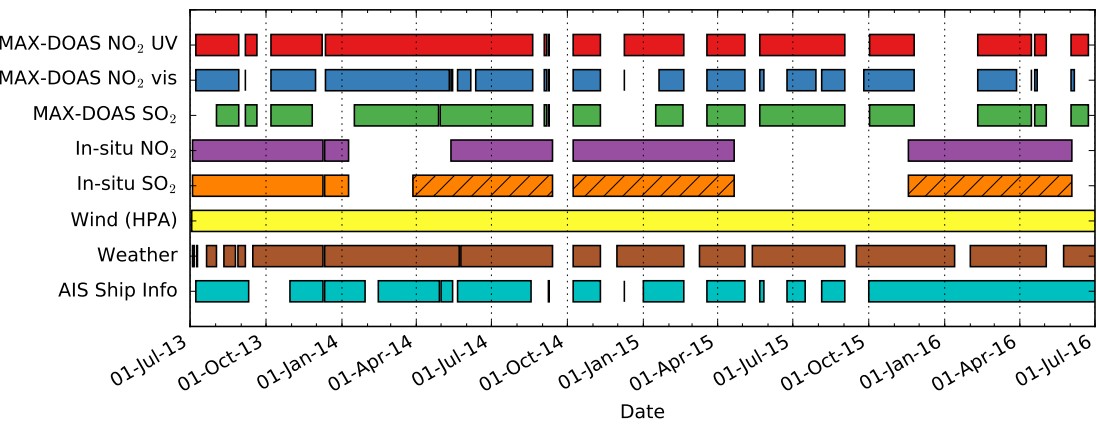

**Figure 4.** Data availability in the analyzed measurement period between July 2013 and July 2016. From March 2014 on (hatched), there were instrumental problems with the in-situ $SO_2$ instrument resulting in a strong oscillation of $\pm 0.5\,\mathrm{ppb}$ superimposing the data. However, this data can still be used for the comparison of long-term averages.

## 3.2 MAX-DOAS instrument and viewing geometry

The Multi-AXis DOAS (MAX-DOAS) technique (Hönninger et al., 2004; Wittrock et al., 2004) is a passive remote

sensing method measuring scattered sunlight. The MAX-DOAS instrument used in this study, comprises of a telescope mounted on a pan-tilt head, an optical fiber bundle, two spectrometers for UV and visible spectral range respectively, equipped with two CCD (charge coupled device) 2D array detectors operated by a computer. The telescope which is attached to the outer sheathing of the circular platform of the Neuwerk radar tower is used to collect the light from a specific viewing direction and to focus the light onto the entrance of the optical fiber. The combination of converging lens and light fiber leads to an field-of-view of approximately 1°. The pan-tilt head allows the instrument to point in different azimuth angles (panning) as well as different elevation angles (tilting). Dark measurements, which are needed for the determination of the CCD's dark signal are undertaken on a daily basis. Also on a daily basis line lamp measurements are taken using an internally mounted HgCd lamp for the wavelength calibration of the spectra and the determination of the slit function of the instrument. The spectral resolution, represented by the FWHM of the slit function of the instrument, is about 0.4 nm for the UV and 0.7 nm for the visible channel.

The Y-shaped optical light fiber cable is a bundle of $2 \times 38$ cylindrical, thin and flexible quartz fibers, guiding the light from the telescope to the two temperature-stabilized spectrometers with attached CCD detectors inside the weatherproof platform building. Each single fiber has a diameter of 150 μm and is 20 m long.

The UV and visible instrument consist of identical Andor Shamrock SR-303i imaging spectrographs, a grating spectrometer in "Czerny-Turner" design with a focal length of 303 mm. The gratings in use are different, the UV instrument is equipped with a 1200 grooves/mm, 300 nm blaze angle grating and the visible instrument with a 600 grooves/mm, 500 nm blaze angle grating. The UV instrument covers the wavelength range 304.6–371.7 nm, the visible spectrometer covers 398.8–536.7 nm. For the UV, a Princeton NTE/CCD 1340/400-EMB detector with a resolution of $1340 \times 400$ pixels and a pixel size of $20 \times 20$ microns, cooled to -35 °C, is used. For the visible spectral range, an Andor iDus DV420-BU back-illuminated CCD detector with a resolution of $1024 \times 255$ pixels and a pixel size of $26 \times 26$ microns, cooled as well to -35 °C, is used.

The measurement geometry for the ground-based MAX-DOAS measurements on Neuwerk is sketched in Fig. 5. To measure ship emissions, the telescope is pointed towards the horizon, collecting light that passed directly through the emitted ship plumes. A close-in-time zenith sky measurement is used as a reference so that the retrieved tropospheric differential slant column density (DSCD) is the difference of the slant column densities (SCD) along the two paths 1 and 2 in Fig. 5: $DSCD = SCD_1 - SCD_2 = SCD_{\text{off-axis}} - SCD_{\text{reference}}$. The stratospheric light path and trace gas absorption is approximately the same for both measurements and therefore cancels out which is important for $NO_2$ which

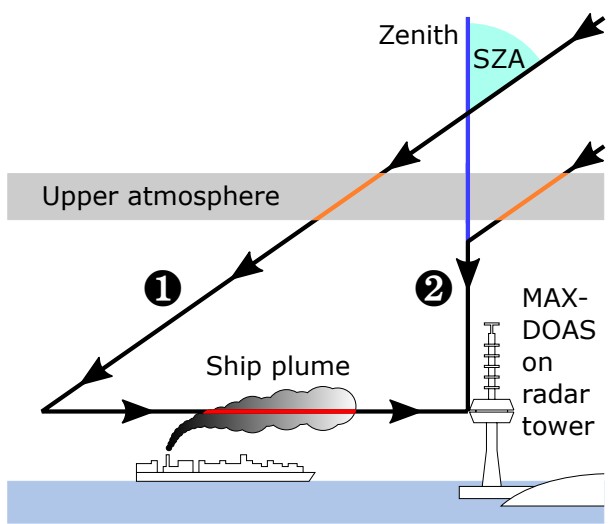

**Figure 5.** Measurement geometry for MAX-DOAS measurements on Neuwerk with schematic light paths for off-axis (1) and zenith sky reference measurements (2) for an exemplary solar zenith angle (SZA) of 55°

is also present in the stratosphere. This approach also minimizes possible instrumental artifacts.

The assumption that the vertical part of the light path cancels out when taking the difference between off-axis and zenith sky (reference) measurement off course is only valid if the $NO_2$ in the air above the instrument, which is of no interest to us here, is spatially homogeneously distributed. This is usually the case for stratospheric $NO_2$. If a spatially limited pollution plume from point sources like ships or power plants is blown above the radar tower and no plume is in the horizontal light path, the mentioned assumption is violated, leading to an underestimation of the derived DSCD. Also clouds or fog can make the interpretation of the measured DSCD more challenging due to multiple scattering.

### 3.3 DOAS data analysis and fit settings

The recorded spectra are spectrally calibrated using a daily acquired HgCd line lamp spectrum and the dark signal of the CCD detector is corrected using daily nighttime dark measurements. The logarithm of the ratio of measured off-axis (viewing towards the horizon) spectrum and reference (zenith sky) spectrum gives the optical thickness (also called optical depth). Multiple (differential) trace gas absorption cross sections obtained from laboratory measurements, as well as a low-order polynomial, are then fitted simultaneously to the optical depth. The retrieved fit parameters are the slant column densities of the various absorbers and the coefficients of the polynomial. The fits were performed with the software NLIN_D (Richter, 1997).

The settings and fitted absorbers vary according to the spectral range used. For the retrieval of $NO_2$ in the UV, a

fitting window of 338–370 nm was used and for $NO_2$ in the visible a fitting window of 425–497 nm, both adapted from experiences during the CINDI (Roscoe et al., 2010) and MAD-CAT (http://joseba.mpch-mainz.mpg.de/mad_cat.htm) inter-comparison campaigns. The oxygen-collision complex $O_2-O_2$, often denoted as $O_4$, is simultaneously retrieved from both $NO_2$ fits. The fit parameters for the DOAS fit of $NO_2$ and $SO_2$ are summarized in detail in Table 1.

For the retrieval of $SO_2$, several different fitting windows between 303 and 325 nm have been used in previous ground-based studies (Bobrowski and Platt, 2007; Lee et al., 2008; Galle et al., 2010; Irie et al., 2011; Wang et al., 2014a). This results from the need to find a compromise between the low light intensity caused by the strong ozone absorption around 300 nm on the one hand and the rapid decrease of the differential absorption of $SO_2$ at higher wavelengths on the other hand, limiting the choice of the fitting window. In this study, a fitting window of 307.5–317.5 nm was found as the optimal range for our instrument, which is similar to recommendations in Wang et al. (2014a). The fit parameters for the DOAS fit of $SO_2$ are summarized in detail in Table 2.

Only $SO_2$ measurements with a RMS lower than $2.5 \times 10^{-3}$ have been taken into account for the statistics, filtering out bad fits with ozone interferences in low light and bad weather conditions.

Under optimal conditions, the typical fit RMS is around $1 \times 10^{-4}$ for $NO_2$ in the visible, $2 \times 10^{-4}$ for $NO_2$ in the UV and $5 \times 10^{-4}$ for $SO_2$. By assuming that an optical density of twice the RMS can be detected (Peters, 2013), it is possible to estimate the detection limit of our instrument regarding the different trace gases. The differential absorption cross section of $NO_2$ is in the order of $1 \times 10^{-19}$ $cm^2$ $molec^{-1}$, for $SO_2$ in the order of $2 \times 10^{-19}$ $cm^2$ $molec^{-1}$. Combining this yields a $NO_2$ detection limit of around $1 \times 10^{15}$ molec $cm^{-2}$ corresponding to 0.05 pbb in the visible and $2 \times 10^{15}$ molec $cm^{-2}$ corresponding to 0.1 pbb in the UV. The $SO_2$ detection limit lies around $2.5 \times 10^{16}$ molec $cm^{-2}$ corresponding to 0.2 ppb. The typical absolute fit errors are $2$–$3 \times 10^{14}$ molec $cm^{-2}$ for $NO_2$ in the visible, $5$–$6 \times 10^{14}$ molec $cm^{-2}$ for $NO_2$ in the UV and $2 \times 10^{15}$ molec $cm^{-2}$ for $SO_2$, a factor of 5 to 10 smaller than the detection limit.

### 3.4 Retrieval of path-averaged near-surface VMRs from MAX-DOAS SCDs

To measure shipping emissions at our measurement site, our MAX-DOAS telescope is pointed towards the horizon, where the ships pass our site in a distance of 6–7 km. Since our instrument has a field-of-view of approximately $1°$, the lowest usable elevation angle avoiding looking onto the ground is $0.5°$, providing us with the highest sensitivity to near-surface pollutants. This is the elevation in which at our site usually the highest slant columns are measured. To convert a MAX-DOAS trace gas column which is the concentration of the absorber integrated along the effective light path into concen-

trations or volume mixing ratios, the length of this light path has to be known. This effective light path length depends on the atmospheric visibility, which is limited by scattering on air molecules as well as aerosols. As described in Section 3.2, trace gas absorptions in the higher atmosphere like stratospheric $NO_2$ nearly cancel out using a close-in-time zenith-sky reference spectrum. Following this, we can assume that the signal for our horizontal line-of-sight is dominated by the horizontal part of the light path after the last scattering event. As introduced by Sinreich et al. (2013), the length $L$ of this horizontal part of the light path can then be estimated using the slant column density of the $O_4$-molecule which has a well-known number density in the atmosphere:

$$L_{O_4} = \frac{SCD_{O_4,\text{horiz}} - SCD_{O_4,\text{zenith}}}{n_{O_4}} = \frac{DSCD_{O_4}}{n_{O_4}} \qquad (1)$$

The surface number density of $O_4$ is proportional to the square of the molecular oxygen concentration (Greenblatt et al., 1990; Wagner et al., 2004) and can be easily calculated from the temperature and pressure measured on the radar tower:

$$n_{O_4} = (n_{O_2})^2 = (0.20942 \cdot n_{\text{air}})^2 \qquad (2)$$

$$\text{with} \qquad n_{\text{air}} = \frac{N_{\text{air}}}{V_{\text{air}}} = \frac{p_{\text{air}} \cdot k_B}{T_{\text{air}}} = \frac{p_{\text{air}} \cdot N_A}{T_{\text{air}} \cdot R} \qquad (3)$$

with the Boltzmann constant $k_B$, Avogadro constant $N_A$ and universal gas constant $R$.

Knowing the path length, it is then possible to calculate the average number density of our trace gas $x$ along this horizontal path and the path-averaged volume mixing ratio:

$$n_x = \frac{SCD_{x,\text{horiz}} - SCD_{x,\text{zenith}}}{L_{O_4}} = \frac{DSCD_x}{L_{O_4}} \qquad (4)$$

$$\text{and with that:} \qquad VMR_x = \frac{n_x}{n_{\text{air}}} \qquad (5)$$

This $O_4$-scaling in principle takes into account the actual light path and its variation with aerosol loading and also needs no assumption on the typical mixing layer height, therefore overcoming the disadvantages of a simple geometric approximation.

However, when the atmospheric profile of the investigated trace gas $x$ has a shape that differs from that of the proxy $O_4$, systematic errors are introduced as has been shown by Sinreich et al. (2013) and Wang et al. (2014b) in extensive and comprehensive radiative transfer model (RTM) simulations. Pollutants like $NO_2$ and $SO_2$ have a profile shape very different from $O_4$. They are emitted close to the ground (e.g. from ships), have high concentrations in low altitude layers and tend to decrease very rapidly with height above the boundary layer. They are often approximated as box profiles, while the

**Table 1.** DOAS fit settings for the retrieval of $NO_2$ and $O_4$ in UV and visible spectral range

| Parameter | $NO_2$ (UV) | $NO_2$ (visible) |
|---|---|---|
| **Fitting window** | 338–370 nm | 425–497 nm |
| **Polynomial degree** | 4 | 3 |
| **Intensity offset** | Constant | Constant |
| **Zenith reference** | Coinciding zenith measurement[1] | Coinciding zenith measurement[1] |
| **SZA range** | Up to 85° SZA | Up to 85° SZA |
| **O3** | 223 K & 243 K (Serdyuchenko et al., 2014) | 223 K (Serdyuchenko et al., 2014) |
| **$NO_2$** | 298 K (Vandaele et al., 1996) | 298 K (Vandaele et al., 1996) |
| **$O_4$** | 293 K (Thalman and Volkamer, 2013) | 293 K (Thalman and Volkamer, 2013) |
| **H2O** | – | 293 K (Lampel et al., 2015) |
| **HCHO** | 297 K (Meller and Moortgat, 2000) | – |
| **Ring** | SCIATRAN (Rozanov et al., 2014) | SCIATRAN (Rozanov et al., 2014) |

[1] Interpolation in time between the zenith measurements directly before and after the off-axis scan.

**Table 2.** DOAS fit settings for the retrieval of $NO_2$ and $O_4$ in UV and visible spectral range

| Parameter | $SO_2$ (UV) |
|---|---|
| **Fitting window** | 307.5–317.5 nm |
| **Polynomial degree** | 3 |
| **Intensity offset** | Constant & slope |
| **Zenith reference** | Coinciding zenith measurement[1] |
| **SZA range** | Up to 75° SZA |
| **O3** | 223 K & 243 K (Serdyuchenko et al., 2014) |
| **$NO_2$** | 298 K (Vandaele et al., 1996) |
| **$SO_2$** | 293 K (Bogumil et al., 2003) |
| **Ring** | SCIATRAN (Rozanov et al., 2014) |

[1] Interpolation in time between the zenith measurements directly before and after the off-axis scan.

$O_4$ concentration simply decreases exponentially with altitude. This difference in profile shapes violates the basic assumption that the $O_4$ DSCD is a good proxy for the light path through the $NO_2$ and $SO_2$ layers. The resulting near-surface volume mixing ratios will not be representative for the amount of trace gases directly at the surface, but for some kind of average over a certain height range in the boundary layer.

The studies like Sinreich et al. (2013) and Wang et al. (2014b) use correction factors from radiative transfer calculations to account for this. These correction factors depend on the amount of aerosols present in the atmosphere, often described by the aerosol optical density (AOD), the solar zenith angle (SZA) as well as the relative solar azimuth angle (RSAA), the height of the pollutant box profile and the extend and vertical position of the aerosol layer in relation to this box profile (Sinreich et al., 2013). The strong dependence of the correction factors on the height of the box profile for trace gas layer heights of less than 1 km makes it necessary for the application of the suggested parameterization method to have additional knowledge about the trace gas layer height, ideally from measurements (e.g. LIDAR) or otherwise from estimations. The use of this method for low boundary layer heights below 500 m without knowing the actual height is not recommended by the authors (Sinreich et al., 2013).

At our measurement site, no additional knowledge (measurements) about the height of the $NO_2$ and $SO_2$ layers is available and the trace gay layer heights are typically around 200–300 m. A comparison of the uncorrected MAX-DOAS VMRs retrieved with the upper equations to our simultaneous in-situ measurements (see Section 4.5) confirms the need for a correction factor but also shows that the scaling factor needed changes from day to day as well as during the course of the day. This indicates, that the $NO_2$ and $SO_2$ layer height is very variable, depending on wind speed, wind direction, atmospheric conditions and chemistry. The lack of comparability between both measurement techniques and geometries, which is further discussed in Section 4.5, prevents us from estimating diurnally varying correction factors from this.

The non-consideration of these scaling factors will lead to a systematic overestimation of the effective horizontal path length and therefore to a systematic underestimation of

MAX-DOAS VMRs, up to a factor of three (Sinreich et al., 2013; Wang et al., 2014b).

In summary, a detailed radiative transfer study for the determination of the right correction factors is out of scope of this study which focuses on the statistic evaluation of a three year dataset of shipping emission measurements in the German Bight. Therefore, when in the following MAX-DOAS VMRs are shown, it has to be kept in mind that these are uncorrected VMRs obtained by above formulas.

This approach has been applied successfully by Sinreich et al. (2013) and Wang et al. (2014b) for measurements in urban polluted air masses over Mexico City and the city of Hefei (China) using MAX-DOAS measurements in $1°$ and $3°$ (Sinreich et al., 2013) and only in $1°$ elevation (Wang et al., 2014b), respectively. Gomez et al. (2014) applied this approach to measurements on a high mountain site at the Izaña Atmospheric observatory on Tenerife (Canary Islands), Schreier et al. (2016) at Zugspitze (Germany) and Pico Espejo (Venezuela). Due to the low aerosol amounts in such heights the latter two studies applied the approach without using correction factors. The fact that our instrument is located on a radar tower in a height of about $30\,\mathrm{m}$ above totally flat surroundings (the German Wadden Sea) allows an unblocked view to the horizon in all feasible azimuthal viewing directions. This led to the idea of trying to apply this approach to our shipping emission measurements on Neuwerk.

Since the $O_4$-DSCD is retrieved simultaneously to $NO_2$ in both the UV and visible DOAS fit for $NO_2$, this approach can be applied to $NO_2$ retrieved in both fitting ranges. The approach can also be applied to $SO_2$, although the difference of light paths due to the different fitting windows in the UV for $O_4$ ($NO_2$) and $SO_2$ introduces an uncertainty which has to be accounted for. Wang et al. (2014b) derived an empirical formula from RTM calculations for a variety of aerosol scenarios to convert the path length at $310\,\mathrm{nm}$ from the path length at the $O_4$ absorption at 360nm:

$$L_{310} = 0.136 + 0.897 \times L_{360} - 0.023 \times L_{360}^2 \qquad (6)$$

where $L_{310}$ and $L_{360}$ are given in km. This formula was also applied to our measurements to correct the light path length for the $SO_2$ fitting window. Although this formula has been calculated for polluted sites, the authors state that the deviations for other sites with different conditions are expected to be small (Wang et al., 2014b).

Using equations 1 to 5, several problems can arise from the division by the differential slant column density of $O_4$. For example if the $O_4$ DSCD is negative, which can happen at low signal-to-noise-ratio DOAS fits (e. g. under bad weather conditions), the resulting path length will be negative. If at the same time the trace gas DSCD is positive, then the trace gas volume mixing ratio will be negative as well, a nonphysical result. However, even when there is no $NO_2$ or $SO_2$, there is still some noise and therefore the retrieved VMR are

not exactly zero, but scatter around zero, so slightly negative values have to be included when averaging over time to avoid creating a systematic bias. If, on the other hand, the $O_4$ DSCD is close to zero, the path length will be very small leading to extremely high (positive or negative) mixing ratios which are also unrealistic. To address both problems, measurements with negative or small retrieved horizontal path lengths are discarded. For the measurements on Neuwerk, with respect to the characteristics of the measurement site, a minimum path length of $5\,\mathrm{km}$ seems to be a reasonable limit. This value provides the best compromise between the number of rejected bad measurements and the total number of remaining measurements for $NO_2$ in UV and visible as well as for $SO_2$. For statistics on differential slant column densities on the other hand, no such filtering is applied since negative values are not unphysical in this case and just mean that there is more trace gas absorption in the reference measurement than in the off-axis measurement.

## 3.5   In-situ instrumentation

In addition to the MAX-DOAS instrument, also in-situ observations are taken, using the Airpointer, a commercially available system which combines four different instruments in a compact, air-conditioned housing. The manufacturer is recordum (Austria), distributed by MLU (http://mlu.eu/recordum-airpointer/). The Airpointer device measures carbon dioxide ($CO_2$), nitrogen oxides ($NO_x = NO + NO_2$), sulfur dioxide ($SO_2$) and ozone ($O_3$) using standard procedures. Table 3 shows more detailed information about the different included instruments, their measurement methods, precision, and time resolution.

In this study the in-situ 1-minute-means of all compounds were used. $NO_2$ itself is not directly measured but calculated internally by subtracting the measured NO from the measured $NO_x$ concentration.

## 4   Results

### 4.1   Measured slant column densities of $NO_2$ and $SO_2$

In this study, three years of continuous MAX-DOAS measurements on Neuwerk have been evaluated. Figure 6 shows for one example day in summer 2014 the measured differential slant column densities of $NO_2$ in UV and visible spectral range as well as of $SO_2$ for the $0.5°$ elevation angle (viewing to the horizon) and the $-25°$ azimuth angle (approximately NNW direction, see Fig. 1). Sharp peaks in the curves originate from ship emission plumes passing the line of sight of the instrument. On this day, elevated levels of $NO_2$ have been measured in the morning, corresponding to a polluted air mass coming from land, which appears as an enhanced, slowly varying $NO_2$ background signal below the peaks. The systematic difference between the $NO_2$ in the UV (red curve) and the $NO_2$ in the visible (blue curve) emerges from the

**Table 3.** Specifications of the Airpointer in-situ device: measured trace gases, corresponding measuring techniques, measuring ranges and detection limits [Source: recordum/MLU (manufacturer), http://mlu.eu/recordum-airpointer/]

| Trace gas | $CO_2$ | $O_3$ | $NO, NO_2$ | $SO_2$ |
|---|---|---|---|---|
| **Measuring technique** | Non-dispersive IR spectroscopy LI-COR LI820 | UV absorption (EN 14625) | NO Chemi-luminescence (EN 14211) | UV fluorescence (EN 14212) |
| **Detection limit** | 1 ppm | 0.5 ppb | 0.4 ppb | 0.25 ppb |
| **Measuring range** | up to 20 000 ppm | up to 200 ppm | up to 20 ppm | up to 10 ppm |
| **Time resolution** | 1 s | < 30 s | < 60 s | < 90 s |

longer light-path in the visible due to stronger Rayleigh scattering in the UV (wavelength dependence $\propto \lambda^{-4}$). This is further investigated in Sect. 4.3 below.

By comparing $SO_2$ (green curve) with $NO_2$ (red and blue curves) it can be seen that for many of the $NO_2$ peaks there is a corresponding and simultaneous $SO_2$ peak, but not for all of them. This indicates a varying sulfur content in the fuel of the measured ships. Fuel with higher sulfur content leads to higher $SO_2$ emissions (see also Sect. 1).

By comparing measurements in different azimuthal viewing directions, the movement direction of the ship (and its plume) can be easily distinguished. The zoom in on the right of Fig. 6 shows the visible NO2 measurements in different azimuth directions for one example peak from the time series shown on the left. The color-coded viewing directions (see also Fig. 1) are sketched schematically below. From the measurements it can be seen that the emitted plume was consecutively measured in all directions at different times. It was first measured in the easternmost viewing directions and at last in the westernmost direction, indicating that the ship and its plume moved from east to west.

For the identification of sources for air pollution on Neuwerk, the wind direction distribution for the differential slant column densities of $NO_2$ and $SO_2$ measured in 2013 and 2014 is plotted for four different elevation angles ($0.5°$, $2.5°$, $4.5°$ and $30.5°$) in Fig. 7. When the wind is coming from the open North Sea (blue shaded sector) the measured $NO_2$ and $SO_2$ DSCD are clearly lower than for other directions, for which the wind is coming from the coast (green and yellow shaded sectors) and blows land-based air pollution to the island. The wind direction dependence is more or less similar for both trace gases but with a higher fraction of ship related signals in the overall $SO_2$ columns. The values are especially high when the wind is coming from the cities of Cuxhaven (ESE direction) and Bremerhaven (SSE) for both $NO_2$ and $SO_2$.

Elevation angle sequences of slant columns (i.e. vertical scanning) contain information on the vertical distribution of trace gases. For lower elevation angles, the measured trace gas slant columns for tropospheric absorbers are usu-

ally higher because of the longer light path in the boundary layer.

As expected, higher elevations show on average lower DSCDs due to the shorter light path in the boundary layer. The highest $NO_2$ and $SO_2$ DSCD in the lowest elevation angle ($0.5°$, blue bars) in relation to DSCDs in higher elevations are measured especially for wind from all northern directions, in a sector ranging from WSW to ESE. These directions coincide with the course of the main shipping lane coming from the WSW direction (the English Channel, the Netherlands, East Frisian Islands), passing the island in the north and running close to the city of Cuxhaven (ESE direction) into the river Elbe. This indicates that the enhanced columns in the $0.5°$ elevation angle is pollution emitted from ships in a surface-near layer.

For southerly wind directions no major shipping lane is in the direct surrounding and land-based pollution sources dominate. The average DSCDs in $0.5°$ and $2.5°$ elevation are nearly the same for both $NO_2$ and $SO_2$ indicating that the pollution is located higher up in the troposphere.

## 4.2   Volume mixing ratios of $NO_2$ and $SO_2$

For the example day presented in Fig. 6 the path-averaged volume mixing ratios retrieved with the approach presented in Sect. 3.4 are shown in Fig. 8.

From the mathematics of the approach one would expect a good agreement between the $NO_2$ volume mixing ratios retrieved in UV and visible if $NO_2$ is well mixed in the boundary layer, since averaging constant values over different paths should give equal mean values. In the figure, in fact one can see a very good agreement between both $NO_2$ volume mixing ratios, in particular for situations characterized by background pollution.

Although the light path in the visible spectral range is clearly longer than in the UV, for all the peaks shown here the UV instrument measured a higher path-averaged VMR. The reason for that are spatial inhomogeneities along the line-of-sight.

If $NO_2$ is not distributed homogeneously along the light path, which is the case in the presence of individual ship exhaust plumes, one can expect different values for the means

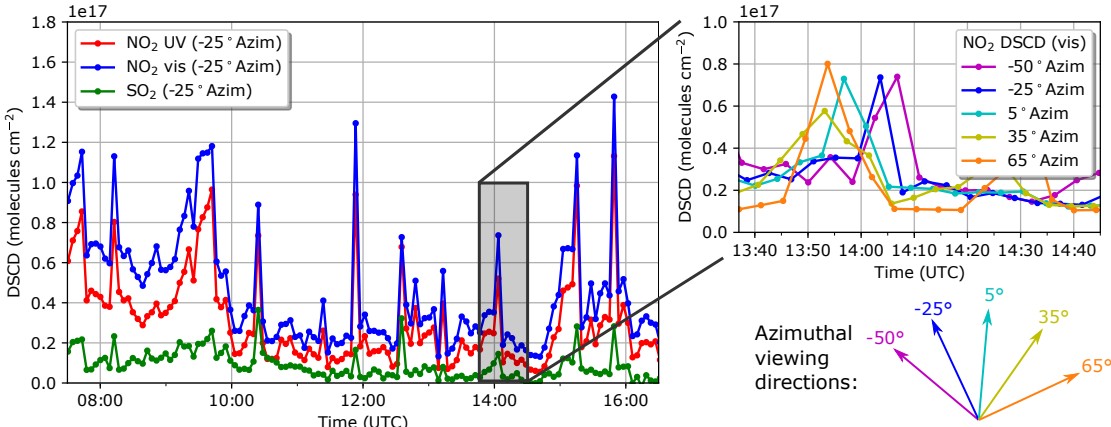

**Figure 6.** $NO_2$ (UV and visible) and $SO_2$ differential slant column densities measured in 0.5° elevation and the -25° viewing azimuth angle (approximately NNW direction) on Neuwerk on Wednesday, 23 July 2014. The excerpt on the right shows for one example peak the $NO_2$ (vis) measurements in different azimuth viewing directions.

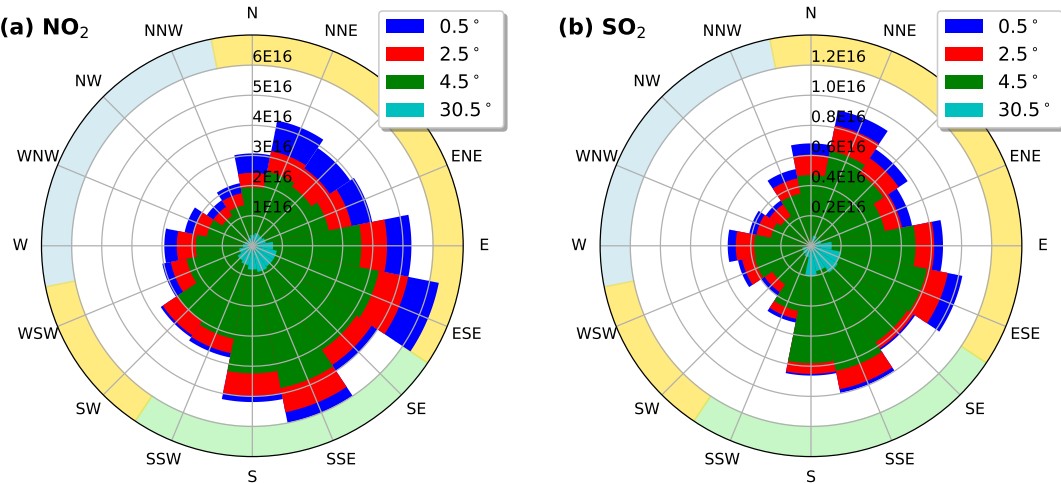

**Figure 7.** Overlayed wind roses for different elevation angles showing the wind direction distribution of the UV $NO_2$ **(a)** and $SO_2$ **(a)** differential slant column densities measured in the main viewing direction in 0.5°, 2.5°, 4.5° and 30.5° elevation in the years 2013 and 2014. The wind roses are plotted on top of each other, i. e. the highest values were measured in the lowest elevation angle (blue bars). The colored sectors show directions with wind from land (green), open North Sea (blue) and mixed origin (yellow).

over the two light paths as they probe different parts of the $NO_2$ field. Such differences can be identified in the figure by looking at the peaks.

The light path in the visible spectral range is longer than in the UV because of more intensive Rayleigh scattering in the UV. The difference between UV and visible peak values depends on the exact location of the plume within the light paths.

A short distance of the plume to the instrument and its complete coverage by the shorter UV path leads to higher values in the UV since the part of the light path probing the higher $NO_2$ values has a larger relative contribution to the signal than for the longer visible path.

If the plume is further away from the instrument and only in the visible path or close to the UV scattering point, one will retrieve a higher volume mixing ratio in the visible. This relationship contains information on the horizontal distribution of the absorber and will be further investigated in a second manuscript.

### 4.3   Statistical evaluation of UV and visible $NO_2$ data

To investigate quantitatively the relationship between the $NO_2$ slant column densities measured simultaneously in the UV and visible spectral range, all single pairs of DSCD mea-

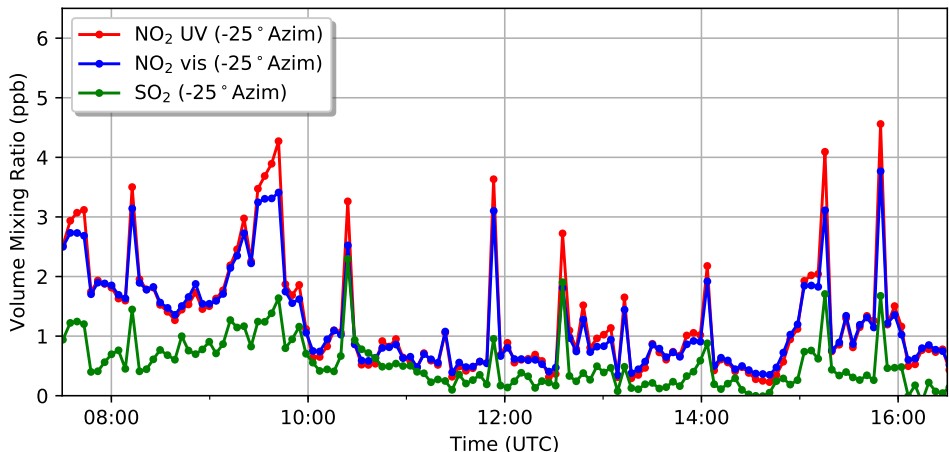

**Figure 8.** $NO_2$ (UV and visible) and $SO_2$ path-averaged volume mixing ratios measured in 0.5° elevation angle and -25° viewing azimuth angle (approximately NNW direction) on Neuwerk on Wednesday, 23 July 2014.

surements with an RMS better than $1 \times 10^{-3}$ are plotted into a scatter plot, shown in Panel (a) of Fig. 9.

As can be seen from the figure, $NO_2$ DSCDs in UV and visible are strongly positively correlated with a Pearson correlation coefficient of 0.983. Because of the difference in the horizontal light path lengths in both spectral regions (due to more intense Rayleigh scattering in the UV), the slope of the regression line is 1.30 corresponding to a 30 % longer light path in the visible. The intercept of the regression line is small. Panel (b) of Fig. 9 shows a histogram of the ratios between both slant column densities. The distribution peaks for ratios of 1.3, in good agreement with the retrieved slope from the scatter plot.

When converting the slant column densities to mixing ratios using the $O_4$-scaling, the dependence on light path should be removed and quantitative agreement is expected between the UV and visible VMRs. A scatter plot for the horizontal path averaged volume mixing ratios is shown in Panel (c) of Fig. 9. It is clearly visible that the points scatter symmetrically along the 1:1 identity line. Comparing this plot with the plot in Panel (a) shows that the difference in light path lengths is in fact corrected for by the $O_4$-scaling approach. The slope of the regression line is close to unity and the intercept is very small. The Pearson correlation coefficient has further increased to 0.984. The histogram (Panel d of Fig. 9) peaks at 1.0.

As discussed above, differences are still expected not only as a result of measurement uncertainties but also due to different averaging volumes in case of inhomogeneous $NO_2$ distributions (which is especially the case for ship plumes under certain wind directions). For the horizontal light path lengths, a mean value of 9.3 km with a standard deviation of 2.3 km was retrieved in the UV, and a mean value of 12.9 km with a standard deviation of 4.5 km was retrieved in the visi-

ble. On days with optimal measurement conditions (clear sky days), typical horizontal light paths are around 10 km in the UV and 15 km in the visible spectral range.

### 4.4 Allocation of ship emission peaks to ships using wind and AIS data

The detailed information on passing ships transmitted via the *Automatic Identification System* (AIS) and the acquired weather and wind data can be used to allocate the measured pollutant peaks to individual ships.

Measurements from Wednesday, 9 July 2014 are shown in Fig. 10. Panel (a) shows the MAX-DOAS differential slant column density of $NO_2$. Panel (b) includes various information about passing ships: The vertical bars indicate when a ship was in the line-of-sight of the MAX-DOAS instrument. Solid bars represent ships coming from the left and going to the right (from west to east, i. e. sailing into the river Elbe), dashed bars vice versa. The colors of the bars indicate the ship length, with small ships shown in blue and very large ships ($> 350$ m) in red. Panel (c) displays the wind speed and direction.

On this day, the wind was coming from northern directions, directly from the shipping lane, with moderate wind speeds of 10 to 35 km/h, resulting in low background pollution values ($1$–$2 \times 10^{16}$ molec/cm$^2$) as well as sharp and distinct ship emission peaks (up to $1.2 \times 10^{17}$ molec/cm$^2$) of $NO_2$. By comparing the ship emission peak positions to the vertical bars (representing times when ships crossed the MAX-DOAS line-of-sight) in the schematic representation below it can be seen that most of the peaks can be allocated to individual ships. In some cases, when two or more ships simultaneously cross the line-of-sight, the single contributions can not be separated. Large ships (orange and red bars) tend to exhaust more $NO_2$ while the contribution of small ships

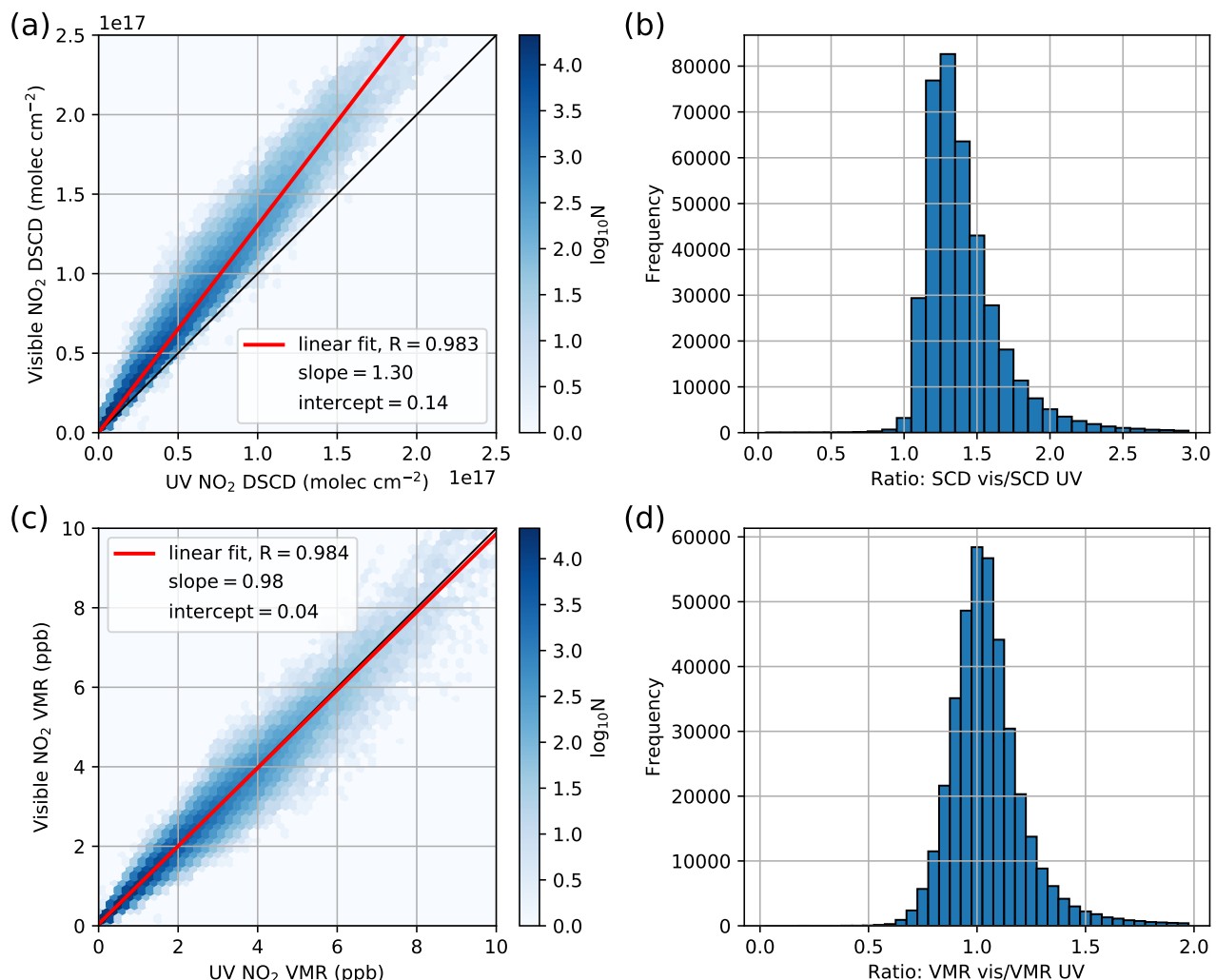

**Figure 9. (a)** Scatter plot: $NO_2$ slant column density retrieved in the visible vs. UV measured in all azimuth angles at $0.5°$ elevation for solar zenith angles smaller than $75°$. The parameters derived from the linear fit by orthogonal distance regression (Deming regression) are also shown. **(b)** Histogram of the ratio of the two $NO_2$ slant column densities (visible/UV). **(c)** As (a), but for volume mixing ratios. **(d)** Histogram of the ratio of the two $NO_2$ volume mixing ratios (visible/UV).

(length $< 30\,\mathrm{m}$) represented by the dark blue bars is usually not measurable.

### 4.5  Comparison of MAX-DOAS VMR to in-situ measurements

The fact that our measurement site is also equipped with an in-situ device (see Section 3.5 for a description), makes it possible to compare the MAX-DOAS VMRs of $NO_2$ and $SO_2$ to our simultaneous in-situ measurements. The differences of both measurement techniques need to be considered for such a comparison: The MAX-DOAS averages over a long horizontal light path, while the in-situ device measures at a single location inside the plume. Since ship plumes usually never cover the whole light path but rather a small

fraction of it, very high concentration peaks are usually underestimated in the MAX-DOAS VMR.

Figure 11 shows the horizontal path averaged $NO_2$ volume mixing ratio retrieved from the differential slant column densities shown in Fig. 10 as well as the in-situ $NO_2$ volume mixing ratio (Panel a) in combination with ship data (Panel b) and wind data (Panel c).

Ship emission peaks measured by the in-situ instrument are both higher and broader than the corresponding MAX-DOAS peaks, leading to a considerably larger integrated peak area, showing the systematic underestimation of the $NO_2$ concentrations inside ship plumes by the MAX-DOAS instrument due to the averaging along the horizontal light path.

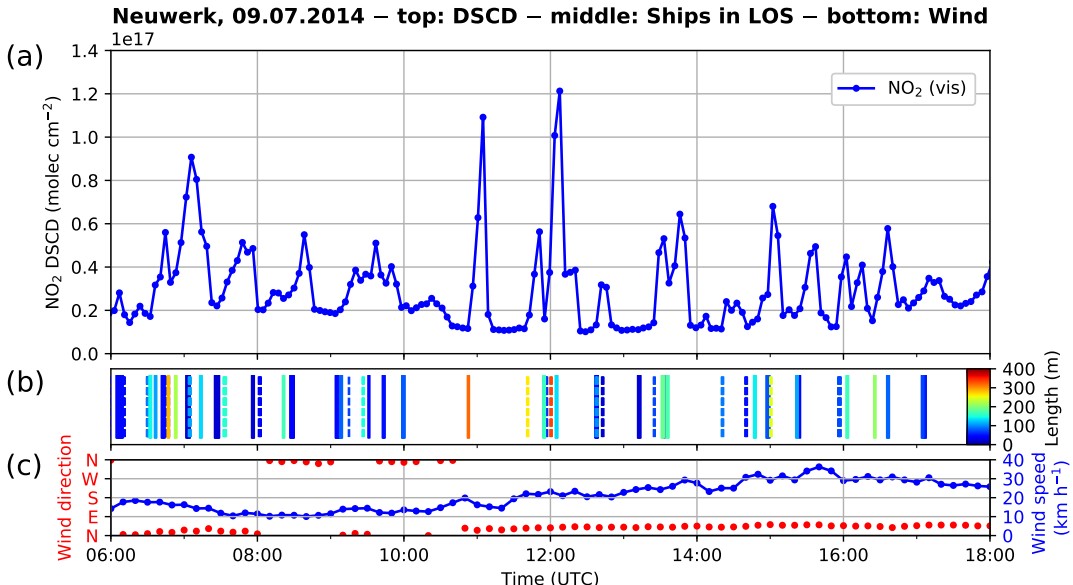

**Figure 10.** NO$_2$ differential slant column densities, AIS and wind data for Neuwerk on Wednesday, 9 July 2014.
**(a)** NO$_2$ DSCD in 0.5° elevation for the 35° azimuth viewing direction
**(b)** Vertical bars indicating that a ship is in the line-of-sight of the instrument, solid bars: ship moves from left to right (west to east), dashed vice versa, colors representing ship length
**(c)** Wind speed and direction measured on Scharhörn (HPA)

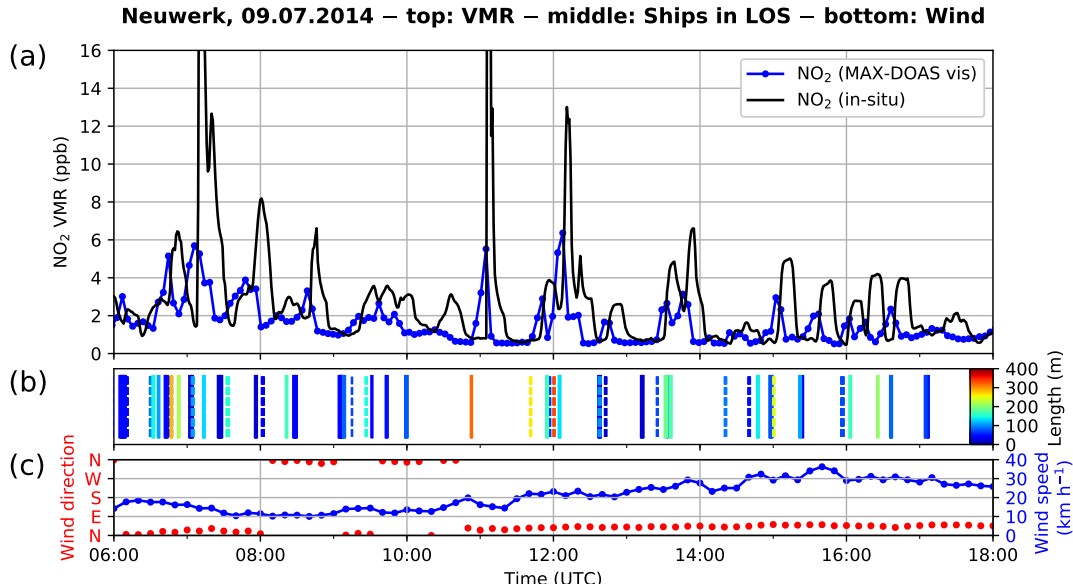

**Figure 11.** MAX-DOAS and in-situ NO$_2$ volume mixing ratio, AIS and wind data on Wednesday, 9 July 2014:
**(a)** MAX-DOAS (visible) and in-situ NO$_2$ VMR
**(b)** Vertical bars indicating that a ship is in the line-of-sight of the instrument, solid bars: ship moves from left to right (west to east), dashed vice versa, colors representing ship length
**(c)** Wind speed and direction measured on Scharhörn (HPA)

Normally, a time-shift between MAX-DOAS and in-situ peaks exists, which is due to the long distance of about 6–7 km to the shipping lane, that the plumes have to travel until they reach the radar tower. This time-shift depends on the wind velocity and gets smaller for higher wind speeds. In the figure, this dependency can be seen when comparing the magnitude of the time delay for measurements in the morning (low wind speeds) and evening (higher wind speeds) This travel time also explains the broader peaks in the in-situ measurements, since the emitted plume spreads and dilutes on its way to the radar tower.

However, if the pollution is horizontally well-mixed in the measured air mass, which is approximately the case for background pollution coming from the coast but not for ship plumes, MAX-DOAS and in-situ instrument should in principle measure the same values. However, as discussed in Section 3.4, correction factors need to be applied to the MAX-DOAS VMRs to account for the different profile shapes of $O_4$ and the investigated pollutants $NO_2$ and $SO_2$, but in our case cannot be determined because no measurements of the height of the $NO_2$ and $SO_2$ layer exist. The uncorrected VMRs shown here can be strongly underestimated (up to a factor of 3), because they have been calculated with an overestimated path length. This is the case for background pollution as well as shipping emission measurements.

Since the lack of comparability between both instruments for individual measurements, for a meaningful comparison and the computation of a correlation coefficient at this measurement site an averaging over longer time spans was applied to reduce the impact of the differences between both measurement methods. The fact that MAX-DOAS averages over a large horizontal distance should therefore cancel out on temporal average when comparing to in-situ measurements.

Figure 12 shows in Panel (a) three months of daily mean $NO_2$ VMRs from the in-situ and MAX-DOAS UV instrument in summer 2014 and in Panel (b) due to instrumental problems with the in-situ $SO_2$ device (see Fig. 4) six weeks of $SO_2$ daily mean VMRs from summer 2013. To have comparable conditions, for the in-situ instrument all measurements between the start of the MAX-DOAS measurements in the morning (with sunrise) and the end of measurements in the evening (with sunset) have been averaged. The shaded areas show the corresponding standard deviation and indicate the variability during the single days.

The long gap in the $SO_2$ time series was caused by a power outage.

It is clearly visible that the in-situ $NO_2$ VMRs are systematically higher than the uncorrected MAX-DOAS VMRs. The scaling factors which would be needed to bring both time series into agreement differ from day to day. A closer look into the individual days shows that these scaling factors also vary over the course of the day, even when wind direction and speed do not change. The scatter plot for this time-series of $NO_2$ measurements in Fig. 13 Panel (a) shows a good correlation between MAX-DOAS and in-situ daily means, but a slope strongly deviating from one and also some scatter.

The most important reason for the systematic differences is certainly the non-consideration of the correction factors arising from the different profile shapes of $O_4$ and $NO_2$, leading to a systematic underestimation of the VMRs from the MAX-DOAS instrument (see Section 3.4 for a more detailed discussion). But also "light dilution", i.e. light scattered into the line-of-sight between the instrument and the trace gas plume (Kern et al., 2010) might play a role reducing the measured off-axis SCDs .

For $SO_2$, the daily mean VMRs from MAX-DOAS and in-situ instrument in Fig. 12 Panel (b) show a much better agreement. The scatter plot in 13 Panel (b) confirms this with a slope much closer to unity, but more scatter around the fitted line.

The difference in scaling factors for $NO_2$ and $SO_2$ can be attributed to plume chemistry. During combustion, mainly nitric oxide (NO) is produced. This has to be converted to $NO_2$ (through reaction with tropospheric ozone) before it can be measured by the MAX-DOAS instrument. Since the MAX-DOAS instrument sees the ship plumes in an earlier state, the fraction of $NO_2$ should be lower than in the in-situ measurements, explaining at least a part of the difference.

Although MAX-DOAS and in-situ VMRs show systematic deviations in the absolute values, a very good agreement of the shape (the course) of the curves is found for $NO_2$ as well as $SO_2$. This illustrates that MAX-DOAS can determine day-to-day trends as in-situ measurements, even though no correction factors have been applied.

## 4.6 Diurnal and weekly variability of $NO_2$

Although our measurement station is located on a small island in the German Bight close to the mouths of the Elbe and Weser river, our measurements are strongly influenced by air pollution from traffic and industry on land, depending on the prevailing wind direction. As can be seen from Fig. 1 (a) and 3, wind coming from northeasterly, easterly, southerly and southwesterly directions will blow polluted air masses from the German North Sea Coast and hinterland to our site. In Figure 14 the average diurnal variation of the measured $NO_2$ volume mixing ratios is shown as hourly mean values. Solid curves show the respective curve for all measurements (with all wind directions), dashed lines show the subset of measurements with wind coming only from the open North Sea with no coastal background pollution. Looking at the diurnal variation in all measurements, the typical daily cycle for road-traffic-influenced air masses with enhanced values in the morning and in the late afternoon during rush hour can be seen. If we restrain the data to periods with wind from the open North Sea (dashed curves), this diurnal cycle vanishes and values are more or less constant over day and also considerably lower. This result is in accordance with the ex-

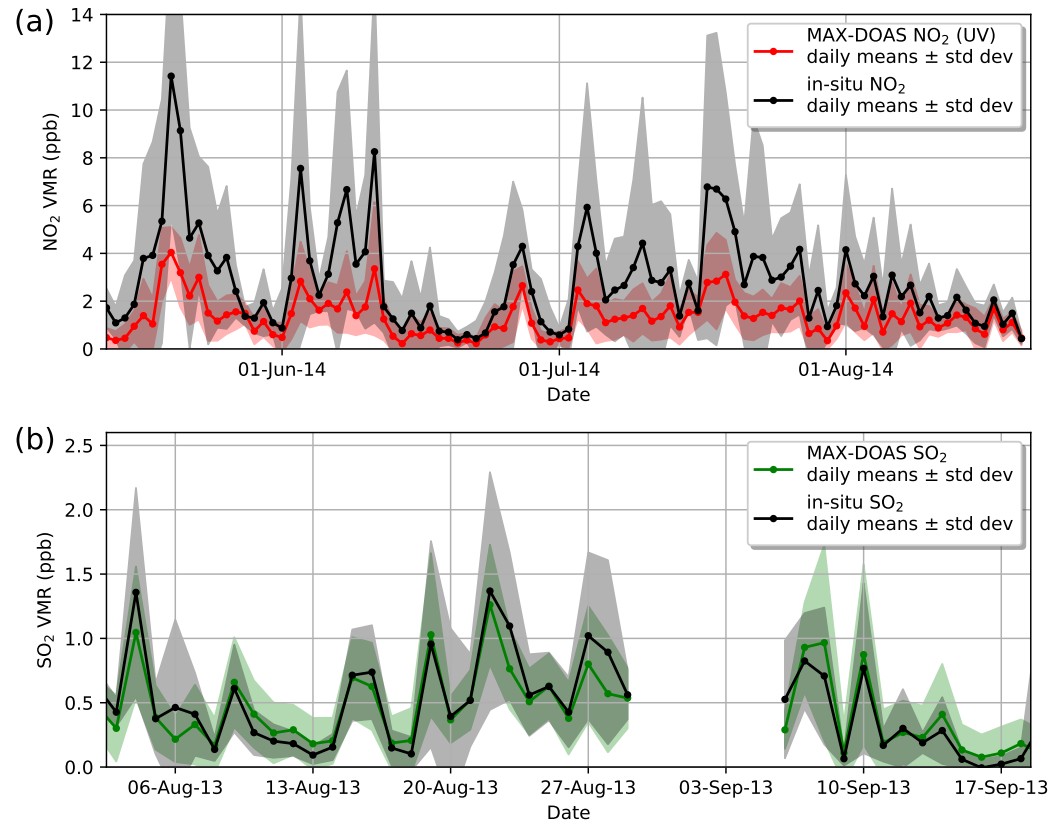

**Figure 12.** Comparison of MAX-DOAS (UV) and in-situ daily mean VMRs of $NO_2$ **(a)** during summer 2014 and $SO_2$ **(b)** during summer 2013. Shaded areas show the standard deviation for each daily mean value.

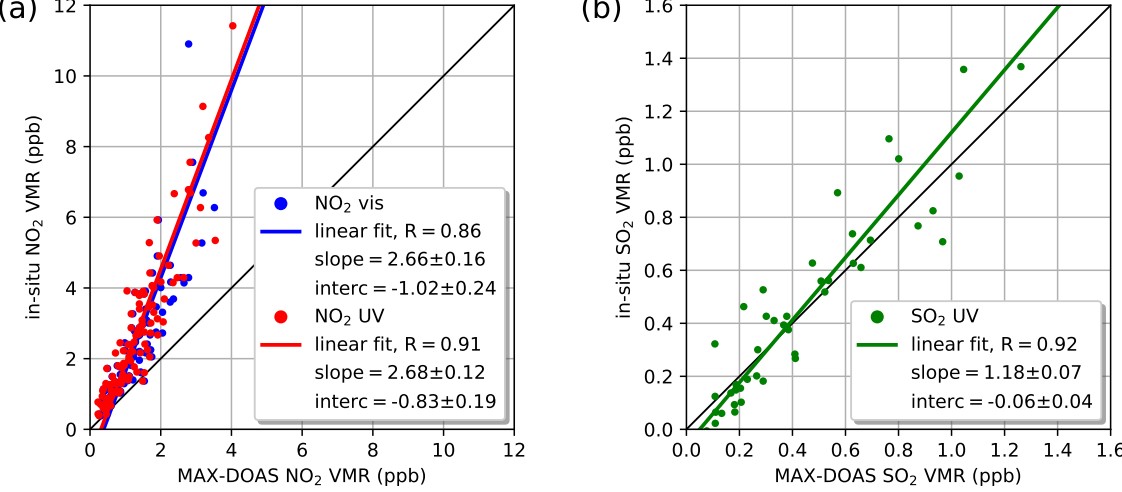

**Figure 13.** Scatter plot of **(a)** $NO_2$ VMR and **(b)** $SO_2$ VMR from MAX-DOAS vs. in-situ. For $NO_2$ daily means from summer 2014, for $SO_2$ daily means from summer 2013 are shown. For the MAX-DOAS instrument, to get a better statistic, all measurements in all azimuth viewing directions have been averaged. For the in-situ instrument, the mean of all measurements during the daily MAX-DOAS measurement periods (sunrise till sunset) has been taken. The linear fits were calculated with orthogonal distance regression (Deming regression), parameters are shown in the figures.

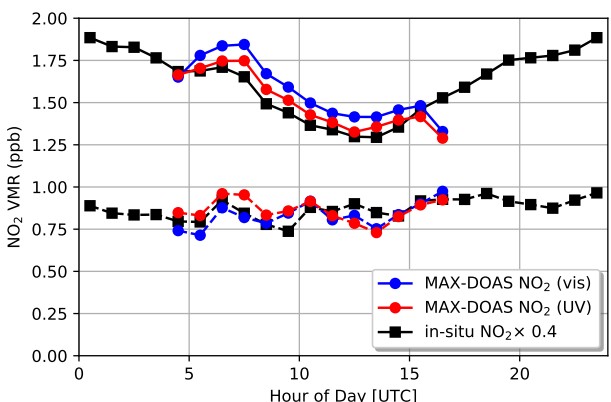

**Figure 14.** Average diurnal cycle of MAX-DOAS (UV and visible) and in-situ $NO_2$ volume mixing ratios for all measurements (solid lines) and for a subset of measurements with wind from the open North Sea (dashed lines). For a better visual comparability the in-situ values are scaled by a factor of 0.4.

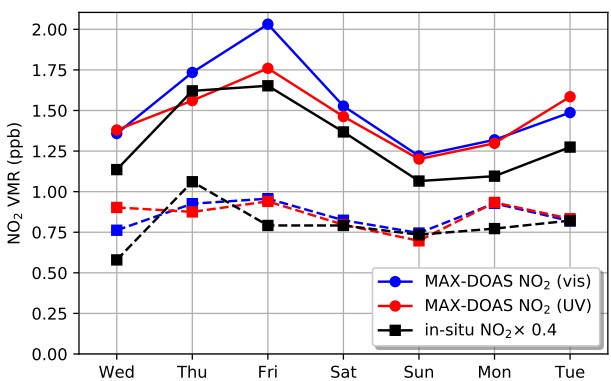

**Figure 15.** Average weekly cycle of MAX-DOAS (UV and visible) and in-situ $NO_2$ volume mixing ratios for all measurements (solid lines) and for a subset of measurements with wind from the open North Sea (dashed lines). For a better visual comparability the in-situ values are scaled by a factor of 0.4.

pectations that the amount of ship traffic should be almost independent from the time of day.

The mean $NO_2$ volume mixing ratios for each weekday shown in Fig. 15 illustrate again the influence of land-based road traffic. If we consider the whole time series (solid lines), lowest values are measured on Sundays, when road traffic is less intense. There is only little weekly cycle for air masses coming from the open North Sea (dashed lines). Measurements are more or less constant and again considerably lower. Such a weekly cycle for $NO_2$ in polluted regions has been observed and discussed several times before, for example in Beirle et al. (2003), Kaynak et al. (2009), Bell et al. (2009) and Ialongo et al. (2016).

It is also remarkable that except for a scaling factor of approximately 0.4, the shape of the diurnal and weekly cycle retrieved from MAX-DOAS and in-situ measurements agrees very well for both instruments.

## 4.7 Dependence of $NO_2$ and $SO_2$ pollution levels on wind direction

As already mentioned in Sect. 1, on the 1st of January 2015, the sulfur content of marine fuels allowed inside the North and Baltic Sea Emission Control Areas (ECA) has been substantially decreased from $1.0\%$ to $0.1\%$. Therefore, one would expect lower sulfur dioxide ($SO_2$) values in 2015 compared to the years before, especially when the wind is blowing from the open North Sea, where shipping emissions are the only source of $SO_2$. This expectation is confirmed by the measurements. In the data since 2015, no distinct ship emission peaks are visible anymore (for an example day see Section 4.9 below). For a more detailed analysis, mean values over the whole time series before and after 1 January 2015 have been investigated, separated according to the prevailing wind direction.

Two days of $SO_2$ measurements (20 and 30 October 2014) showing very high values over several hours have been excluded from the time-series. Comparisons with our simultaneous in-situ measurements and measurements from the German Umweltbundesamt at the coast of the North Sea in Westerland/Sylt and at the coast of the Baltic Sea on the island Zingst showing a similar behavior as well as HYSPLIT backward trajectories suggest that on both days $SO_2$ plumes of the Icelandic volcano Bárdarbunga have influenced the measurements in northern Germany.

Figure 16 shows the wind direction distribution of the mean $NO_2$ and $SO_2$ path averaged volume mixing ratios for all measurements before and after the change in fuel sulfur limit regulations.

For $SO_2$, a significant decrease is found, particularly for wind directions from West to North with wind from the open North Sea. For this sector, values in 2015 are close to zero. This shows that the new and more restrictive fuel sulfur content limits lead to a clear improvement in coastal air quality. For wind directions with mainly land-based sources, no or only a small decrease is observed.

The typical average $SO_2$ concentrations measured by the German Federal Environmental Agency (Umweltbundesamt, 2017) in 2016 for rural stations in Northern Germany are around $0.5$–$1\,\mu g\,m^{-3}$, corresponding to $0.2$–$0.4\,ppb$ (Conversion factor: $1\,ppb \,\widehat{=}\, 2.62\,\mu g\,m^{-3}$ for $SO_2$). Measurements in cities and especially close to industrial areas show higher values. Bremerhaven, which is the station closest to our instrument, has a mean concentration of $1.77\,\mu g\,m^{-3}$, corresponding to $0.67\,ppb$. The reported values for rural stations are in good agreement with our measurements of $0.3$–$0.4\,ppb$ for wind directions with mainly land-based pollution sources (green sector in Fig. 16 Panel b) since January 2015.

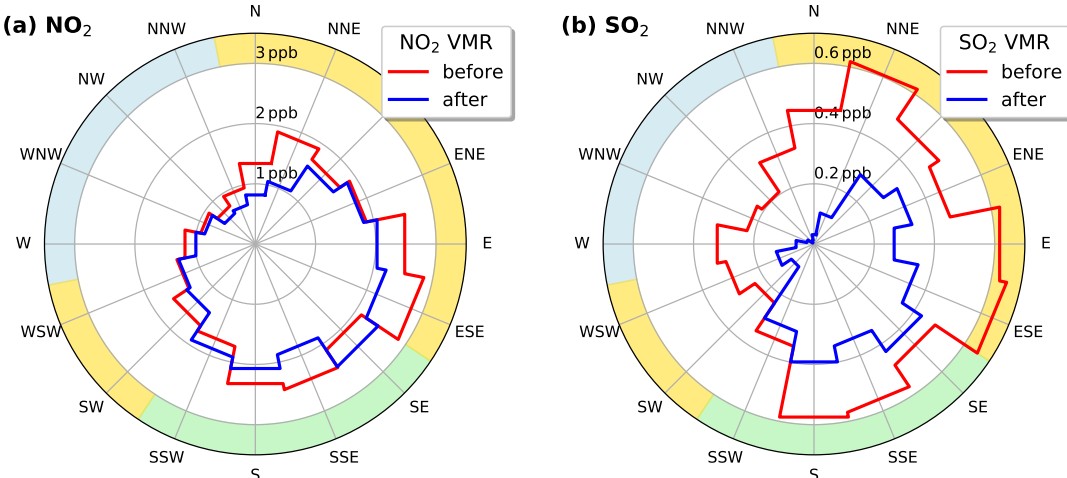

**Figure 16.** Wind direction distribution of the measured $NO_2$ **(a)** and $SO_2$ **(b)** volume mixing ratio in $0.5°$ elevation before and after the change in fuel sulfur limit regulations on 1 January 2015. The colored sectors show directions with wind mainly from land (green), open North Sea (blue) and mixed origin (yellow).

For $NO_2$ on the other hand, both the directional distribution and the absolute values are nearly identical for both time periods, implying no considerable changes in $NO_x$ emissions. This result meets the expectations, since no $NO_x$ emission limits have been set into force up to now for the North and Baltic Sea emission control area.

## 4.8 Contributions of ships vs. land-based pollution sources on coastal air quality on Neuwerk

The distribution of measured $NO_2$ and $SO_2$ volume mixing ratios depending on the wind direction shown in Fig. 16 can be used to estimate the contributions of ships and land-based sources to coastal air pollution levels. To trade ship emissions off against land-based emissions (e.g. industry, road transport), two representative sectors of wind directions have been chosen, both 90 degrees wide: A north-westerly sector ($258.75°$ to $348.75°$) with wind from the open North Sea and ships as the only local source of air pollution and a south-easterly sector ($123.75°$ to $213.75°$) with wind mainly coming from land and almost no ship traffic. Air masses brought by wind from the other directions, for example from the mouth of the river Elbe in the East of Neuwerk, can contain emissions from land-based pollution sources as well as ship emissions. These remaining directions will be called "mixed" in the following. It is now assumed, that trace gas concentrations measured during periods with wind from one of these sectors have their source in the according sector. For getting a good statistic, measurements in all azimuth angles have been included. Figure 17 shows the results in several pie charts.

For both $NO_2$ and $SO_2$, more than half (around 50–60 %) of all measurements have been taken while wind was coming from either the assigned sea or land sector. This implies that

not only a small sample, but the majority of measurements can be used for the estimation of source contributions, making the assumption of using these sectors as representative samples for ships and land-based source regions a reasonable approximation. There are differences in the time series of $NO_2$ and $SO_2$ coming from the fact that the $SO_2$ fit delivers realistic values only up to $75°$ solar zenith angle and the $NO_2$ was fitted until $85°$ SZA, leading to less measurements for $SO_2$ than for $NO_2$, especially pronounced in winter times. Despite this, the general distribution pattern of wind direction frequency for $NO_2$ and $SO_2$ is quite similar, with wind coming from the sea 32–42 % of the time and from the land sector 18–24 % of the time.

For $NO_2$ (upper row in Fig. 17), more than half of the total $NO_2$ measured on Neuwerk can be attributed to wind from either of both sectors, with 21 % coming from ships and 31 % coming from land.

If we consider only the two sectors, for which we can identify the primary sources and take theses as representative, we can say that 40 % of the $NO_2$ on Neuwerk is coming from shipping emissions, but with 60 %, the majority, is coming from land. One reason for that is that the island Neuwerk is relatively close to the coastline (around $10\,km$) and is obviously still impacted by polluted air masses from land, which has also been observed in the diurnal and weekly cycle analysis shown in Figures 14 and 15. This might also give us a hint that in coastal regions in Germany land-based sources like road traffic and industry are, despite the heavy ship traffic, the strongest source of air pollution and ship emissions come in second.

For $SO_2$ the whole time series of measurements from 2013 to 2016 was divided into two periods of nearly the same length: The first period is 2013 and 2014, which was before

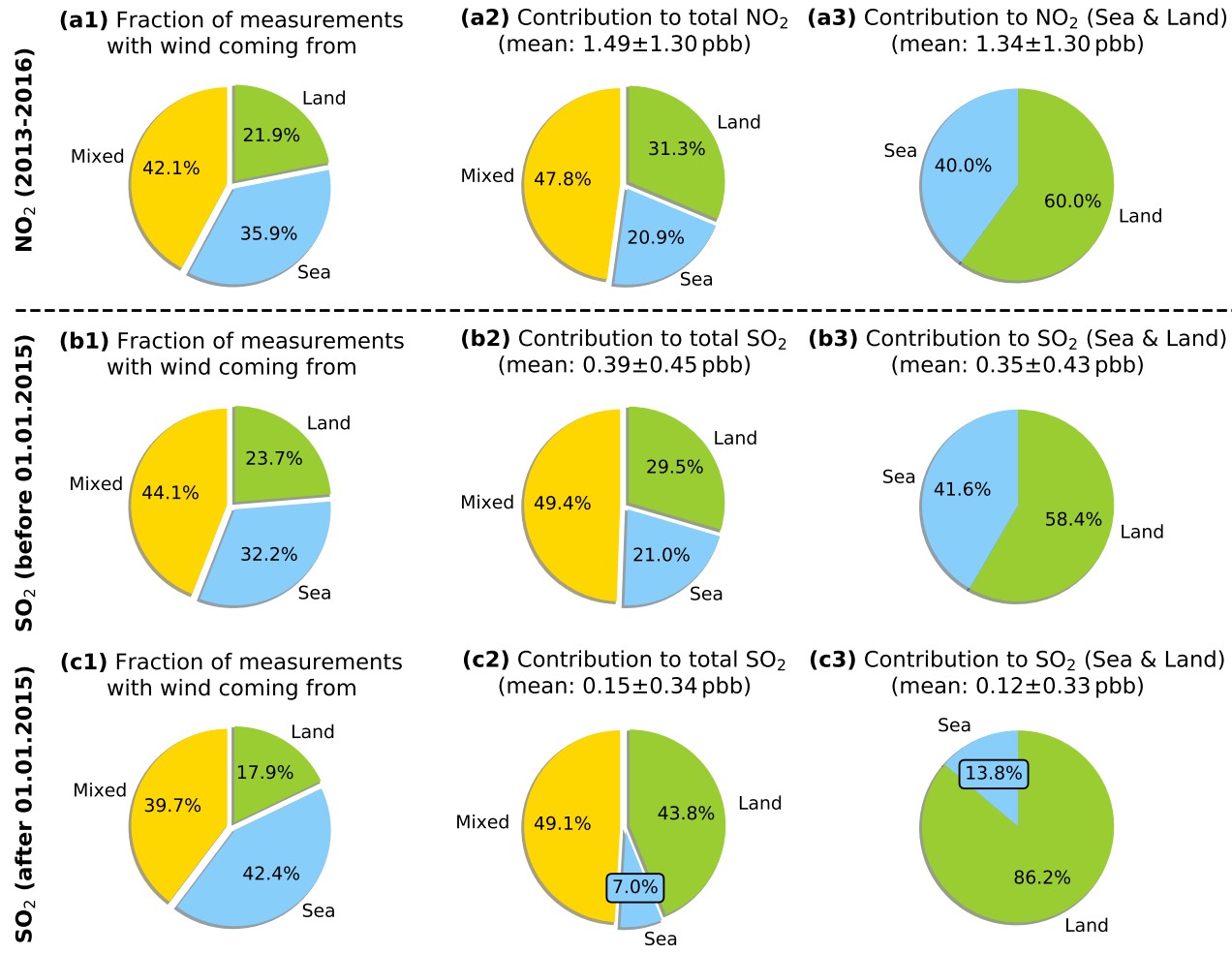

**Figure 17.** Contributions of ships and land-based pollution sources to measured $NO_2$ and $SO_2$ levels on Neuwerk:
**(a1), (b1) and (c1):** Percentage of measurements with wind coming mainly from land (green), only from sea (blue) and from directions with mixed contributions (yellow) for all $NO_2$ data **(a1)**, $SO_2$ data before (b1) and after the change in fuel sulfur content limits **(c1)**.
**(a2), (b2) and (c2):** Contributions to the integrated volume mixing ratios of $NO_2$ **(a2)** and $SO_2$ **(b2, c2)** from the source regions in percent.
**(a3), (b3) and (c3):** Contributions to the integrated volume mixing ratios when considering only the land and sea sector. It can clearly be seen that the lower fuel sulfur limit lead to a strong decrease in the $SO_2$ contribution from shipping since 2015.

the introduction of stricter sulfur limits for maritime fuels in the North Sea on 1 January 2015. The according statistics to this period are shown in the middle row in Fig. 17. The second time period, after the change in fuel sulfur limits, includes all measurements from 2015 and 2016, with the corresponding pie plots in the bottom row of Fig. 17.

Before the change, 32 % of the measurements were taken when the wind was coming from the sea sector and about 24 % when it was blowing from the dedicated land sector. After the change, the wind was coming a bit more often from sea (42 %) and less often from land (18 %), but in general the situation was quite similar.

The contributions of the three sectors (land, sea and mixed) to the total integrated $SO_2$ with 21 % coming from

ships, 30 % from land and 49 % from the mixed sector for the time before the change in sulfur limits are very similar to those of $NO_2$, too. After the change, the contribution from the sea sector shrinks significantly from 21 to 7 %, while the relative contribution from the land sector increased from 29 to 44 %, the contribution from the mixed sector staying the same as around 49 %. This increase for the land source sector is only a relative increase while the absolute contributions slightly decreased, as can be seen from Fig. 16. The relative contribution from the sea sector (shipping only source) decreased by a factor of 3 while the absolute contribution from this sector decreased by a factor of 8, even though the wind was coincidentally blowing more often from the open sea in this time period.

The overall mean $SO_2$ volume mixing ratio before 2015 is $0.39 \pm 0.45$ ppb (mean $\pm$ standard deviation). For 2015 and 2016, the total mean value declined by two-thirds to $0.15 \pm 0.34$ ppb (mean $\pm$ standard deviation).

These results show clearly that the stricter limitations on the fuel sulfur content are working and significantly improved air quality in the North Sea coastal regions with respect to $SO_2$. This is in good agreement with other studies such as Kattner et al. (2015), who found that around 95 % of the ships are sticking to the new limits. This implies that the cheaper high sulfur heavy oil fuel is no longer in use in the region of measurement.

If again the two selected sectors are considered as representative for both land and sea sources, the shares of the contributions from sea/land changed from 42 % : 58 % (which is very similar to those of $NO_2$) to 14 % : 86 %. This again shows that since 2015, the vast majority of $SO_2$ emissions can be attributed to land sources and ships play only a negligible role. Prior to 2015, shipping emissions have been a significant source for $SO_2$ in coastal regions.

One aspect which is neglected in the source allocation to wind sectors is that in situations with good visibility and low wind speeds even for wind coming from southern directions, the MAX-DOAS instrument can measure ship emissions peaks in the north of the island, but being typically very small. Compared to the often strongly enhanced background pollution in cases with southerly winds, the contribution from these peaks is negligible (around 1–3 %), but certainly leads to a small overestimation of land sources.

## 4.9 Determination of $SO_2$ to $NO_2$ ratios in ship plumes

A monitoring of emissions from single ships requires the analysis of individual plume peaks in the $NO_2$ and $SO_2$ data sets. It is difficult to derive the absolute amounts (e.g. in mass units) of the emitted gaseous pollutants by our MAX-DOAS remote sensing technique. The height and width of the measured peaks does not only depend on the amount of emitted pollutants), but also strongly on the geometry, while getting the highest values when measuring alongside the plumes, and much smaller values when the plume moves orthogonal to the line-of-sight of our instrument. In addition to that, also the time span between emission and measurement plays a role for the height of the $NO_2$ peaks because of NO to $NO_2$ titration.

To determine the mixing ratio inside the plumes, additional information on the length of the light path inside the plume would be needed, which cannot be retrieved from our measurements. This means that without further assumptions, we cannot determine emission factors for the emitted gases (e.g for emission inventories, which are used as input for model simulations).

Although emission factors cannot be measured by MAX-DOAS directly, the $NO_2$ and $SO_2$ signals yield the ratio of both. These ratios can then be compared to ratios of emission factors reported in other studies as well as measurements on other sites or with different instruments, bearing in mind possible deviations due to NO to $NO_2$ titration.

By comparing $SO_2$ to $NO_2$ ratios from different ships it is possible to roughly distinguish whether a ship is using fuel with high or low sulfur content (giving a high or low $SO_2$ to $NO_2$ ratio). Beecken and Mellqvist from Chalmers University (Sweden) use this relationship for airborne DOAS measurements of ship exhaust plumes on an operational basis in the CompMon project (Compliance monitoring pilot for MARPOL Annex VI) (Van Roy, 2016). Following the ships and measuring across the stack gas plume they can discriminate between low (0.1 %) and high (1 %) fuel sulfur content ships with a probability of 80–90 % (Van Roy, 2016).

From the spectra measured by our MAX-DOAS UV instrument both $SO_2$ and $NO_2$ columns can be retrieved at once. The two columns are measured at the exact same time along nearly the same light path. To calculate $SO_2$ to $NO_2$ ratios for the measured pollutant peaks simply the ratio of the measured differential slant column densities has to be computed.

In order to separate ship related signals from smooth background pollution, first a running median filter was applied to the time series of $NO_2$ and $SO_2$ measurements with a large kernel size (e.g. over 21 points). If too many broad peaks are contained in the time series this is not sufficient and the resulting median might be systematically higher than the actual baseline. In this case, on the values in the lower 50 % quantile again a running median with a smaller kernel size (e.g. 5) was applied, giving a good approximation of the real baseline.

In the next step, this baseline is subtracted from the raw signal. A simple peak detection algorithm was used to identify the peaks in the baseline-corrected $NO_2$ signal. Then the corresponding peaks in the $SO_2$ were assigned, thus accounting for cases when no $SO_2$ enhancement is measured. In a final manual checkup, all the identified peaks were looked through, filtering out for example all the cases when peaks are too close together to be separated and fine-tuning the baseline detection algorithm parameters if necessary.

To achieve a better signal-to-noise ratio, the integrals over both the $NO_2$ and $SO_2$ peak are calculated and the ratio of both values is computed in the last step.

Figure 18 shows the approach as well as the results for an example day in summer 2014, before the stricter fuel sulfur content limits were introduced. Both the $NO_2$ and $SO_2$ signal show high and sharp peaks, originating from ship plumes. Most of the peaks are of similar shape in $NO_2$ as well as $SO_2$ signal. The measured $SO_2$ to $NO_2$ ratios lie in the range from 0.17 to 0.41. The $SO_2$ to $NO_2$ ratio can vary strongly for different ships. For example, the plume of the ship passing the line-of-sight around 12:00 UTC has a high $NO_2$ content, but is low in $SO_2$, whereas the opposite is true for the ship passing at 12:30 UTC, indicating that the second ship was

using fuel with a considerably higher sulfur content than the first one.

Figure 19 shows one example day in summer 2015, after the establishment of stricter sulfur limits. For better comparison to Fig. 18, the y-axis limits are the same. High NO2 peaks also occur on this day. However, the $SO_2$ signal shows no clearly distinguishable peaks anymore, a result of much less sulfur in the fuel. Consequentially, the measured $SO_2$ to $NO_2$ ratios are much smaller on this day and range from 0 to 0.09. There might be some small peaks in the $SO_2$ signal, but for most of them it cannot be determined if these are real enhancements or just noise fluctuations. The two peaks at 10:40 and 14:00 UTC, slightly above noise level but still very small, might be real $SO_2$ signals from ships with a higher than average fuel sulfur content.

For a statistically meaningful comparison of both time periods two representative samples of ship emission peaks have been selected by hand for days with good measurement conditions, which were identified by using the solar radiation measurement data of our weather station. One sample of more than 1000 peaks, measured in 2013 and 2014 representing the state before introduction of stricter fuel sulfur content limits, and another equally-sized sample of more than 1000 peaks measured in 2015 and 2016, representing the situation afterwards, were analyzed in a semi-automatic way. It has to be noted that it cannot be ruled out that a certain fraction of ships were measured repeatedly on different days. It is also highly probable that the plume from some individual ships was measured multiple times at different locations in the different azimuth directions while the ship was passing the island.

The distributions of the $SO_2$ to $NO_2$ ratios derived from the peak integrals for the two samples are shown in a histogram in Fig. 20. It can be seen that $SO_2$ to $NO_2$ ratios were considerably higher before 2015, with a mean of 0.30, a standard deviation of 0.13 and a median value of 0.28. After the change in fuel sulfur content limits, the $SO_2$ to $NO_2$ ratios became much lower with a mean of 0.007, a standard deviation of 0.089 and a median value of 0.013, a drastic reduction. A Welch's t-test (unequal variances t-test) shows that the reduction is statistically highly significant. These results can be compared to the overall average $SO_2$ to $NO_2$ ratios on all days with good measurement conditions from which the peaks have been selected: For the time before 2015, this gives a mean value of 0.10 and a median of 0.17 and for 2015 and 2016, one gets a mean value of 0.024 and a median of 0.058. As expected, these values are significantly lower than the $SO_2$ to $NO_2$ ratios obtained from the ship plumes which do not include background pollution.

It is also interesting to compare our results with those from other studies, bearing in mind possible systematic differences due to different measurement geometries, techniques and sites and therefore different NO to $NO_2$ titration in the plumes.

McLaren et al. (2012) measured $NO_2$ to $SO_2$ emission ratios in marine vessel plumes in the Strait of Georgia in summer 2005. In a sample of 17 analyzed plumes, a median molar $NO_2/SO_2$ ratio of 2.86 was found. Translated into a $SO_2/NO_2$ ratio this yields a value of 0.35 which is, considering the small sample size, in good agreement with our findings for the time before 2015.

Another study was carried out by Diesch et al. (2013) measuring gaseous and particulate emissions from various marine vessel types and a total of 139 ships on the banks of the river Elbe in 2011. $SO_2$ to $NO_2$ emission ratios can also be derived from from their reported $SO_2$ and $NO_2$ emission factors: For small ships ($< 5\,000$ tons) a ratio 0.13 and an average fuel sulfur content (FSC) of $0.22 \pm 0.21\,\%$ was found, for medium size ships ($5\,000$–$30\,000$ tons) a ratio of 0.24 and a FSC of $0.46 \pm 0.40\,\%$ and for large ships ($> 30\,000$ tons) a ratio of 0.28 and a FSC of $0.55 \pm 0.20\,\%$. Especially the values for medium size and large ships fit quite well to our results while plumes from very small vessels (if measurable at all) have often not been taken into account for the statistic because of the low signal-to-noise ratio.

When assuming that the dependency of $SO_2$ to $NO_2$ ratio to fuel sulfur content is also applicable to our dataset, we can roughly estimate that the ships measured by us before 2015 used an average sulfur content of 0.5–0.7 %, in good agreement with the results of Kattner et al. (2015), which since 2015 decreased drastically with 0.1 % as an upper limit.

## 5  Conclusions

In this study, three years of MAX-DOAS observations of $NO_2$ and $SO_2$ taken on the island of Neuwerk close to the shipping lane towards the harbor of Hamburg, Germany were analyzed for pollution emitted from ships. Using measurements taken at $0.5°$ elevation and different azimuthal directions, both background pollution and plumes from individual ships could be identified. Using simultaneously retrieved $O_4$ columns, path averaged volume mixing ratios for $NO_2$ and $SO_2$ could be determined. Comparison of $NO_2$ measurements in the UV and visible parts of the spectrum showed excellent agreement between mixing ratios determined from the two retrievals, demonstrating consistency in the results.

MAX-DOAS measurements were also compared to co-located in-situ observations. High correlation was found between mixing ratios derived with the two methods on average, in-situ measurements showing systematically larger values, in particular during ship emission peaks. These deviations can be understood by the difference in measurement volume, the MAX-DOAS measurements averaging over light paths of several kilometers and a systematic underestimation of MAX-DOAS VMRs due to different profile shapes of $O_4$ and the pollutants $NO_2$ and $SO_2$. For $NO_2$, the difference is larger than for $SO_2$, probably because of conversion of NO to $NO_2$ during the transport from the ship where the signal is

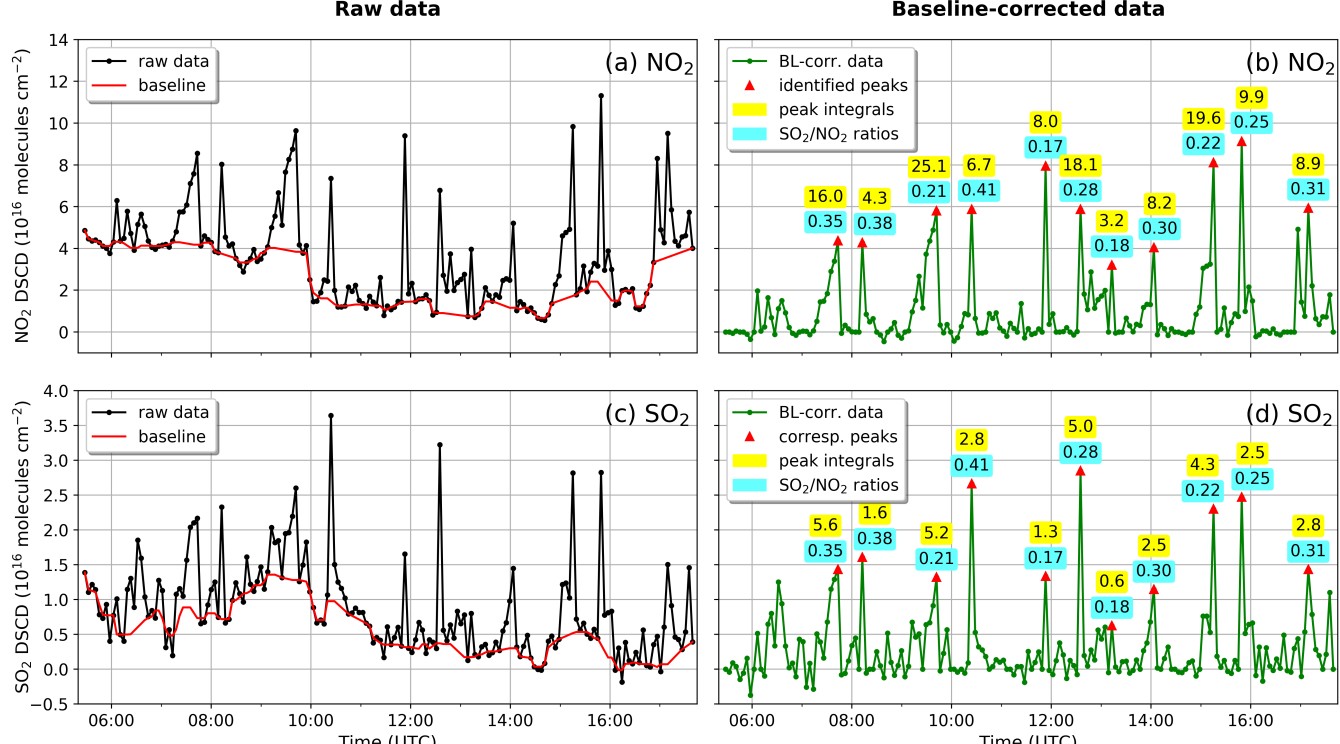

**Figure 18.** Calculation of $SO_2$ to $NO_2$ ratios for ship emission peaks for one example day (23 July 2014) before the change in sulfur emission limits. Panel **(a)** shows the UV $NO_2$-DSCD raw data for $0.5°$ elevation and $-25°$ azimuth and the determined baseline. Panel b shows the baseline-corrected $NO_2$ data for which the automatically identified peaks are highlighted with red triangles. Numbers close to the peaks denote the peak integrals in $10^{16}$ molecules/cm$^2$ (marked in yellow) and the $SO_2$ to $NO_2$ ratios (marked in blue). Panels **(c)** and **(d)** show the corresponding plots for $SO_2$.

detected by MAX-DOAS to the measurement site where the in-situ instrument was located.

Although the measurement site is within a few kilometers from one of the main shipping lanes, it is influenced by land based pollution depending on wind direction. Comparing measurements taken under wind direction from the shipping lane and from land, systematic differences in the diurnal and weekly cycles of $NO_2$ are found. While $NO_2$ from land shows high values in the morning and evening and lower values around noon and on weekends, $NO_2$ levels from sea are more or less constant over time as expected from continuous shipping operations. These results are found in both MAX-DOAS and in-situ observations. Both $NO_2$ and $SO_2$ levels are often higher when wind is coming from land, indicating that land based sources contribute significantly to pollution levels on the island in spite of its vicinity to the shipping lanes. Analyzing the wind dependence of the signals in more detail, and excluding data with mixed air mass origin, the contribution of shipping sources to pollution on Neuwerk could be estimated to be 40 % for $NO_2$ and 41 % for $SO_2$ in the years 2013 and 2014. As nearly half of the measurements were taken under wind coming from mixed directions, this is only a rough estimate but is still a surprisingly small fraction.

Although the MAX-DOAS measurements cannot be used to directly determine $NO_x$ or $SO_2$ emissions from individual ships due to the measurement geometry, the ratio of $SO_2$ to $NO_2$ column averaged mixing ratios gives a good estimate of the $SO_2$ to $NO_x$ emission ratio. Using the data from Neuwerk, more than 2000 individual ship emission plumes were identified and the ratio of $SO_2$ to $NO_2$ computed after subtraction of the background values. The results varied between ships but on average yielded values of about 0.3 for the years 2013/2014, in good agreement with results from other studies.

Since January 2015, much lower fuel sulfur content limits of 0.1 % apply in the North and Baltic Sea. This resulted in large changes in $SO_2$ levels in the MAX-DOAS measurements when the wind is coming from the shipping lanes. In fact, ship related $SO_2$ peaks are rarely observed anymore since 2015. Applying the same analysis as for the period before the change in legislation, no significant changes were found for $NO_2$ in terms of ratio between ship and land contribution or absolute levels. For $SO_2$ in contrast overall levels were reduced by two-thirds, and the relative contribution of shipping sources was reduced from 41 % to 14 %. It is interesting to note that a reduction in $SO_2$ levels was also

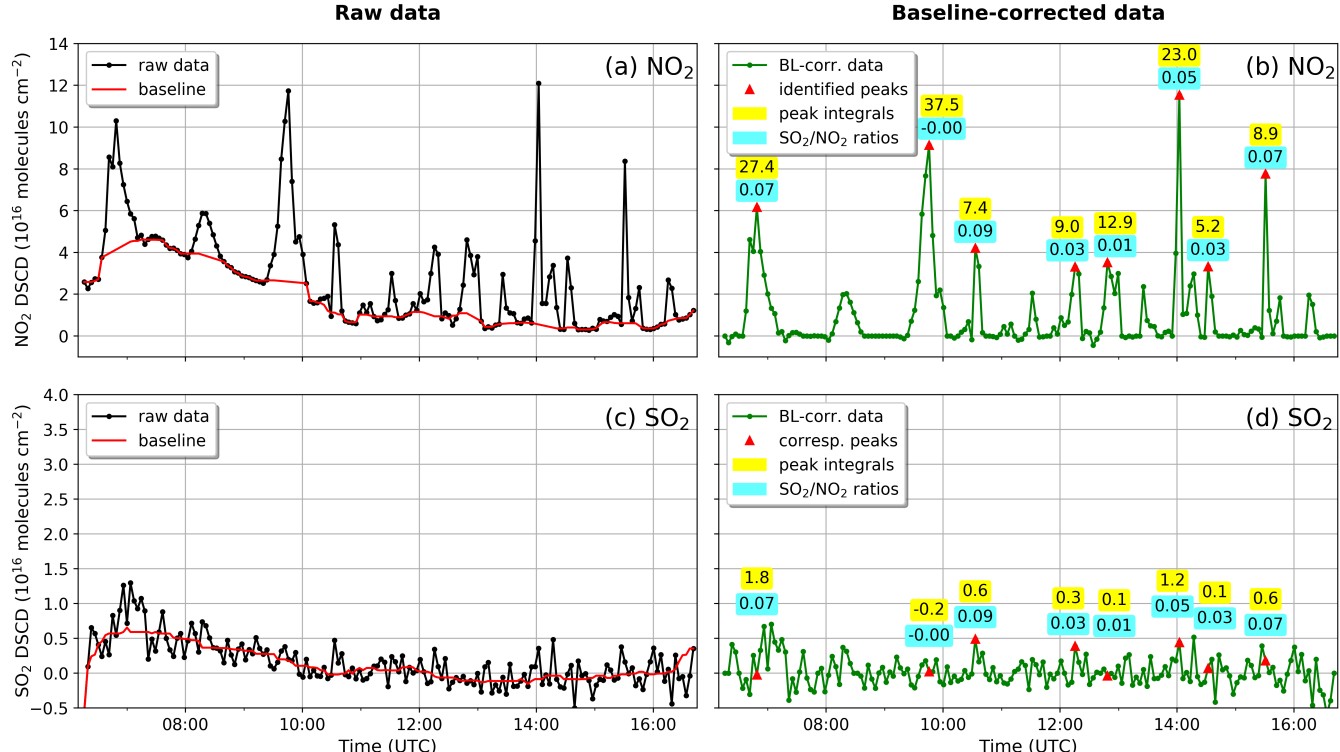

**Figure 19.** As Figure 18 but for an example day (3 July 2015) after the introduction of stricter fuel sulfur content limits. Measurements in $0.5°$ elevation and $65°$ azimuth are shown. Peak integrals are given in $10^{16}$ molecules/cm$^2$.

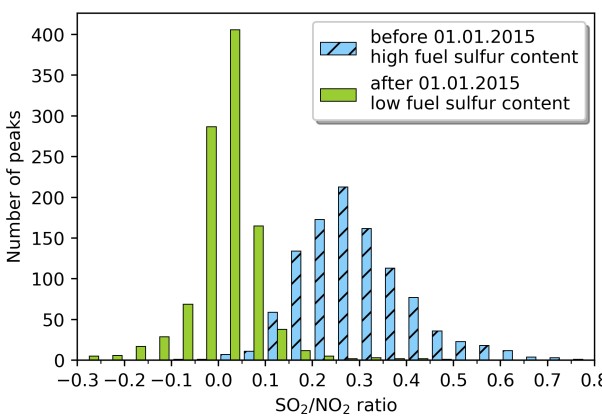

**Figure 20.** Histogram showing the distribution $SO_2$ to $NO_2$ ratios in two samples ($N = 1055$ for each) of ship emission peaks measured in $0.5°$ elevation and all azimuth angles for the time before (blue) and after (green) the change in fuel sulfur content regulation on the 1st of January 2015.

observed in most wind directions coming from land, presumably because shipping emissions also contributed to $SO_2$ levels in coastal areas.

In summary, long-term measurements of $NO_2$ and $SO_2$ using a MAX-DOAS instrument demonstrated the feasibility of monitoring pollution originating from ships remotely. Pollution signals from individual ships can be identified and path averaged mixing ratios can be determined, which on average correlate well with in-situ observations, reproducing day-to-day trends. MAX-DOAS measurements do not provide emission estimates for individual ships but allow statistical analysis of signals from thousands of ships at a distance and even under unfavorable wind conditions. Implementation of stricter sulfur limits in shipping fuel lead to a large reduction in $SO_2/NO_x$ ratios in shipping emissions and a significant reduction in $SO_2$ levels at the German coast. The amounts of $NO_2$ are as expected not significantly impacted by the change of sulfur content in the fuel. This implies that combustion temperatures were probably not significantly changed. The overall contribution of ship emissions to pollution levels at the measurement site is large but land based sources still dominate, even in the immediate vicinity of shipping lanes.

*Data availability.* The data used in this study are available from the cited references and directly from the authors upon request.

*Competing interests.* The authors declare that they have no conflict of interest.

*Acknowledgements.* The research project which facilitated the reported study was funded in part by the German Federal Maritime and Hydrographic Agency (Bundesamt für Seeschifffahrt und Hydrographie, BSH) and the University of Bremen. The authors thank the Waterways and Shipping Office Cuxhaven (Wasser- und Schifffahrtsamt, WSA) and the Hamburg Port Authority (HPA) for their help and support. Many thanks to the Co-Editor, Robert Harley, and to two anonymous referees for their valuable comments and suggestions, which helped to improve this publication.

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
