# Peer review of "Monitoring shipping emissions in the German Bight using MAX-DOAS measurements"

_Atmospheric Chemistry and Physics, 2016_

## Referee Comment (RC1) · Anonymous Referee #1 · 23 Feb 2017

**General comments**

The manuscript entitled "Monitoring shipping emissions in the German Bight using MAX-DOAS measurements" presents remote sensing observations of NO2 and SO2 along a main shipping route towards the harbour of Hamburg. Ship emissions significantly contribute to the air pollution in these areas, and a monitoring of the air quality, in particular with respect to the impacts of the recent regulations of the fuel sulphur content, is of high scientific and political relevance. Therefore the topic of the manuscript is well suited for publication in ACP.

The paper is well structured and the scientific approach is clearly described. It provides a comprehensive introduction into the subject of ship emissions. The data is interpreted regarding the contribution of land- and seaborne emissions in a systematic way and the

impact of the reduction of fuel sulphur content on atmospheric SO2 levels is discussed on the basis of statistical analyses. However, there are several aspects regarding the interpretation of the data which need to be revised. In particular, my impression is that the impact of horizontal inhomogeneities on the measurements and the fact that the remote sensing measurements average over a certain altitude range need to be considered more carefully. It appears to me that the latter is the main reason for the discrepancy between MAX-DOAS and in situ, which can and should be corrected for by accounting for the different vertical distribution of O4 and the target gases and thus different AMFs, as done during previous studies (e.g., Sinreich et al., AMT, 2013).

**Specific comments**

L56: Do you refer to fuel consumption and emission per vessel or in total (the latter would be obvious given the large increase in the number of ships).

Section 1.2:

Maybe the discussion of halogen chemistry should be removed since it is not of relevance for the present study. To my knowledge, the role of halogen radicals in polluted air is not well understood, and it is unclear whether the NO + XO reaction is of importance. In clean air, the conversion of XO to X proceeds either via self-reaction or reaction with HO2. In polluted air, reaction with NOx is likely to lead to a removal of halogen radicals by formation of halogen nitrates.

Section 3.1:

It could be mentioned that Equation (2) follows from (1) if the temperature and pressure dependence of the absorption cross section can be neglected.

Section 3.2:

Please provide technical information on the fibre bundle (number of fibres, diameter, arrangement at both sides).

What are the wavelength ranges of both spectrographs?

I am confused about the definition of the elevation angle. Usually, it should be the centre of the field of view, but here the definition is unusual and rather unspecific, being something like the lower edge of the field of view, which yields an offset to the common definition of $0.6°$. Please specify the elevation angle as the centre of the field of view throughout the paper.

Section 3.3:

It is important to include a discussion of the fit errors, detection limits and RMS residual for the target gases.

L345ff: Apart from the ozone absorption, a limitation for a retrieval window at shorter wavelengths is the lower light intensity.

Section 3.4:

The definition of the volume mixing ratio and its calculation from number concentration is well known and there is no need to discuss this here.

The uncertainties of the O4 scaling approach need to be discussed. For example, O4 usually has a profile shape very different from NO2 and SO2, which violates the basic assumption that the O4 dSCD is a good proxy for the light path through the NO2 and SO2 layers. Other studies use correction factors from radiative transfer calculations to account for this (Sinreich et al., AMT, 2013). Furthermore, the resulting near-surface volume mixing ratios will not be representative for the amount of trace gases directly at the surface, but for some kind of average over a certain height range in the boundary layer. There is also "light dilution", i.e. light scattered into the line of sight between the instrument and the trace gas plume (see e.g. Kern et al., Bulletin of Volcanology, 2010), which further reduces the measured SCDs . My impression is that these are the main reasons for the discrepancies between in situ and MAX-DOAS, and not horizontal inhomogeneities as speculated later in the paper by the authors (these would

cancel out when averaging the data). The discussion of the data needs to be revised accordingly in order to account for the influence of these aspects.

Equation 5: It should be mentioned that $n_{O_4}$ is the $O_4$ concentration at the surface.

The remarks regarding the elevation angle from Section 3.2 are repeated at the end of this section. See my comments above. A deviation of $0.5°$ in elevation angle is certainly not negligible at very low elevation angles.

Section 4.1:

L474: It is not obvious to me why a thicker trace gas layer should lead to a reduction of the ratio between dSCDs near the horizon and in zenith. Wouldn't horizontal inhomogeneities, with more NO2 over the shipping lane than over the instrument, be a much more likely explanation for these findings?

Section 4.2:

The title of this section is too long and complicated. I suggest to replace it by something like "Volume mixing ratios of NO2 and SO2"

Section 4.5:

As already stated above, the fact that MAX-DOAS averages over a large horizontal distance should cancel out on temporal average when comparing to in situ measurements. Instead, a more probable explanation for the systematically lower mixing ratios is the fact that MAX-DOAS averages over a certain altitude range and that the differences in O4 and target gas profile shapes has not been considered. Light dilution will also play a certain role. The argument that MAX-DOAS yields lower values when the plume is orthogonal to the viewing direction does not seem convincing to me, because in this case the polluted air is also not transported towards the in situ instrument, which means that the in situ instrument might even miss particular plumes which are detected by MAX-DOAS.

**Section 4.6:**

L655ff: NO2 concentrations at a particular location strongly depend on local sources, such as traffic, industry, domestic heating, etc., as well as on the distance to these sources and on the rate of vertical mixing. Therefore, the fact that amount of NO2 in background air observed in the Arctic is similar to the present study might be mere coincidence.

**Section 4.7:**

L667: Detection limits are mentioned here for the first time. They should instead be discussed in Section 3.3.

**Section 4.8:**

An attempt is made here to separate shipping emissions from other sources by classifying the data according to the wind direction. The limitations of this approach need to be discussed more carefully. While I agree that northerly winds are little affected by background pollution, I strongly doubt that shipping emissions do not influence the measurements significantly when the wind is coming from the south. Data is filtered for light paths longer than 5 km, which means that for most observations the light path crosses the main shipping lane and probes air polluted by ship traffic. You reach this conclusion yourself in Section 4.1 (L438ff) in the context of the discussion of Figure 6, which shows that peaks from ship emissions clearly occur when air polluted by land-based sources is present. Thus, air masses classified as "Land" are likely to be partly affected by ship emissions.

**Section 4.9:**

Given that SO2 scatters around a smooth (near-zero) background level, it is surprising to see that no negative SO2 to NO2 ratios were derived. It seems that negative values have been set to zero (Panel D of Fig. 20), which would significantly (and falsely) affect the statistics.

**Technical corrections**

Equations: Please use single characters for variables (e.g., "$S$" instead of "SCD", "$R$" instead of "RESIDUAL", "$H$" instead of "MLH"). There is a difference between an abbreviation (e.g., "SCD" for slant column density) and the according mathematical symbol (e.g., "$S$").

L11: Provide a number for the distance between measurement site and shipping lane

L13: The fact that the site is close to the shipping lane is repeated. Delete "which is a few kilometres from the shipping lane"

L17: "retrieved from NO2 retrievals" -> "determined from NO2 retrievals"

L54: "... from around 31 000 .... over 52 000 ... to 89 000 ..."

L68: "... molecular nitrogen (N2) and oxygen (O2) ..."

L81: comma after "radicals"

L81: "... hydroperoxyl (HO2) or organic peroxy radicals (RO2) or halogen oxides (XO, were X = Cl, Br or I)

L84: "X atoms" -> "halogen atoms"

L85: "reacts" -> "react"; "reaction" -> "reaction rate"

L86: "Owing to the lack of photolysis, NO reacts rapidly .... during the night"

L87: "In addition, the nitrate radical (NO3) is formed..."

L110: Comma after "regions"

L112: "ecosystem" -> "ecosystems"

L121: Put "3" in "m3" into superscript

L162: Comma after "emissions"

L173: Comma after "example"

L206: Incomplete sentence. Replace, e.g., by ". . . first the measurement site is described, followed by a presentation of the wind statistics and data availability."

L215: ". . . were taken on Neuwerk, a small island in the North Sea with the size of . . ."

L221: "island of Neuwerk" or simply "Neuwerk" (here and anywhere else). Delete "where our measurement site is located" (repetition)

L225: Do you refer to a specific document from the "Statistische Ämter" or can you provide an url to the data?

L224: Is this height above sea level?

L248: "site for the measurements" -> "site"

L285: To "inject" light into the fibre sounds strange since this term suggests that the light is somehow transported actively. Replace by something like "focused on the entrance of the optical fiber"

L286: "opening angle" -> "field of view"

L307: Define what "SCD1" and "SCD2" refer to. Replace by variable names consisting of single letters.

L316: It should be mentioned that a spatially limited plume directly over the instrument leads to an underestimation of the retrieved dSCDs.

Table 1: Only list the polynomial degree, not the number of coefficients.

L399: "filtered" -> "discarded"

L439: Delete "the pollutant"

L441: ". . . difference between NO2 in the UV (red curve) and in the visible (blue curve). . ." (the discussion is about NO2 and not about the colors of the curves)

L442: "more intense" -> "stronger"

L445: "By comparing SO2 (black curve) with NO2 (red and blue curves), it can be seen. . ."

L447: Delete "A more dirty"

L454: "points in time" -> "times"

L497: "The difference between UV and visible peak values depends. . ."

L498: "A short distance of the plume to the instrument and its complete coverage by the shorter UV path leads to higher values in the UV. . ."

The title of section 4.3 does not make sense. It implies that the approach is statistically evaluated. Instead, the data is statistically evaluated. Replace with something like "Statistical evaluation of the NO2 and SO2 data"

L507: ". . .all single pairs of simultaneous measurements" -> "all single pairs of DSCD measurements. . ."

L508: "the left subplot in the upper row" -> "Panel A"

L509: "both measurements" -> "NO2 and SO2 DSCDs"

L513: "The right subplot in the upper row" -> "Panel B"

L518: "the left subplot in the bottom row" -> "Panel C"

L523: "(right plot)" -> "(Panel D of Fig. 9)"

L533: "applied on mountains" -> "applied to mountain-based measurements"

L535: Delete "However"

L537: "This should lead" -> "This leads" (the enhancement in path length in a cleaner and less dense atmosphere is obvious)

L541: "various" -> "detailed" or "comprehensive"

L543: Delete "emitting"

Figure 10: Mark the three panels as "A", "B" and "C" (from top to bottom)

L544: "Measurements from Wednesday, 9 July 2014 are shown in Figure 10. Panel A shows..."

L545: "The middle one" -> "Panel B"

L549: "The lower sub-plot" -> "Panel C"

L563: "The differences of both measurement techniques need to be considered for such a comparison:"

L565: "at one point" -> "at a single location"

L568: Insert "the" before "line-of-sight"

L569: delete "line-of-sight" (it is already mentioned at the beginning of the sentence)

L579: Delete "From the Figure, it can be easily identified that"

L581: Delete "nicely"

L586: Delete "It is also clearly visible, that"

L591: "it's" -> "its"

L592: Delete the first sentence of this paragraph

L594: "the upper subplots" -> "Panel A"; Add ", respectively" to the end of the sentence.

L606: "makes no sense" -> "is of little use"

L603: "the lower subplot" -> "Panel B"

L613: Delete "As can be seen in the figures"; delete "usually"

L614: What do you mean with "progression of both curves"?

L623: Insert comma after "combustion"

L643: "The mean NO2 volume mixing ratios for each weekday shown in Fig. 16 illustrate the influence of land-based road traffic."

L647: "There is only little weekly cycle for air masses coming from the open North Sea. Measurements . . ."

L665: "single day measurements" -> "Single day of measurements"

L704: Delete comma after "This implies"

L738: "like expected" -> "as expected"

L743: "It can be seen that this increase for the land source sector is only a relative increase by comparing. . ."

L765: "roll" -> "role"

L767: "A monitoring of emissions from single ships requires the analysis of individual plume peaks in the NO2 and SO2 data sets."

L780: I am not familiar with the term "emission factor". Do you mean "emission rate"?

L780: delete "both"

L796: "one can get rid of the background pollution" -> "the background pollution can be removed"

L801: "To achieve a better signal-to-noise ratio, the integrals . . .. in the last step"

L803: "one" -> "an"

L804: "In both the NO2 and SO2 signal" -> "Both the NO2 and SO2 signal show"

L805: delete "are visible"; delete "measured"; "The shape of the peaks is also often

quite similar" -> "Most of the peaks are of similar shape"

L807: "The SO2 to NO2 ratio can vary strongly for different ships. For example, the plume of the ship passing the line of sight around 12:00 UTC has a high NO2 content, but is low in SO2, whereas the opposite is true for the ship passing at 12:30 UTC, indicating that the second ship. . ."

L811: Delete "In contrast to this,"

L813: "High NO2 peaks also occur on this day. However,. . ."

L818: "From this plot one can also see that" -> "As can be seen from this plot, "

L819: "overestimate" -> "overestimates"

L826: "retrieved" -> "analyzed"

L836: Insert comma before "indicating"

L839: "and for 2015 and 2016, one gets a mean value of . . . " -> ", and a mean value of . . . for 2015 and 2016"

L842: "leading to overestimation" -> "leads to an overestimation"

L850: "from" -> "by"

L851: "SO2 and NO2 emission ratios can also be derived from. . ."

L858: "the dependency of SO2 to NO2 ratio to fuels sulfur content"

L863: "Island Neuwerk" -> "Island of Neuwerk"

L865: "into" - > "and"

L871: Delete "also"

L882: "NO2" -> "daily averaged NO2"

L908: Insert "can" after "ratios"

---

## Referee Comment (RC2) · Anonymous Referee #2 · 11 Apr 2017

General comment: This paper describes a 3 year series of multi axis DOAS measurements carried out from the German island Neuwerk, just south of the entry to the river Elbe. This is in the main ship channel of the port of Hamburg and the main aim of the measurements was to study these by observing UV and visible light horizontally towards the ship channel. The paper is well written, with good language and instructive graphs. The paper is a nice combination of measurements methodology and results paper. It shows the methodology to measure mixing ratios in a coastal places, together with ship plume measurements and some results about the effect of new IMO legislation. However, the OBJECTIVE and AIM should be declared more clearly in the text. The paper is also rather long, and I would recommend to shorten it, by removing sections which are outside the main scope of the paper. Forinstance merging and shortening sect 4.5 and 4.6 corresponding to mixing ratio measurements

and comparisons. .All in all, I believe the paper should be published, with some minor improvements, based on answering my specific comments below:

Specific comments: Row 71, p 2: It is claimed that 25% of the NOx emerges as NO2 from the stack, but usually 10% is assumed from fluegas stacks; please give more details: I assume you also assume some titration?

Row 278, p 9: IN the equation do you fit differential absorption cross sections or the absolute ones? Since you are using prime I assume you mean the differential ones; IN

row 336 I however get the impression that you use the absolute ones.

Row 311, p 10: It is claimed that the vertical paths cancels out between path 1 and 2 in Fig 5; I agree with the stratospheric portion but for the tropospheric part there should be a cos (SZA) difference, even if NO2 is homogenously distributed in the troposphere?

Row 387, p 13: Is it assumed that the wavelength difference in O4 signal is linear; if so what are the uncertainties involved?

Row 406, p 14: It is claimed that the conditions at the Neuwerk radar tower is similar to measurements from high mountains; please motivate better. Eq 4, p 13: It is difficult to follow how you get the expression in eq 4.

Row 464 p 16: On this place, and some others, its is claimed that the differential slant columns are higher for SSE and ESE and (more elevated). But part of this should be wind speed effect since I would imagine that the wind speed will be higher from the sea and this will dilute the slant columns more. Has this been investigated ?

Row 470 p 16: Graph 7 is not totally clear. If I understand right the plot correspond to overlayed windroses for different elevation angles rather than that the area of each color represents the wind rose information. I interpreted the latter and I think this should be clarified forinstance in the figure text.

Row 500 p 17: You here discuss the results in Fig 8. The differences in the UV and

visible are explained from the penetration length, but should the Visible not in general be higher since it gives the chance of penetrating plumes further away, rather than the opposite which appears to be the case for all plumes here? You explain that the UV should be stronger for close by plumes since a higher fraction of the photons are then affected by absorption. Is is not so that the O4 can only simulate slow variations? Please elaborate..

Row 614 p 25: You claim that fig 12/fig 13 shows good agreement between MAX DOAS and in situ, but in my mind this is the case for Fig 13 but not for fig 12, where there appears to be rather big difference in he averages of the two sensors with factor 2-3?

Row 665 p 27: You suddenly refer to fig 20, without having mentioned fig 17-19 yet in the text. You should consider reordering.

Row 891 p 35: As concluded here and discussed in section 4.9, the ratio of SO2/NO2 gives an indication of sulfur fuel content in ship plumes. Are you aware that SO2/NO2 ratio measurements from airborne DOAS is used operationally since 2015 by Beecken and Mellqvist (Chalmers University) in the CompMon project and surveillance around Denmark and that this has been presented on several official workshops last year? You mention that the +2015 measurements are biased by noise since you don't really observe any SO2 then. I don't think it then makes sense to show the green data (+2015) in figure 5 since these histograms then only represent noise? Secondly you don't mention when comparing to other measurements that the amount of NO to NO2 titration is very important for the ratio, and this will depend on the distance to the plume, whether you are over land or sea etc. Please add some discussion on this.

Row 903 p 35: It is mentioned that there are still SO2 coming from land. This is surprising since there are very few SO2 emission sources anymore and power plants generally have abatement equipment. It would be interesting to understand this better ?

Technical Corrections: Well written in most places. Row 812 p 31: Change limis to

limits Row 873 p 33: Change This to These

---

## Author Comment (AC1) · 6 Jun 2017

The comment was uploaded in the form of a supplement:
http://www.atmos-chem-phys-discuss.net/acp-2016-1153/acp-2016-1153-AC1-supplement.pdf

---

## Author Response (AR1)

**Letter to the Editor**

Bremen, 23.06.2017

Dear Robert Harley,

you kindly accepted the editorship of our manuscript "Monitoring shipping emissions in the German Bight using MAX-DOAS measurements" and we would like to take the opportunity to thank you for the consideration of your work.

We individually answered point-by-point to all comments and questions of Referee #1 and Referee #2. We revised the original manuscript according to their suggestions and provided additional information the referees asked for. Below, you find again the answers to the referees that we also uploaded to the ACP web page. We also provide here a version of the revised manuscript in which changes in comparison to the initial version are marked color-coded. In addition to that, a version of the manuscript in the Copernicus two-column style (using the Copernicus Latex-Template) is attached. For this version, a few small changes have been applied to comply with the Copernicus Manuscript Preparation Guidelines (e.g. changing panel names from uppercase to lowercase letters, adjustments in figure sizes for the two column layout).

We hope that with the submission of the author's comments and the revision of the manuscript, our article will be accepted for publication in ACP.

Yours sincerely,

André Seyler

List of Attachments

- Author comments to Referee #1
- Author comments to Referee #2
- Revised manuscript
- Revised manuscript with color-coded changes
- Revised manuscript in Copernicus two-column style (Copernicus Latex-Template)

The manuscript entitled "Monitoring shipping emissions in the German Bight using MAX-DOAS measurements" presents remote sensing observations of NO2 and SO2 along a main shipping route towards the harbour of Hamburg. Ship emissions significantly contribute to the air pollution in these areas, and a monitoring of the air quality, in particular with respect to the impacts of the recent regulations of the fuel sulphur content, is of high scientific and political relevance. Therefore the topic of the manuscript is well suited for publication in ACP.

The paper is well structured and the scientific approach is clearly described. It provides a comprehensive introduction into the subject of ship emissions. The data is interpreted regarding the contribution of land- and seaborne emissions in a systematic way and the impact of the reduction of fuel sulphur content on atmospheric SO2 levels is discussed on the basis of statistical analyses. However, there are several aspects regarding the interpretation of the data which need to be revised. In particular, my impression is that the impact of horizontal inhomogeneities on the measurements and the fact that the remote sensing measurements average over a certain altitude range need to be considered more carefully. It appears to me that the latter is the main reason for the discrepancy between MAX-DOAS and in situ, which can and should be corrected for by accounting for the different vertical distribution of O4 and the target gases and thus different AMFs, as done during previous studies (e.g., Sinreich et al., AMT, 2013).

First, we would like to thank Anonymous Referee #1 for his/her helpful comments. Below, we reply point-by-point to the specific comments . As far as possible, we have considered the suggestions in the revised manuscript.

**Specific comments**

L56: Do you refer to fuel consumption and emission per vessel or in total (the latter would be obvious given the large increase in the number of ships).

We (and the cited studies) are referring to total fuel consumption and emissions. Following your suggestion, we included the word "total" to make this clearer:

"At the same time, *total* fuel consumption and emissions increased as well (Corbett and Koehler, 2003; Eyring et al., 2005a,b; Eyring et al., 2010b)."

Section 1.2:
Maybe the discussion of halogen chemistry should be removed since it is not of relevance for the present study. To my knowledge, the role of halogen radicals in polluted air is not well understood, and it is unclear whether the NO + XO reaction is of importance. In clean air, the conversion of XO to X proceeds either via self-reaction or reaction with HO2. In polluted air, reaction with NOx is likely to lead to a removal of halogen radicals by formation of halogen nitrates.

We removed the corresponding paragraph.

Section 3.1:
It could be mentioned that Equation (2) follows from (1) if the temperature and pressure dependence of the absorption cross section can be neglected.

Thank you very much for your thorough proof-reading, we had forgotten to include this important information. The corresponding paragraph now reads:

"Neglecting the temperature and pressure dependence of the absorption cross sections, polynomial and differential cross sections are fitted to the measured optical thickness ln (I/I0) in the linearized so-called DOAS equation:"

Section 3.2:
Please provide technical information on the fibre bundle (number of fibres, diameter, arrangement at both sides).

Type: Y-shaped quartz fiber bundle
Number of fibers: 2 x 38 = 76 single fibers
Diameter: 150µm each,
Length: 20m
Arrangement sketch:

[Figure]

Telescope side:
76 fibers,
circular arranged

18 m

2 m

Spectrometer side:
38 fibers, in-line
arrangement

We added this information to the text of the manuscript.

What are the wavelength ranges of both spectrographs?

We added the following sentence:

"The UV instrument covers the wavelength range 304.6-371.7 nm, the visible spectrometer covers 398.8-536.7 nm."

I am confused about the definition of the elevation angle. Usually, it should be the centre of the field of view, but here the definition is unusual and rather unspecific, being something like the lower edge of the field of view, which yields an offset to the common definition of 0.6_. Please specify the elevation angle as the centre of the field of view throughout the paper.

We changed the elevation angles to the actual values (center of field-of-view) throughout the paper.

Section 3.3:
It is important to include a discussion of the fit errors, detection limits and RMS residual for the target gases.

We added a paragraph on this in Section 3.3:

"Under good conditions, the typical fit RMS is around $1\times10^4$ for $NO_2$ in the visible, $2\times10^4$ for $NO_2$ in the UV and $5\times10^4$ for $SO_2$. By assuming that an optical density of twice the RMS can be detected (Peters, 2013), it is possible to estimate the detection limit of our instrument regarding the different trace gases. The differential absorption cross section of NO2 is in the order of $1\times10^{-19}$ $cm^2$/molec, for $SO_2$ in the order of $2\times10^{-19}$ $cm^2$/molec. Combining this yields a $NO_2$ detection limit of around $1\times10^{15}$ molec/$cm^2$ corresponding to 0.05 pbb in the visible and $2\times10^{15}$ molec/$cm^2$ corresponding to 0.1 pbb in the UV. The $SO_2$ detection limit is around $2.5\times10^{16}$ molec/$cm^2$ corresponding to 0.2 pbb.

The typical absolute fit errors are 2-3 * 10^14 molec/$cm^2$ for $NO_2$ in the visible,  5-6 * 10^14 molec/$cm^2$ for $NO_2$ in the UV and 2 * 10^15 molec/$cm^2$ for $SO_2$, which is a factor of 5 to 10 smaller than the detection limit."

L345ff: Apart from the ozone absorption, a limitation for a retrieval window at shorter wavelengths is the lower light intensity.

This sentence was changed to the following:

"This results from the need to find a compromise between *the low light intensity caused by* the strong ozone absorption around 300nm on the one hand and the rapid decrease of the differential absorption of $SO_2$ at higher wavelengths on the other hand, limiting the choice of the fitting window."

Section 3.4:
The definition of the volume mixing ratio and its calculation from number concentration is well known and there is no need to discuss this here.

We removed the unnecessary explanation of volume mixing ratios and have completely rewritten this section focussing on the $O_4$ scaling approach and its limitations.

The uncertainties of the O4 scaling approach need to be discussed. For example, O4 usually has a profile shape very different from NO2 and SO2, which violates the basic assumption that the O4 dSCD is a good proxy for the light path through the NO2 and SO2 layers. Other studies use correction factors from radiative transfer calculations to account for this (Sinreich et al., AMT, 2013). Furthermore, the resulting near-surface volume mixing ratios will not be representative for the amount of trace gases directly at the surface, but for some kind of average over a certain height range in the boundary layer. There is also "light dilution", i.e. light scattered into the line of sight between the instrument and the trace gas plume (see e.g. Kern et al., Bulletin of Volcanology, 2010), which further reduces the measured SCDs . My impression is that these are the main reasons for the discrepancies between in situ and MAX-DOAS, and not horizontal inhomogeneities as speculated later in the paper by the authors (these would cancel out when averaging the data). The discussion of the data needs to be revised accordingly in order to account for the influence of these aspects.

We have completely rewritten this section discussing in detail the limitations of the $O_4$ scaling approach according to Sinreich et al. (2013)[1] and Wang et al. (2014)[2] as well as explaining the reasons why the suggested correction factors have not been applied to the data in this study: The height of the $NO_2$ and $SO_2$ layer is unknown and no additional measurements of the layer height exist. Furthermore, a comparison to our in-situ measurements indicates that the layer height and therefore the correction factors vary from day to day as well as over the course of individual days. In addition to that, an extensive RTM study like it was performed by Sinreich et al. (2013) and Wang et al. (2014) was out of the scope of this publication and the comparision to our in-situ instrument not the main point of the paper.

Equation 5: It should be mentioned that nO4 is the O4 concentration at the surface.

We added this information.

The remarks regarding the elevation angle from Section 3.2 are repeated at the end of this section. See my comments above. A deviation of 0.5_ in elevation angle is certainly not negligible at very low elevation angles.
* * *
[1] Sinreich, R., Merten, A., Molina, L., & Volkamer, R. (2013). Parameterizing radiative transfer to convert MAX-DOAS dSCDs into near-surface box-averaged mixing ratios. *Atmospheric Measurement Techniques*, *6*(6), 1521–1532. https://doi.org/10.5194/amt-6-1521-2013

[2] Wang, Y., Li, A., Xie, P. H., Wagner, T., Chen, H., Liu, W. Q., & Liu, J. G. (2014). A rapid method to derive horizontal distributions of trace gases and aerosols near the surface using multi-axis differential optical absorption spectroscopy. *Atmospheric Measurement Techniques*, *7*(6), 1663–1680. https://doi.org/10.5194/amt-7-1663-2014

We changed the elevation angles to the actual values (center of field-of-view) throughout the paper and removed the remarks. We also added a discussion on the limitations of the O4-scaling due to non-consideration of correction factors (see also above).

L474: It is not obvious to me why a thicker trace gas layer should lead to a reduction of the ratio between dSCDs near the horizon and in zenith. Wouldn't horizontal inhomogeneities, with more NO2 over the shipping lane than over the instrument, be a much more likely explanation for these findings?

This Paragraph is not about differences between SCDs near the horizon and in zenith sky direction, but about systematically higher $NO_2$ and $SO_2$ DSCDs in 0.5° Elevation compared to the 2.5° Elevation for wind from the shipping lane, indicating a low pollution layer over the shipping lane. For southerly winds, on average the $NO_2$ and $SO_2$ DSCDs in 0.5° and 2.5° Elevation are nearly equal. We have rephrased the paragraph to make it clearer:

"The highest $NO_2$ and $SO_2$ DSCD in the lowest elevation angle (0.5°, blue bars) in relation to DSCDs in higher elevations are measured especially for wind from all northern directions, in a sector ranging from WSW to ESE. These directions coincide with the course of the main shipping lane, which comes from the WSW direction (the English Channel, the Netherlands, East Frisian Islands), passes the island in the north and runs close to the city of Cuxhaven (ESE direction) into the river Elbe. This indicates that these enhanced columns in the 0.5° elevation angle is pollution emitted from ships in a surface-near layer.

For southerly wind directions, where no larger shipping lane is in the direct surrounding and land-based pollution sources dominate, the average DSCDs in 0.5° and 2.5° elevation are nearly the same for both $NO_2$ and $SO_2$."

Section 4.2:
The title of this section is too long and complicated. I suggest to replace it by something like "Volume mixing ratios of NO2 and SO2"

Done.

Section 4.5:
As already stated above, the fact that MAX-DOAS averages over a large horizontal distance should cancel out on temporal average when comparing to in situ measurements. Instead, a more probable explanation for the systematically lower mixing ratios is the fact that MAX-DOAS averages over a certain altitude range and that the differences in O4 and target gas profile shapes has not been considered. Light dilution will also play a certain role. The argument that MAX-DOAS yields lower values when the plume is orthogonal to the viewing direction does not seem convincing to me, because in this case the polluted air is also not transported towards the in situ instrument, which means that the in situ instrument might even miss particular plumes which are detected by MAX-DOAS.

This section has been completely rewritten discussing again the systematic deviations produced by the non-consideration of correction factors for the different profile shapes. Also a remark about light dilution as an uncertainty source reducing the actual measured SCDs was incorporated.

Section 4.6:
L655ff: NO2 concentrations at a particular location strongly depend on local sources, such as traffic, industry, domestic heating, etc., as well as on the distance to these sources and on the rate of vertical mixing. Therefore, the fact that amount of NO2 in background air observed in the Arctic is similar to the present study might be mere coincidence.

We have removed this paragraph.

Section 4.7:
L667: Detection limits are mentioned here for the first time. They should instead be discussed in Section 3.3.

We added a paragraph about detection limits in Section 3.3.

"By assuming that an optical density of twice the RMS can be detected (Peters et al., 2013), it is possible to estimate the detection limit of our instrument regarding the different trace gases. The differential absorption cross section of $NO_2$ is in the order of $1 \times 10^{-19}$ cm²/molec, for $SO_2$ in the order of $2 \times 10^{-19}$ cm²/molec. Combining this yields a $NO_2$ detection limit of around $1 \times 10^{15}$ molec/cm² corresponding to 0.05 pbb in the visible and $2 \times 10^{15}$ molec/cm² corresponding to 0.1 pbb in the UV. The $SO_2$ detection limit is around $2.5 \times 10^{16}$ molec/cm² corresponding to 0.2 pbb."

Section 4.8:
An attempt is made here to separate shipping emissions from other sources by classifying the data according to the wind direction. The limitations of this approach need to be discussed more carefully. While I agree that northerly winds are little affected by background pollution, I strongly doubt that shipping emissions do not influence the measurements significantly when the wind is coming from the south. Data is filtered for light paths longer than 5 km, which means that for most observations the light path crosses the main shipping lane and probes air polluted by ship traffic. You reach this conclusion yourself in Section 4.1 (L438ff) in the context of the discussion of Figure 6, which shows that peaks from ship emissions clearly occur when air polluted by landbased sources is present. Thus, air masses classified as "Land" are likely to be partly affected by ship emissions.

You are right, air masses classified as "land" usually contain small fractions of shipping emissions as well. So the land source is slightly overestimated in these cases. How prominent ship emission peaks are in our measurements and how strong the contribution is compared to land based emissions depends on the wind direction. The wind sector classification is sketched in the following figure:

[Figure]

Blue sector: wind from open North Sea, shipping is the only pollution source

Green sector: mainly land-based air pollution (traffic, industry, …)

Yellow sector: air mass contains shipping emissions as well as land-based air pollution (mixed origin)

On 23 July 2014, the day in Figure 6 which you are referring to, in the morning the wind was coming from NE-ENE, and later turned towards NNE. Those wind directions are not included in the sector we chose for pollution coming from land. Wind from those directions clearly can contain large fractions of shipping emissions as well as pollution from land and are therefore classified as directions with "mixed" pollution origin in the study.
When the wind is coming from southerly directions, ship emission peaks are much less prominent in our measured time series of $NO_2$ or $SO_2$. A day which illustrates this nicely is 17 July 2014, shown in the plot below. Until noon, wind was coming from the south and later changed to northerly directions. Measurements were done in 0° elevation towards north. In the morning, although a lot of ships are present, as can be seen from the colored bars in Panel B, ship emission peaks are very small and hardly visible. A few are still visible, marked by the green arrows. As expected, the peaks are higher in the visible than in the UV, because wind is blowing the ship plumes away from the radar tower and our instrument. In the afternoon, ship emission peaks are higher and much more prominent. The contribution of shipping emissions to the overall $NO_2$ measured is certainly much higher in the afternoon than in the morning.

[Figure]

Another example is shown in the next plot for 6 August 2014.

[Figure]

To quantify the overestimation of the land source sector, the fraction of shipping emissions on the overall emissions on such a day has to be compared to the fraction of land sources. This is shown in the next plot: It is another example day with wind from southerly directions, 6 August 2014. Blue and red line show the NO$_2$ DSCD measured in the visible and UV. The cyan and magenta lines show the signal with removed ship emission peaks.

[Figure]

In the UV, the difference in the integral between "with ships" and "without ships" is around 1.4%, in the visible it is around 1.6%. So on this day, the NO2 classified as "land source" is overestimated by around 1.5% due to shipping emissions which are still contained in the data set. In other measurement directions, to the NE or NW for example, this overestimation is be a bit higher, but never exceeds 3%.

So this overestimation is a small error and was therefore neglected in the study.

Section 4.9:
Given that SO2 scatters around a smooth (near-zero) background level, it is surprising to see that no negative SO2 to NO2 ratios were derived. It seems that negative values have been set to zero (Panel D of Fig. 20), which would significantly (and falsely) affect the statistics.

To address this shortcoming in our study we have completely redone the SO$_2$ to NO$_2$ ratio peak analysis. The baseline determination has improved substantially (using a second running median filter applied to the lower 50% quartile when necessary) and the positive bias in the measurements since 2015 is now gone. In addition to that, the section has been rewritten taking into account your comments. Also, the importance of NO to NO$_2$ titration especially for the comparison to other studies is now mentioned in this section.

Updated plots:

For an example day (23.07.2014) before the change in regulations:

[Figure]

For an example day (03.07.2015) after the change in regulations:

[Figure]

And the updated histogram:

[Figure]

**Technical corrections**

Equations: Please use single characters for variables (e.g., "S" instead of "SCD", "R" instead of "RESIDUAL", "H" instead of "MLH"). There is a difference between an abbreviation (e.g., "SCD" for slant column density) and the according mathematical symbol (e.g., "S").

This is a matter of personal taste. In the DOAS community, using variable names like "SCD" or "AMF" is quite common. Checking the most recent final revised papers in ACP and AMT containing "DOAS" in the title we found 20 papers using multi-letter variable names like "SCD", "AMF", "AOD", 1 paper using single letter variable names "S" and "M" etc. (and 8 papers without any such equations).

L11: Provide a number for the distance between measurement site and shipping lane

We added the distance:

"The island is located in the German Bight, close to the main shipping lane *(in a distance of 6-7 km)* into the river Elbe towards the harbor of Hamburg."

L13: The fact that the site is close to the shipping lane is repeated. Delete "which is a few kilometres from the shipping lane"

Done.

L17: "retrieved from NO2 retrievals" -> "determined from NO2 retrievals"

Done.

L54: ". . . from around 31 000 . . .. over 52 000 . . . to 89 000 . . ."

Done.

L68: ". . . molecular nitrogen (N2) and oxygen (O2) . . ."

Done.

L81: comma after "radicals"

This paragraph was removed.

L81: ". . . hydroperoxyl (HO2) or organic peroxy radicals (RO2) or halogen oxides (XO, were X = Cl, Br or I)

This paragraph was removed.

L84: "X atoms" -> "halogen atoms"

This paragraph was removed.

L85: "reacts" -> "react"; "reaction" -> "reaction rate"

This paragraph was removed.

L86: "Owing to the lack of photolysis, NO reacts rapidly . . .. during the night"

Done.

L87: "In addition, the nitrate radical (NO3) is formed. . ."

Done.

L110: Comma after "regions"

Done.

L112: "ecosystem" -> "ecosystems"

Done.

L121: Put "3" in "m3" into superscript

Done.

L162: Comma after "emissions"

Done.

L173: Comma after "example"

Done.

L206: Incomplete sentence. Replace, e.g., by ". . . first the measurement site is described, followed by a presentation of the wind statistics and data availability."

Done.

L215: ". . . were taken on Neuwerk, a small island in the North Sea with the size of . . ."

Done.

L221: "island of Neuwerk" or simply "Neuwerk" (here and anywhere else). Delete "where our measurement site is located" (repetition)

Done.

L225: Do you refer to a specific document from the "Statistische Ämter" or can you

provide an url to the data?

We are referring to a specific document/URL from the "Statistische Ämter…" from the year 2015. The URL to the document can be found in the bibliography under "Statistische Ämter… (2015)". The correct way how to cite this information is specified on the web page from the Statistische Ämter.

L224: Is this height above sea level?

No, this is height above ground. But the difference to height above sea level, as you can see from the photo, might be 1-2 meters and therefore negligible.

L248: "site for the measurements" -> "site"

Done.

L285: To "inject" light into the fibre sounds strange since this term suggests that the light is somehow transported actively. Replace by something like "focused on the entrance of the optical fiber"

Done.

L286: "opening angle" -> "field of view"

Done.

L307: Define what "SCD1" and "SCD2" refer to. Replace by variable names consisting of single letters.

We rephrased the sentence to make this more clear.

Regarding the single letter variable names: This is a matter of personal taste. In the DOAS community, using variable names like "SCD" or "AMF" is quite common. Checking the most recent final revised papers in ACP and AMT containing "DOAS" in the title we found 20 papers using multi-letter variable names like "SCD", "AMF", "AOD", 1 paper using single letter variable names "S" and "M" etc. (and 8 papers without any equations).

L316: It should be mentioned that a spatially limited plume directly over the instrument leads to an underestimation of the retrieved dSCDs.

Done.

Table 1: Only list the polynomial degree, not the number of coefficients.

I listed both since the definiton of the polynomial degree can be ambiguos, according to whether you count the lowest order linear term as index 0 or 1.

L399: "filtered" -> "discarded"

Done.

L439: Delete "the pollutant"

Done.

L441: ". . . difference between NO2 in the UV (red curve) and in the visible (blue curve). . ." (the discussion is about NO2 and not about the colors of the curves)

Done.

L442: "more intense" -> "stronger"

Done.

L445: "By comparing SO2 (black curve) with NO2 (red and blue curves), it can be seen. . ."

Done.

L447: Delete "A more dirty"

Done.

L454: "points in time" -> "times"

Done.
L497: "The difference between UV and visible peak values depends. . ."

Done.

L498: "A short distance of the plume to the instrument and its complete coverage by the shorter UV path leads to higher values in the UV. . ."

Done.

The title of section 4.3 does not make sense. It implies that the approach is statistically evaluated. Instead, the data is statistically evaluated. Replace with something like "Statistical evaluation of the NO2 and SO2 data"

This section is not about $NO_2$ and $SO_2$ but about $NO_2$ in UV and visible for DSCDs and VMRs. We changed the title to "Statistical evaluation of UV and visible $NO_2$ data"

L507: ". . .all single pairs of simultaneous measurements" -> "all single pairs of DSCD measurements. . ."

Done.

L508: "the left subplot in the upper row" -> "Panel A"

Done.

L509: "both measurements" -> "NO2 and SO2 DSCDs"

This section is not about $NO_2$ and $SO_2$ but about $NO_2$ in UV and visible. We changed „both measurements" to "$NO_2$ DSCDs in UV and visible".

L513: "The right subplot in the upper row" -> "Panel B"

Done.

L518: "the left subplot in the bottom row" -> "Panel C"

Done.

L523: "(right plot)" -> "(Panel D of Fig. 9)"

Done.

L533: "applied on mountains" -> "applied to mountain-based measurements"

Done.

L535: Delete "However"

Changed "However" to "In contrast to our site".

L537: "This should lead" -> "This leads" (the enhancement in path length in a cleaner and less dense atmosphere is obvious)

Done.

L541: "various" -> "detailed" or "comprehensive"

Done.

L543: Delete "emitting"

Done.

Figure 10: Mark the three panels as "A", "B" and "C" (from top to bottom)

Done.

L544: "Measurements from Wednesday, 9 July 2014 are shown in Figure 10. Panel A shows. . ."

Done.

L545: "The middle one" -> "Panel B"

Done.

L549: "The lower sub-plot" -> "Panel C"

Done.

L563: "The differences of both measurement techniques need to be considered for such a comparison:"

Done.

L565: "at one point" -> "at a single location"

Done.

L568: Insert "the" before "line-of-sight"

Done.

L569: delete "line-of-sight" (it is already mentioned at the beginning of the sentence)

Done.

L579: Delete "From the Figure, it can be easily identified that"

Done.

L581: Delete "nicely"

Done.

L586: Delete "It is also clearly visible, that"

Done.

L591: "it's" -> "its"

Done.

L592: Delete the first sentence of this paragraph

Done.

L594: "the upper subplots" -> "Panel A"; Add ", respectively" to the end of the sentence.

Done.

L606: "makes no sense" -> "is of little use"

Done.

L603: "the lower subplot" -> "Panel B"

Done.

L613: Delete "As can be seen in the figures"; delete "usually"

Done. Done.

L614: What do you mean with "progression of both curves"?

We mean "curve shape" or "course of the curves". We changed the formulation to the latter.

L623: Insert comma after "combustion"

Done.

L643: "The mean NO2 volume mixing ratios for each weekday shown in Fig. 16 illustrate the influence of land-based road traffic."

Done.

L647: "There is only little weekly cycle for air masses coming from the open North Sea. Measurements . . ."

Done.

L665: "single day measurements" -> "Single day of measurements"

Done.

L704: Delete comma after "This implies"

Done.

L738: "like expected" -> "as expected"

Done.

L743: "It can be seen that this increase for the land source sector is only a relative increase by comparing. . ."

Done.

L765: "roll" -> "role"

Done.

L767: "A monitoring of emissions from single ships requires the analysis of individual plume peaks in the NO2 and SO2 data sets."

Done.

L780: I am not familiar with the term "emission factor". Do you mean "emission rate"?

Both terms mean more or less the same, in the sense of an emission intensity. However, in the community of shipping emission measurements, the term "emission factor" is more commonly used.

780: delete "both"

Done.

L796: "one can get rid of the background pollution" -> "the background pollution can be removed"

Done.

L801: "To achieve a better signal-to-noise ratio, the integrals . . .. in the last step"

Done.

L803: "one" -> "an"

Done.

L804: "In both the NO2 and SO2 signal" -> "Both the NO2 and SO2 signal show"

Done.

L805: delete "are visible"; delete "measured"; "The shape of the peaks is also often quite similar" -> "Most of the peaks are of similar shape"

Done. Done. Done.

L807: "The SO2 to NO2 ratio can vary strongly for different ships. For example, the plume of the ship passing the line of sight around 12:00 UTC has a high NO2 content, but is low in SO2, whereas the opposite is true for the ship passing at 12:30 UTC, indicating that the second ship. . ."

Done.

L811: Delete "In contrast to this,"

Done.

L813: "High NO2 peaks also occur on this day. However,. . ."

Done.

L818: "From this plot one can also see that" -> "As can be seen from this plot, "

Done.

L819: "overestimate" -> "overestimates"

Done.

L826: "retrieved" -> "analyzed"

Done.

L836: Insert comma before "indicating"

Done.

L839: "and for 2015 and 2016, one gets a mean value of . . . " -> ", and a mean value of . . . for 2015 and 2016"

Done.

L842: "leading to overestimation" -> "leads to an overestimation"

Done.

L850: "from" -> "by"

Done.

L851: "SO2 and NO2 emission ratios can also be derived from. . ."

Done.

L858: "the dependency of SO2 to NO2 ratio to fuels sulfur content"

Done.

L863: "Island Neuwerk" -> "Island of Neuwerk"

Done.

L865: "into" - > "and"

Done.

L871: Delete "also"

Done.

L882: "NO2" -> "daily averaged NO2"

This sentence is not about daily averages but about the weekly cycle (averages according to weekday) and diurnal cycle (averaged values according to the hour of the day) of $NO_2$.

L908: Insert "can" after "ratios"

Done.

Thanks again for your thorough proof-reading. This helped us a lot.

Atmos. Chem. Phys. Discuss.,
doi:10.5194/acp-2016-1153-RC2, 2017
General comment: This paper describes a 3 year series of multi axis DOAS measurements carried out from the German island Neuwerk, just south of the entry to the river Elbe. This is in the main ship channel of the port of Hamburg and the main aim of the measurements was to study these by observing UV and visible light horizontally towards the ship channel. The paper is well written, with good language and instructive graphs. The paper is a nice combination of measurements methodology and results paper. It shows the methodology to measure mixing ratios in a coastal places, together with ship plume measurements and some results about the effect of new IMO legislation. However, the OBJECTIVE and AIM should be declared more clearly in the text. The paper is also rather long, and I would recommend to shorten it, by removing sections which are outside the main scope of the paper. Forinstance merging and shortening sect 4.5 and 4.6 corresponding to mixing ratio measurements and comparisons. All in all, I believe the paper should be published, with some minor improvements, based on answering my specific comments below:

First, we would like to thank Anonymous Referee #2 for his/her helpful comments. Below, we reply point-by-point to the specific comments . As far as possible, we have considered the suggestions in the revised manuscript.
We tried to shorten the manuscript and omitted unnecessarily repeated information. Section 4.5 was shortened and two plots have been deleted and the remaining ones merged to a common figure. Also 40% of the pieplots in Figure 18 have been removed. At several places paragraphs have been rewritten to make the text more precise and shorter.

**Specific comments:**

Row 71, p 2: It is claimed that 25% of the NOx emerges as NO2 from the stack, but usually 10% is assumed from fluegas stacks; please give more details: I assume you also assume some titration?

The relevant text passage (Row 71) reads: "The emitted $NO_x$ comprises mainly NO, with less than 25% of $NO_x$ being emitted as $NO_2$ (Alföldy et al., 2013)." We are referring here to results from a study of Alföldy et al. from the year 2013[1]. In this study, the chemical composition of the plumes of 497 seagoing ships was measured in the port of Rotterdam in September 2008 and a statistical evaluation of emission factors was provided.  For the scope of our study, especially the results shown in Figure 17 are interesting:

[Figure]

Figure from: Alföldy et al. (2013)

**Fig. 17.** Distribution of the $NO_2/NO_x$ molar ratio among the studied ships. The total molar ratio range was divided into 19 bins. Frequen-

[1] Alföldy, B., Lööv, J. B., Lagler, F., Mellqvist, J., Berg, N., Beecken, J., … Hjorth, J. (2013). Measurements of air pollution emission factors for marine transportation in SECA. *Atmospheric Measurement Techniques*, *6*(7), 1777–1791. http://doi.org/10.5194/amt-6-1777-2013

The conclusion of the authors:

"The molar NO2-to-NOx emission ratio, calculated from the mixing ratios of the two components in the plume (%, n/N), is presented in Fig. 17. As can be seen, nitrogen oxides are mostly emitted as NO, the ratio of $NO_2$ emission is less than 25% at the majority of the ships."

In Row 71 we are simply referring to this result as background knowledge on the NO/$NO_2$ ratio in ship plumes, being important for our own study. We are not doing any assumptions here.

Row 278, p 9: IN the equation do you fit differential absorption cross sections or the absolute ones? Since you are using prime I assume you mean the differential ones; IN row 336 I however get the impression that you use the absolute ones.

We are fitting the differential absorption cross sections together with a low order polynomial to the measured optical depth. We have changed the sentence to make this point more clear:

"Multiple *(differential)* trace gas absorption cross sections obtained from laboratory measurements, as well as a low-order polynomial, are then fitted simultaneously to the optical depth."

Row 311, p 10: It is claimed that the vertical paths cancels out between path 1 and 2 in Fig 5; I agree with the stratospheric portion but for the tropospheric part there should be a cos (SZA) difference, even if NO2 is homogenously distributed in the troposphere?

That is a good objection. It is in fact true that this is only an assumption. To make things more clear we show a more detailed sketch below. The presented approach of using the $O_4$ column to estimate the effective horizontal light path length assumes single-scattering geometry. For the vertical paths in a layer of homogenously distributed $NO_2$ in the troposphere, like it is shown in the first sketch, to cancel out, the reference measurement must have the "assumed path". This means, it is assumed that the scattering point for the zenith reference is at the altitude of the instrument. In reality, of course, this is not the case. The real scattering altitude for light measured in zenith direction will be in an effective scattering height $h$, as it is shown in the sketch.

[Figure]

As can be seen from the figure, this leads to an underestimation of $NO_2$ in the reference and therefore to an overestimation of the $NO_2$ concentration. However, not only the $NO_2$ is overestimated, but also the $O_4$ path length is overestimated in a similar way.

Gomez et al. (2014)[2], applying this approach to MAX-DOAS measurements from a high mountain site, did a thorough error analysis in Section 3.2 of their publication. They showed that first, this scattering height $h$ is nearly constant up to an SZA of 75°. Secondly, the path error depends only on the vertical distribution of the $NO_2$ (or $SO_2$) and on the differences in air mass factors (AMF) of $NO_2$ and $O_4$. By assuming a homogeneous layer, like it is shown in the sketch, the error comes from differences in the

[2] Gomez, L., Navarro-Comas, M., Puentedura, O., Gonzalez, Y., Cuevas, E., & Gil-Ojeda, M. (2014). Long-path averaged mixing ratios of O3 and NO2 in the free troposphere from mountain MAX-DOAS. *Atmospheric Measurement Techniques*, *7*(10), 3373–3386. http://doi.org/10.5194/amt-7-3373-2014

AMF. The SZA dependence of the error of the approach has been plotted by the authors in the following figure:

[Figure]

Figure from:
Gomez et al.
(2014)

As can be seen from the figure, the error of the approach is less than 10 percent for typical daytime SZAs.

To keep the approach simple, this amount of uncertainty has to be accepted.

**Figure 5.** Estimated error of the MGA versus the SZA.

More important for this study is of course the measurement of ship emission plumes. When the wind is coming from the open North Sea, there is negligible background $NO_2$ and $SO_2$ in the lower troposphere. When a ship plume is in the horizontal path of the off-axis measurement, like it is sketched below, the difference between assumed and real reference path is irrelevant, introducing no additional error.

[Figure]

Row 387, p 13: Is it assumed that the wavelength difference in O4 signal is linear; if so what are the uncertainties involved?

We changed our method from a simple linear extrapolated scaling factor to using the empirically determined (from RTM simulations) formula from Wang et al. (2014)[3] to improve this source of uncertainty:

$$L_{310} = 0.136 + 0.897 \times L_{360} - 0.023 \times L_{360}^2. \qquad (6)$$

Which was determined from RTM simulations for a variety of aerosol conditions, which results are shown in the following figure:

[Figure]

Figure from:
Wang et al.
(2014)

**Figure 1.** Scatter plot of light path lengths at 310 nm against light path lengths at 360 nm for 60 aerosol scenarios and combinations of three SZAs and RAAs. The statistical parameters derived from a second-order polynomial fitted to the simulation results are also shown.

Row 406, p 14: It is claimed that the conditions at the Neuwerk radar tower is similar to measurements from high mountains; please motivate better.

This section was completely rewritten to make our motivation clearer:

"This approach has been applied successfully by Sinreich et al. (2013) and Wang et al. (2014b) for measurements in urban polluted air masses over Mexico City and the city of Hefei (China) using MAXDOAS measurements in 1° and 3° (Sinreich et al., 2013) and only in 1° elevation (Wang et al., 2014b), respectively. Gomez et al. (2014) applied this approach to measurements on a high mountain site at the Izana Atmospheric observatory on Tenerife (Canary Islands), Schreier et al. (2016) at Zugspitze (Germany) and Pico Espejo (Venezuela). Due to the low aerosol amounts in such heights the latter two studies applied the approach without using correction factors. The fact that our instrument is located on a radar tower in a height of about 30m above totally at surroundings (the German Wadden Sea) allows an unblocked view to the horizon in all feasible azimuthal viewing directions. This led to theidea of trying to apply this approach to our shipping emission measurements on Neuwerk."

[3] Wang, Y., Li, A., Xie, P. H., Wagner, T., Chen, H., Liu, W. Q., & Liu, J. G. (2014). A rapid method to derive horizontal distributions of trace gases and aerosols near the surface using multi-axis differential optical absorption spectroscopy. *Atmospheric Measurement Techniques*, *7*(6), 1663–1680. https://doi.org/10.5194/amt-7-1663-2014

Eq 4, p 13: It is difficult to follow how you get the expression in eq 4.

We deleted this equation and focus now on the approach which was actually applied to the data (the O4-scaling method).

Row 464 p 16: On this place, and some others, its is claimed that the differential slant columns are higher for SSE and ESE and (more elevated). But part of this should be wind speed effect since I would imagine that the wind speed will be higher from the sea and this will dilute the slant columns more. Has this been investigated ?

We have looked into this: The following polar plot shows the mean wind speed depending on wind direction. Wind speeds from the land sector (light green) are not substantially lower than wind speeds from the open Sea sector (light blue), so such a dependence has not been observed.

[Figure]

Row 470 p 16: Graph 7 is not totally clear. If I understand right the plot correspond to overlayed windroses for different elevation angles rather than that the area of each color represents the wind rose information. I interpreted the latter and I think this should be clarified forinstance in the figure text.

The former is correct. We changed the figure caption to make it clearer:

[Figure]

Figure 7: Overlayed wind roses for different elevation angles showing the wind direction distribution of the UV $NO_2$ (A) and $SO_2$ (B) differential slant column densities measured in the main viewing direction in 0.5°, 2.5°, 4.5° and 30.5° elevation in the years 2013 and 2014. The wind roses are plotted on top of each other, i.e. the highest values were measured in the lowest elevation angle (blue bars). The colored sectors show directions with wind from land (green), open North Sea (blue) and mixed origin (yellow).

Row 500 p 17: You here discuss the results in Fig 8. The differences in the UV and visible are explained from the penetration length, but should the Visible not in general be higher since it gives the chance of penetrating plumes further away, rather than the opposite which appears to be the case for all plumes here? You explain that the UV should be stronger for close by plumes since a higher fraction of the photons are then affected by absorption. Is is not so that the O4 can only simulate slow variations? Please elaborate..

The relevant point here is the position of the ships relative to the measurement site. On average, the measured NO2 slant column densities are higher in the visible than in the UV due to the longer horizontal light path. However, the data shown in Figure 8 are path-averaged mean volume mixing ratios. Typical path lengths are 10 km in the UV and 15 km in the visible. If the measured ship exhaust plume is closer to the instrument than 10 km, which is usually the case for all northerly wind directions since the ships pass the instrument in a distance of 6 to 7 km, the path averaged volume mixing ratio on the visible path will be lower due to the longer averaging distance. We have included two sketches below to make things clearer:

[Figure]

[Figure]

Map showing the azimuthal viewing directions and the typical averaging path lengths. The green and red dotted lines highlight the boundaries (line of buoys) of the main shipping lane.

Row 614 p 25: You claim that fig 12/fig 13 shows good agreement between MAX DOAS and in situ, but in my mind this is the case for Fig 13 but not for fig 12, where there appears to be rather big difference in he averages of the two sensors with factor 2-3?

In this Section we do not claim a good agreement in absolute values (which due to the characteristics of both measurement techniques and different measurement geometries is also not expected), but a good agreement in the shape (or course) of the curves. This means that apart from a scaling factor, the day-to-day trends in both time series of daily means are well reproduced.

Row 665 p 27: You suddenly refer to fig 20, without having mentioned fig 17-19 yet in the text. You should consider reordering.

This reference has been deleted.

Row 891 p 35: As concluded here and discussed in section 4.9, the ratio of SO2/NO2 gives an indication of sulfur fuel content in ship plumes. Are you aware that SO2/NO2 ratio measurements from airborne DOAS is used operationally since 2015 by Beecken and Mellqvist (Chalmers University) in the CompMon project and surveillance around Denmark and that this has been presented on several official workshops last year?

Thank you very much for that hint. The methodology presented in the CompMon Report "Best Practices Airborne MARPOL Annex VI Monitoring" (Van Roy, 2016)[4] is very interesting. We added the following paragraph to the chapter on $SO_2/NO_2$ ratios:

"By comparing $SO_2$ to NO2 ratios from different ships it is possible to roughly distinguish whether a ship is using fuel with high or low sulfur content (giving a high or low $SO_2$ to $NO_2$ ratio). *Beecken and Mellqvist from Chalmers University (Sweden) use this relationship for airborne DOAS measurements of ship exhaust plumes on an operational basis in the CompMon project (Compliance monitoring pilot for MARPOL Annex VI) (Van Roy, 2016). Following the ships and measuring across the stack gas plume they can discriminate between low (0.1 %) and high (1 %) fuel sulfur content ships with a probability of 80-90% (Van Roy, 2016).*"

You mention that the +2015 measurements are biased by noise since you don't really observe any SO2 then. I don't think it then makes sense to show the green data (+2015) in figure 5 since these histograms then only represent noise? Secondly you don't mention when comparing to other measurements that the amount of NO to NO2 titration is very important for the ratio, and this will depend on the distance to the plume, whether you are over land or sea etc. Please add some discussion on this.

To address this shortcoming in our study we have completely redone the $SO_2$ to $NO_2$ ratio peak analysis. The baseline determination has improved substantially (using a second running median filterapplied to the lower 50% quartile when necessary) and the positive bias in the measurements since 2015 is now gone. In addition to that, the section has been rewritten taking into account your comments. Also, the importance of NO to $NO_2$ titration especially for the comparison to other studies is now mentioned in this section.

Updated plots:
* * *
[4] Van Roy, W. (2016). Best Practices Airborne MARPOL Annex VI Monitoring. Retrieved from http://ec.europa.eu/transparency/regexpert/index.cfm?do=groupDetail.groupDetailDoc&id=29311&no=7

For an example day (23.07.2014) before the change in regulations:

[Figure]

For an example day (03.07.2015) after the change in regulations:

[Figure]

And the updated histogram:

[Figure]

Row 903 p 35: It is mentioned that there are still SO2 coming from land. This is surprising since there are very few SO2 emission sources anymore and power plants generally have abatement equipment. It would be interesting to understand this better ?

SO$_2$ concentrations in Germany decreased significantly in the last decade (-93% since 1990) due to advanced filter techniques and are now stable on a low level [5]. Still, the most important source is energy production, followed by industry. In Bremen, typical annual mean values are 1 to 2 µg/m$^3$, with short-time peaks (maximum 1-hour-means) of 20 to 80 µg/m$^3$, with the highest values close to industrial sites [6]. The German Federal Environmental Agency (Umweltbundesamt) operates a network of several in-situ air quality measurement stations throughout Germany [7]. The following two plots show SO$_2$ daily mean concentrations for the last 18 months for five rural stations and five urban stations in Northern Germany. The overall mean value for each station is given in the legend.

[5] https://www.umweltbundesamt.de/daten/luftbelastung/luftschadstoff-emissionen-in-deutschland/schwefeldioxid-emissionen (16.05.2017)
[6] Der Senator für Umwelt, Bau und Verkehr
Contrescarpe 72, *Das Bremer Luftüberwachungssystem - Jahresbericht 2015*
http://www.bauumwelt.bremen.de/sixcms/media.php/13/BdV_L_2016-08_Jahresbericht_Luftmessnetz_2015_Anhang.pdf (16.05.2017)
[7] https://www.umweltbundesamt.de/daten/luftbelastung/aktuelle-luftdaten (16.05.2017)

[Figure]

The typical average SO₂ concentrations measured by the German Federal Environmental Agency ("Umweltbundesamt") for rural stations are around 0.5 to1 µg/m³, corresponding to 0.2 – 0.4 ppbv (Conversion factor: 1 ppb = 2.62 µg/m³ for SO₂). Measurements in cities and especially close to industrial areas show higher values. Bremerhaven, which is the station closest to our instrument, has a mean concentration of 1.77 µg/m³, corresponding to 0.67 ppbv.

We measured mean SO₂ mixing ratios from land between 0.3 and 0.4 ppbv since January 2015 (see Figure 17), which in our opinion fits very well to those measurements.

Technical Corrections: Well written in most places.

Row 812 p 31: Change limis to limits

Done.

Row 873 p 33: Change This to These

Done.

[revised manuscript text omitted]
_2$ (top row) and $SO_2$ (middle and bottom row) levels on Neuwerk. For $NO_2$ the complete time series of measurements from 2013 to 2016 has been taken into account, for $SO_2$ the data have been divided into the time before and after the change in fuel sulfur content limits. The leftmost column of pie plots show the percentage of measurements with wind coming mainly from land (green), only from sea (blue) and from directions with mixed contributions (yellow). The middle column shows the contributions to the integrated, total volume mixing ratios from these source regions in percent. The rightmost column of pie plots shows analogous the percentage and mean VMR contribution by 
[revised manuscript text omitted]
". In: *Atmospheric Chemistry and Physics* 15.9, pp. 5229–5241. ISSN: 16807324. DOI: 10.5194/acp-15-5229-2015.

Beirle, S., Platt, U., Glasow, R. von, Wenig, M., and Wagner, T. (2004). "Estimate of nitrogen oxide emissions from shipping by satellite remote sensing". In: *Geophysical Research Letters* 31.18, pp. 4–7. ISSN: 00948276. DOI: 10.1029/2004GL020312.

Beirle, S., Platt, U., Wenig, M., and Wagner, T. (2003). "Weekly cycle of NO2 by GOME measurements: a signature of anthropogenic sources". In: *Atmospheric Chemistry and Physics* 3.2, pp. 2225–2232. ISSN: 16807324. DOI: 10.5194/acpd-3-3451-2003.

Bell, T. L., Rosenfeld, D., and Kim, K. M. (2009). "Weekly cycle of lightning: Evidence of storm invigoration by pollution". In: *Geophysical Research Letters* 36.23, pp. 1–5. ISSN: 00948276. DOI: 10.1029/2009GL040915.

Berg, N., Mellqvist, J., Jalkanen, J. P., and Balzani, J. (2012). "Ship emissions of SO 2 and NO 2: DOAS measurements from airborne platforms". In: *Atmospheric Measurement Techniques* 5.5, pp. 1085–1098. ISSN: 18671381. DOI: 10.5194/amt-5-1085-2012.

Bobrowski, N. and Platt, U. (2007). "SO2/BrO ratios studied in five volcanic plumes". In: *Journal of Volcanology and Geothermal Research* 166.3-4, pp. 147–160. ISSN: 03770273. DOI: 10.1016/j.jvolgeores.2007.07.003.

Bogumil, K., Orphal, J., Homann, T., Voigt, S., Spietz, P., Fleischmann, O. C., Vogel, A., Hartmann, M., Kromminga, H., Bovensmann, H., Frerick, J., and Burrows, J. P. (2003). "Measurements of molecular absorption spectra with the SCIAMACHY pre-flight model: instrument characterization and reference data for atmospheric remote-sensing in the 230-2380 nm region". In: *Journal of Photochemistry and Photobiology A: Chemistry* 157.2-3, pp. 167–184. ISSN: 10106030. DOI: 10.1016/S1010-6030(03)00062-5. URL: http://linkinghub.elsevier.com/retrieve/pii/S1010603003000625.

Bollmann, M., Bosch, T., Colijn, F., Ebinghaus, R., Körtzinger, A., Latif, M., Matthiessen, B., Melzner, F., Oschlies, A., Petersen, S., Proelß, A., Quaas, M., Requate, T., Reusch, T., Rosenstiel, P., Schrottke, K., Sichelschmidt, H., Siebert, U., Soltwedel, R., Sommer, U., Stattegger, K., Sterr, H., Sturm, R., Treude, T., Vafeidis, A., Bernem, C. van, Beusekom, J. van, Visbeck, M., Wahl, M.,

985    Wallmann, K., and Weinberger, F. (2010). "Living With the Oceans". In: *World Ocean Review: Living with the oceans* 1, p. 236.

Brasseur, G. P. (1999). *Atmospheric chemistry and global change: [a textbook prepared by scientists at the National Center for Atmospheric Research, Boulder]*. Topics in environmental chemistry. New York, NY [u.a.]: Oxford Univ. Press. ISBN: 0195105214.

990 Chen, G., Huey, L. G., Trainer, M., Nicks, D., Corbett, J., Ryerson, T., Parrish, D., Neuman, J. A., Nowak, J., Tanner, D., Holloway, J., Brock, C., Crawford, J., Olson, J. R., Sullivan, A., Weber, R., Schauffler, S., Donnelly, S., Atlas, E., Roberts, J., Flocke, F., Hübler, G., and Fehsenfeld, F. (2005). "An investigation of the chemistry of ship emission plumes during ITCT 2002". In: *Journal of Geophysical Research D: Atmospheres* 110.10, pp. 1–15. ISSN: 01480227. DOI: 10.1029/
995    2004JD005236.

Corbett, J. J. and Koehler, H. W. (2003). "Updated emissions from ocean shipping". In: *Journal of Geophysical Research* 108.D20, p. 4650. ISSN: 0148-0227. DOI: 10.1029/2003JD003751. URL: http://doi.wiley.com/10.1029/2003JD003751.

Corbett, J. J., Fischbeck, P. S., and Pandis, S. N. (1999). "Global nitrogen and sulfur inventories for
1000    oceangoing ships". In: *Journal of Geophysical Research* 104.D3, pp. 3457–3470. ISSN: 0148-0227. DOI: 10.1029/1998JD100040.

Corbett, J. J., Winebrake, J. J., Green, E. H., Kasibhatla, P., Eyring, V., and Lauer, A. (2007). "Mortality from ship emissions: A global assessment". In: *Environmental Science and Technology* 41.24, pp. 8512–8518. ISSN: 0013936X. DOI: 10.1021/es071686z.

1005 Diesch, J. M., Drewnick, F., Klimach, T., and Borrmann, S. (2013). "Investigation of gaseous and particulate emissions from various marine vessel types measured on the banks of the Elbe in Northern Germany". In: *Atmospheric Chemistry and Physics* 13.7, pp. 3603–3618. ISSN: 16807316. DOI: 10.5194/acp-13-3603-2013.

DNV (2008). "Marpol 73/78 Annex VI - Regulations of Air Polution from Ships - Technical and
1010    Operational implications". In: *Fuel*, pp. 1–32.

Endresen, Ø., Sørgard, E., Sundet, J., Dalsøren, S., Isaksen, I., and Berglen, T. (2003). "Emission from international sea transportation and environmental impact". In: *Journal of Geophysical Research* 108.D17, pp. 1–22. ISSN: 0148-0227. DOI: 10.1029/2002JD002898.

EU (2005). "Directive 2005/33/EC of the European Parliament and of the Council". In: *Official*
1015    *Journal of the European Union* 1882, pp. 59–69. DOI: 10.3000/17252555.L_2009.140.eng. URL: http://eur-lex.europa.eu/LexUriServ/LexUriServ.do?uri=OJ:L:2005:191:0059:0069: EN:PDF.

— (2008). "Directive 2008/50/EC of the European Parliament and of the Council of 21 May 2008 on ambient air quality and cleaner air for Europe". In: *Official Journal of the European Communities*
1020    152, pp. 1–43. URL: http://eur-lex.europa.eu/LexUriServ/LexUriServ.do?uri=OJ:L:2008: 152:0001:0044:EN:PDF.

— (2016). *Air Quality Standards*. URL: http://ec.europa.eu/environment/air/quality/ standards.htm (visited on 12/15/2016).

Eyring, V., Bovensmann, H., Cionni, I., Dall'Amico, M., Franke, K., Khlystova, I., Klinger, C., Lauer,
1025    A., Paxian, A., Righi, M., and Schreier, M. (2010a). *Impact of Ship Emissions on Atmosphere and Climate, SeaKLIM Final Report*. Tech. rep. DLR, p. 23. URL: http://www.pa.op.dlr.de/ SeaKLIM/SeaKLIM%7B%5C_%7DNachwuchsgruppe%7B%5C_%7DSchlussbericht%7B%5C_%7DFINAL. pdf.

Eyring, V., Köhler, H. W., Aardenne, J. van, and Lauer, A. (2005a). "Emissions from international
1030    shipping: 1. The last 50 years". In: *Journal of Geophysical Research* 110.D17, p. D17305. ISSN: 0148-0227. DOI: 10.1029/2004JD005619. URL: http://doi.wiley.com/10.1029/2004JD005619.

Eyring, V., Köhler, H. W., Lauer, A., and Lemper, B. (2005b). "Emissions from international shipping: 2. Impact of future technologies on scenarios until 2050". In: *Journal of Geophysical Research* 110.D17, p. D17306. ISSN: 0148-0227. DOI: 10.1029/2004JD005620. URL: http://doi.wiley.
1035    com/10.1029/2004JD005620.

Eyring, V., Isaksen, I. S., Berntsen, T., Collins, W. J., Corbett, J. J., Endresen, O., Grainger, R. G., Moldanova, J., Schlager, H., and Stevenson, D. S. (2010b). "Transport impacts on atmosphere and climate: Shipping". In: *Atmospheric Environment* 44.37, pp. 4735–4771. ISSN: 13522310. DOI: 10.1016/j.atmosenv.2009.04.059. URL: http://linkinghub.elsevier.com/retrieve/pii/S1352231009003379.

[revised manuscript text omitted]

UNCTAD (2014). *Review of Maritime Transport 2014*, p. 34. ISBN: 9789211128789. URL: http://unctad.org/en/PublicationsLibrary/rmt2014%7B%5C_%7Den.pdf.

— (2015). *Review of Maritime Transport 2015*. October. ISBN: 978-92-1-112892-5. URL: http://unctad.org/en/PublicationsLibrary/rmt2015%7B%5C_%7Den.pdf.

Van Roy, W. (2016). "Best Practices Airborne MARPOL Annex VI Monitoring". In: URL: https://www.trafi.fi/filebank/a/1482762219/d80cdd7de80e58885ce5a4dd0af86c02/23541-Best%7B%5C_%7DPractices%7B%5C_%7DAirborne%7B%5C_%7DMARPOL%7B%5C_%7DAnnex%7B%5C_%7DVI%7B%5C_%7DMonitoring.pdf.

Vandaele, A. C., Hermans, C., Simon, P. C., Roozendael, M. V., Guilmot, J. M., Carleer, M., and Colin, R. (1996). "Fourier Transform Measurement of NO2 Absorption Cross-Section in the Visible Range at Room Temperature". In: *J. Atmos. Chem.* 25, pp. 289–305.

Vinken, G. C. M., Boersma, K. F., Donkelaar, A. van, and Zhang, L. (2014). "Constraints on ship NOx emissions in Europe using GEOS-Chem and OMI satellite NO2 observations". In: *Atmospheric Chemistry and Physics* 14.3, pp. 1353–1369. ISSN: 1680-7324. DOI: 10.5194/acp-14-1353-2014. URL: http://www.atmos-chem-phys.net/14/1353/2014/acp-14-1353-2014.html.

Wagner, T., Dix, B., Friedeburg, C. V., Frieß, U., Sanghavi, S., Sinreich, R., and Platt, U. (2004). "MAX-DOAS O4 measurements: A new technique to derive information on atmospheric aerosols - Principles and information content". In: *Journal of Geophysical Research D: Atmospheres* 109.22, pp. 1–19. ISSN: 01480227. DOI: 10.1029/2004JD004904.

Wang, T., Hendrick, F., Wang, P., Tang, G., Clémer, K., Yu, H., Fayt, C., Hermans, C., Gielen, C., Müller, J. F., Pinardi, G., Theys, N., Brenot, H., and Van Roozendael, M. (2014a). "Evaluation of tropospheric SO2 retrieved from MAX-DOAS measurements in Xianghe, China". In: *Atmospheric Chemistry and Physics* 14.20, pp. 11149–11164. ISSN: 16807324. DOI: 10.5194/acp-14-11149-2014.

Wang, Y., Li, A., Xie, P. H., Wagner, T., Chen, H., Liu, W. Q., and Liu, J. G. (2014b). "A rapid method to derive horizontal distributions of trace gases and aerosols near the surface using multi-axis differential optical absorption spectroscopy". In: *Atmospheric Measurement Techniques* 7.6, pp. 1663–1680. ISSN: 18678548. DOI: 10.5194/amt-7-1663-2014.

Wayne, R. P. (2006). *Chemistry of atmospheres: an introduction to the chemistry of the atmospheres of earth, the planets, and their satellites*. 3. ed., re. Oxford [u.a.]: Oxford Univ. Press. ISBN: 019850375X and 9780198503750.

WHO (2006). *Air Quality Guidelines: Global Update 2005 : Particulate Matter, Ozone, Nitrogen Dioxide, and Sulfur Dioxide*. A EURO Publication. World Health Organization. ISBN: 9789289021920.

Wittrock, F., Oetjen, H., Richter, A., Fietkau, S., Medeke, T., Rozanov, A., and Burrows, J. P. (2004). "MAX-DOAS measurements of atmospheric trace gases in Ny-Ålesund - Radiative transfer studies and their application". In: *Atmos. Chem. Phys.* 4, pp. 955–966.

WSV (2013). *Verkehr und Güterumschlag*. URL: http://www.ast-nordwest.gdws.wsv.de/schifffahrt/verkehr%7B%5C_%7Dgueterumschlag/index.html (visited on 11/01/2016).

— (2014). *Schiffsverkehr und Güterumschlag*. URL: http://www.wsd-nord.wsv.de/Schiff-WaStr/Schifffahrt/Schiffsverkehr%7B%5C_%7Dund%7B%5C_%7DGueterumschlag/index.html (visited on 11/01/2016).

[revised manuscript text omitted]

very fast, leading to a dynamic equilibrium. This is also known as the Leighton photostationary state. ~~Deviations from the Leighton photostationary state occur in air masses, if the rates of the reactions of free radicals such as hydroperoxyl, $HO_2$, or organic peroxy radicals, $RO_2$, or Halogen oxides XO, where X=Cl, Br or I, compete with the reaction of NO with $O_3$. The $NO_2$ formed in the reactions of $HO_2$ or $RO_2$ with NO is photolyzed and the O atoms reacts in the termolecular reaction with oxygen molecules $O_2$ to form $O_3$. In tropospheric air-masses, typically, the X atoms released by the reaction of NO with XO typically reacts rapidly with $O_3$ to reform XO. This changes the reaction of $NO_2$ to NO but does not produce $O_3$. During night due to$NO_3$,~~ ($NO_3$) 
[revised manuscript text omitted]

~~**Remark concerning the elevation angles of our instrument:** The value in the following text is referred to as 0° elevation angle is in reality an elevation angle of 0.6°. The acceptance angle of our telescope is about 1.1° and it has a circular field of view. This means that the field of view extends vertically from 0.05° to 1.05° (nearly 0° to 1°). Thus the 0° line-of-sight represents an average over this field of view. This has the advantage that the surface, which may have spectral structures, is not explicitly probed. The same averaging over the relevant solid angle occurs for the higher elevation angles like 2, 4, 30 and 90°.~~

**3.3 DOAS data analysis and fit settings**

[revised manuscript text omitted]

**3.4  Retrieval of  path-averaged near-surface VMRs from MAX-DOAS SCDs**

To ~~compare DOAS measurements (trace gas columns) with, for example, in-situ measurements (concentrations), the retrieved slant column densities need to be converted to volume mixing ratios. The volume mixing ratio VMR $= n_x/n_{air}$ of a gas $x$ is defined as the ratio of the number densities of the gas and air and describes its atmospheric number fraction. The number density of air can be estimated using the ideal gas law:~~

$$n_{air} = \frac{N_{air}}{V_{air}} = \frac{p_{air} \cdot k_B}{T_{air}} = \frac{p_{air} \cdot N_A}{T_{air} \cdot R}$$

~~One possibility is to use a geometric approximation with a simple geometric air mass factor $AMF_{geom} = \frac{1}{\sin(\alpha)}$ for the elevation angle $\alpha$ to first convert the slant columns to vertical columns. The tropospheric vertical column density (VCD) divided by a typical mixing layer height (MLH) in which the trace gas is assumed to be well-mixed then gives the number density of the trace gas:~~

$$n_{x,geom} = \frac{VCD_{trop}}{MLH} \qquad \text{with} \qquad VCD_{trop} = \frac{SCD_\alpha - SCD_{90°}}{\sin(\alpha)^{-1} - \sin(90°)^{-1}} = \frac{DSCD_\alpha}{\sin(\alpha)^{-1} - 1}$$

~~A disadvantage of this method is the assumption of a typical mixing layer height, if no independent measurements of the MLH (e. g. using LIDAR) is available. Another disadvantage is that this approach does not account for changes in the light path due to changing weather (clouds, fog) and aerosol conditions. In addition, the profile will not be box-shaped in reality and the geometric AMF does only hold for large elevation angles.~~

 measure shipping emissions at our measurement site, our MAX-DOAS telescope is pointed towards the horizon, where the ships pass our site in a distance of 6–7 km. Since our instrument has a field-of-view of approximately 1°, the lowest usable elevation angle avoiding looking onto the ground is 0.5°, providing us with the highest sensitivity to near-surface pollutants. This is the elevation in which at our site usually the highest slant columns are measured. To convert a MAX-DOAS trace gas column which is the concentration of the absorber integrated along the effective light path into concentrations or volume mixing ratios, the length of this light path has to be known. This effective light path length depends on the atmospheric visibility, which is limited by scattering on air molecules as well as aerosols. As described in Section 3.2, trace gas absorptions in the higher atmosphere like stratospheric $NO_2$ nearly cancel out using a close-in-time zenith-sky reference spectrum. Following this, we can assume that the signal for  our horizontal line-of-sight is dominated by the horizontal part of the light path after the last scattering event.  As introduced by Sinreich et al. (2013), the length $L$ of this horizontal part of the light path can then be estimated using the slant column density of the $O_4$-molecule which has a well-known number density in the atmosphere:

$$L_{O_4} = \frac{SCD_{O_4,horiz} - SCD_{O_4,zenith}}{n_{O_4}} = \frac{DSCD_{O_4}}{n_{O_4}} \quad\quad (3)$$

 The surface number density of $O_4$ is proportional to the square of the molecular oxygen concentration (Greenblatt et al., 1990; Wagner et al., 2004) and can be easily calculated  from the temperature and pressure measured on the radar tower:

$$n_{O_4} = (n_{O_2})^2 = (0.20942 \cdot n_{air})^2 \quad \text{with} \quad n_{air} = \frac{N_{air}}{V_{air}} = \frac{p_{air} \cdot k_B}{T_{air}} = \frac{p_{air} \cdot N_A}{T_{air} \cdot R} \quad\quad (4)$$

[revised manuscript text omitted]

~~The $O_4$ scaling approach was previously applied to measurements from high mountain sites only, for example by Gomez et al. (2014) at the Izaña Atmospheric observatory on Tenerife (Canary Islands) or by Schreier et al. (2016) at Zugspitze (Germany) and Pico Espejo (Venezuela). The fact that our instrument is located on a radar tower in a height of about 30 m above totally flat surroundings (the German Wadden Sea) means that it is appropriate to apply this approach to our measurements on Neuwerk.~~

~~**Remark concerning the elevation angles of our instrument:** Since the opening angle or field of view of our instrument is about 1.1°, looking at 0° elevation towards the horizon would result in partially (with the lower half of our circular field of view) looking onto the ground (or sea surface, depending on tide). To avoid possible problems arising from this like spectral interferences, our instrument is looking slightly upward. What in this study is referred to as 0° elevation angle is in reality an elevation angle of around 0.6°. With our opening angle of about 1.1° and a circular field of view this means the field of view extends vertically from 0.05° to 1.05°, so 0° elevation means actually a field of view from nearly 0° to 1°. The same is true for the higher elevation angles like 2, 4, 30 and 90°. Since deviations arising from this are small, this is neglected in the following.~~

**3.5 In-situ instrumentation**

[revised manuscript text omitted]

 and  in-situ

 instrument should in principle measure

~~A) Long-term time-series comparison of NO₂ volume mixing ratios from in-situ and MAX-DOAS (UV) instruments during summer 2014. For the MAX-DOAS instrument, all measurements in all azimuth viewing directions are shown. B) Daily means of NO₂ VMR from MAX-DOAS (UV) and in-situ during summer 2014. For the MAX-DOAS instrument, all measurements in all azimuth viewing directions have been averaged. For the in-situ instrument, the mean of all measurements during the daily MAX-DOAS measurement periods (sunrise till sunset) have been taken.~~

~~A) Long-term time-series comparison of SO₂ volume mixing ratios from in-situ and MAX-DOAS instruments during summer 2013. For the MAX-DOAS instrument, all measurements in all azimuth viewing directions are shown. B) Daily means of SO₂ VMR from MAX-DOAS and in-situ during summer 2013. For the MAX-DOAS instrument, all measurements in all azimuth viewing directions have been averaged. For the in-situ instrument, the mean of all measurements during the daily MAX-DOAS measurement periods (sunrise till sunset) have been taken.~~

 same values. However, as discussed in Section 3.4, correction factors need to be applied to   MAX-DOAS VMRs to account for the different profile shapes of $O_4$ and the investigated pollutants $NO_2$ and $SO_2$, but in our case cannot be determined because no measurements of the height of the $NO_2$ and $SO_2$ layer exist. The uncorrected VMRs shown here can be strongly underestimated (up to a factor of 3), because they have been calculated with an overestimated path length. This is the case for background pollution as well as

of those differences, averaging of individual measurements over certain time periods was applied . shipping emission measurements.

Figures **??** and **??** show in the lower subplot daily means of the measurement periods presented above. Since the lack of comparability between both instruments for individual measurements, for a meaningful comparison and the computation of a correlation coefficient at this measurement site an averaging over longer time spans was applied to reduce the impact of the differences between both measurement methods. The fact that MAX-DOAS averages over a large horizontal distance should therefore cancel out on temporal average when comparing to in-situ measurements.

Figure 12 shows in Panel A three months of daily mean $NO_2$ VMRs from the in-situ and MAX-DOAS UV instrument in summer 2014 and in Panel B due to instrumental problems with the in-situ $SO_2$ device (see Fig. 4) six weeks of $SO_2$ daily mean VMRs from summer 2013. To have comparable conditions, for the in-situ instrument all measurements between the start of the MAX-DOAS measurements in the morning (with sunrise) and the end of measurements in the evening (with sunset) were have been averaged. The shaded areas show the corresponding standard deviation and indicate the variability during the single days.

As can be seen in the figures, even though The long gap in the $SO_2$ time series was caused by a power outage.

[Figure]

Figure 12: Comparison of MAX-DOAS (UV) and in-situ daily mean VMRs of $NO_2$ during summer 2014 (A) and $SO_2$ during summer 2013 (B). Shaded areas show the standard deviation for each daily mean value.

It is clearly visible that the in-situ values are usually systematically higher , as expected, a very good agreement of the progression of both curves is found. This illustrates that $
[revised manuscript text omitted]
. ~~This means that from the mean $NO_2$ level of $(1.49 \pm 1.30)$ ppb (mean $\pm$ standard deviation) measured on Neuwerk (averaged over all measurements), at least 0.31 ppb is attributed to come from shipping emissions and 0.47 ppb from land-originated sources. The remaining 0.71 ppb is either from ships or coming from the land, or, which is most probable, a mixture of both. The precise shares for this contribution cannot be distinguished from the available data.~~

If we consider only the two sectors, for which we can identify the primary sources and take theses as representative, we can say that 40 % of the $NO_2$ on Neuwerk is coming from shipping emissions, but with 60 %, the majority, is coming from land. One reason for that is that the island Neuwerk is relatively close to the coastline (around 10 km) and is obviously still impacted by polluted air masses from land, which has also been observed in the diurnal and weekly cycle analysis shown in Figures 14 and 15. This might also give us a hint that in coastal regions in Germany land-based sources like road traffic and industry are, despite the heavy ship traffic, the strongest source of air pollution and ship emissions come in second.

[Figure]

Figure 17: Contributions of ships and land-based pollution sources to measured NO₂ (top row) and SO₂ (middle and bottom row) levels on Neuwerk. For NO₂ the complete time series of measurements from 2013 to 2016 has been taken into account, for SO₂ the data have been divided into the time before and after the change in fuel sulfur content limits. The leftmost column of pie plots show the percentage of measurements with wind coming mainly from land (green), only from sea (blue) and from directions with mixed contributions (yellow). The middle column shows the contributions to the integrated, total volume mixing ratios from these source regions in percent. The rightmost column of pie plots shows analogous the percentage and mean VMR contribution by considering only the land and sea sector. It can clearly be seen that the lower fuel sulfur limit lead to a strong decrease in the SO₂ contribution from shipping since 2015.

[revised manuscript text omitted]

By comparing SO$_2$ to NO$_2$ ratios from different ships it is possible to roughly distinguish whether 985 a ship is using fuel with high or low sulfur content (giving a high or low SO$_2$ to NO$_2$ ratio). The SO$_2$ to NO$_2$ ratio can also give insights into the chemistry inside the plumes, since the relative amounts of NO$_2$ and NO in the emitted NO$_x$ depend on the time span from stack emission and the presence of tropospheric ozone for the conversion of the mainly produced NO to NO$_2$. Beecken and Mellqvist from Chalmers University (Sweden) use this relationship for airborne DOAS measurements of ship 990 exhaust plumes on an operational basis in the CompMon project (Compliance monitoring pilot for MARPOL Annex VI) (Van Roy, 2016). Following the ships and measuring across the stack gas plume they can discriminate between low (0.1 %) and high (1 %) fuel sulfur content ships with a probability of 80–90 % (Van Roy, 2016).

From the spectra measured by our MAX-DOAS UV instrument both SO$_2$ and NO$_2$ columns can 995 be retrieved at once. The two columns are measured at the exact same time along nearly the same light path. To calculate SO$_2$ to NO$_2$ ratios for the measured pollutant peaks simply the ratio of the measured differential slant column densities has to be computed.

In order to identify separate ship related signals from smooth background pollution, first a running median filter has been was applied to the time series of NO$_2$ and SO$_2$ measurements , to identify 1000 low values and to determine the baseline between the peaks, which originates from slowly varying

[revised manuscript text omitted]
". In: *Atmospheric Measurement Techniques* 6.7, pp. 1777–1791. ISSN: 18671381. DOI: 10.5194/amt-6-1777-2013.

Aliabadi, A. A., Staebler, R. M., and Sharma, S. (2015). "Air quality monitoring in communities of the Canadian Arctic during the high shipping season with a focus on local and marine pollution". In: *Atmospheric Chemistry and Physics* 15.5, pp. 2651–2673. ISSN: 16807324. DOI: 10.5194/acp-15-2651-2015.

Aulinger, A., Matthias, V., Zeretzke, M., Bieser, J., Quante, M., and Backes, A. (2016). "The impact of shipping emissions on air pollution in the greater North Sea region – Part 1: Current emissions and concentrations". In: *Atmospheric Chemistry and Physics* 16.2, pp. 739–758. DOI: 10.5194/acp-16-739-2016. URL: http://www.atmos-chem-phys.net/16/739/2016/.

Balzani Lööv, J. M., Alfoldy, B., Gast, L. F. L., Hjorth, J., Lagler, F., Mellqvist, J., Beecken, J., Berg, N., Duyzer, J., Westrate, H., Swart, D. P. J., Berkhout, A. J. C., Jalkanen, J. P., Prata, A. J., Van Der Hoff, G. R., and Borowiak, A. (2014). "Field test of available methods to measure remotely SOx and NOx emissions from ships". In: *Atmospheric Measurement Techniques* 7.8, pp. 2597–2613. ISSN: 18678548. DOI: 10.5194/amt-7-2597-2014.

Beecken, J., Mellqvist, J., Salo, K., Ekholm, J., and Jalkanen, J. P. (2014). "Airborne emission measurements of SO2, NOx and particles from individual ships using a sniffer technique". In: *Atmospheric Measurement Techniques* 7.7, pp. 1957–1968. ISSN: 18678548. DOI: 10.5194/amt-7-1957-2014.

Beecken, J., Mellqvist, J., Salo, K., Ekholm, J., Jalkanen, J. P., Johansson, L., Litvinenko, V., Volodin, K., and Frank-Kamenetsky, D. A. (2015). "Emission factors of SO2, NOx and particles from ships in Neva Bay from ground-based and helicopter-borne measurements and AIS-based modeling". In: *Atmospheric Chemistry and Physics* 15.9, pp. 5229–5241. ISSN: 16807324. DOI: 10.5194/acp-15-5229-2015.

Beirle, S., Platt, U., Glasow, R. von, Wenig, M., and Wagner, T. (2004). "Estimate of nitrogen oxide emissions from shipping by satellite remote sensing". In: *Geophysical Research Letters* 31.18, pp. 4–7. ISSN: 00948276. DOI: 10.1029/2004GL020312.

Beirle, S., Platt, U., Wenig, M., and Wagner, T. (2003). "Weekly cycle of NO2 by GOME measurements: a signature of anthropogenic sources". In: *Atmospheric Chemistry and Physics* 3.2, pp. 2225–2232. ISSN: 16807324. DOI: 10.5194/acpd-3-3451-2003.

Bell, T. L., Rosenfeld, D., and Kim, K. M. (2009). "Weekly cycle of lightning: Evidence of storm invigoration by pollution". In: *Geophysical Research Letters* 36.23, pp. 1–5. ISSN: 00948276. DOI: 10.1029/2009GL040915.

Berg, N., Mellqvist, J., Jalkanen, J. P., and Balzani, J. (2012). "Ship emissions of SO 2 and NO 2: DOAS measurements from airborne platforms". In: *Atmospheric Measurement Techniques* 5.5, pp. 1085–1098. ISSN: 18671381. DOI: 10.5194/amt-5-1085-2012.

Bobrowski, N. and Platt, U. (2007). "SO2/BrO ratios studied in five volcanic plumes". In: *Journal of Volcanology and Geothermal Research* 166.3-4, pp. 147–160. ISSN: 03770273. DOI: 10.1016/j.jvolgeores.2007.07.003.

Bogumil, K., Orphal, J., Homann, T., Voigt, S., Spietz, P., Fleischmann, O. C., Vogel, A., Hartmann, M., Kromminga, H., Bovensmann, H., Frerick, J., and Burrows, J. P. (2003). "Measurements of molecular absorption spectra with the SCIAMACHY pre-flight model: instrument characterization and reference data for atmospheric remote-sensing in the 230-2380 nm region". In: *Journal of Photochemistry and Photobiology A: Chemistry* 157.2-3, pp. 167–184. ISSN: 10106030. DOI: 10.1016/S1010-6030(03)00062-5. URL: http://linkinghub.elsevier.com/retrieve/pii/S1010603003000625.

Bollmann, M., Bosch, T., Colijn, F., Ebinghaus, R., Körtzinger, A., Latif, M., Matthiessen, B., Melzner, F., Oschlies, A., Petersen, S., Proelß, A., Quaas, M., Requate, T., Reusch, T., Rosenstiel, P., Schrottke, K., Sichelschmidt, H., Siebert, U., Soltwedel, R., Sommer, U., Stattegger, K., Sterr, H., Sturm, R., Treude, T., Vafeidis, A., Bernem, C. van, Beusekom, J. van, Visbeck, M., Wahl, M., Wallmann, K., and Weinberger, F. (2010). "Living With the Oceans". In: *World Ocean Review: Living with the oceans* 1, p. 236.

Brasseur, G. P. (1999). *Atmospheric chemistry and global change: [a textbook prepared by scientists at the National Center for Atmospheric Research, Boulder]*. Topics in environmental chemistry. New York, NY [u.a.]: Oxford Univ. Press. ISBN: 0195105214.

Chen, G., Huey, L. G., Trainer, M., Nicks, D., Corbett, J., Ryerson, T., Parrish, D., Neuman, J. A., Nowak, J., Tanner, D., Holloway, J., Brock, C., Crawford, J., Olson, J. R., Sullivan, A., Weber, R., Schauffler, S., Donnelly, S., Atlas, E., Roberts, J., Flocke, F., Hübler, G., and Fehsenfeld, F. (2005). "An investigation of the chemistry of ship emission plumes during ITCT 2002". In: *Journal of Geophysical Research D: Atmospheres* 110.10, pp. 1–15. ISSN: 01480227. DOI: 10.1029/2004JD005236.

Corbett, J. J. and Koehler, H. W. (2003). "Updated emissions from ocean shipping". In: *Journal of Geophysical Research* 108.D20, p. 4650. ISSN: 0148-0227. DOI: 10.1029/2003JD003751. URL: http://doi.wiley.com/10.1029/2003JD003751.

Corbett, J. J., Fischbeck, P. S., and Pandis, S. N. (1999). "Global nitrogen and sulfur inventories for oceangoing ships". In: *Journal of Geophysical Research* 104.D3, pp. 3457–3470. ISSN: 0148-0227. DOI: 10.1029/1998JD100040.

Corbett, J. J., Winebrake, J. J., Green, E. H., Kasibhatla, P., Eyring, V., and Lauer, A. (2007). "Mortality from ship emissions: A global assessment". In: *Environmental Science and Technology* 41.24, pp. 8512–8518. ISSN: 0013936X. DOI: 10.1021/es071686z.

Diesch, J. M., Drewnick, F., Klimach, T., and Borrmann, S. (2013). "Investigation of gaseous and particulate emissions from various marine vessel types measured on the banks of the Elbe in Northern Germany". In: *Atmospheric Chemistry and Physics* 13.7, pp. 3603–3618. ISSN: 16807316. DOI: 10.5194/acp-13-3603-2013.

DNV (2008). "Marpol 73/78 Annex VI - Regulations of Air Polution from Ships - Technical and Operational implications". In: *Fuel*, pp. 1–32.

Endresen, Ø., Sørgard, E., Sundet, J., Dalsøren, S., Isaksen, I., and Berglen, T. (2003). "Emission from international sea transportation and environmental impact". In: *Journal of Geophysical Research* 108.D17, pp. 1–22. ISSN: 0148-0227. DOI: 10.1029/2002JD002898.

EU (2005). "Directive 2005/33/EC of the European Parliament and of the Council". In: *Official Journal of the European Union* 1882, pp. 59–69. DOI: 10.3000/17252555.L_2009.140.eng. URL: http://eur-lex.europa.eu/LexUriServ/LexUriServ.do?uri=OJ:L:2005:191:0059:0069:EN:PDF.

EU (2008). "Directive 2008/50/EC of the European Parliament and of the Council of 21 May 2008 on ambient air quality and cleaner air for Europe". In: *Official Journal of the European Communities* 152, pp. 1–43. URL: http://eur-lex.europa.eu/LexUriServ/LexUriServ.do?uri=OJ:L:2008:152:0001:0044:EN:PDF.

— (2016). *Air Quality Standards.* URL: http://ec.europa.eu/environment/air/quality/standards.htm (visited on 12/15/2016).

Eyring, V., Bovensmann, H., Cionni, I., Dall'Amico, M., Franke, K., Khlystova, I., Klinger, C., Lauer, A., Paxian, A., Righi, M., and Schreier, M. (2010a). *Impact of Ship Emissions on Atmosphere and Climate, SeaKLIM Final Report.* Tech. rep. DLR, p. 23. URL: http://www.pa.op.dlr.de/SeaKLIM/SeaKLIM%5C_Nachwuchsgruppe%5C_Schlussbericht%5C_FINAL.pdf.

Eyring, V., Köhler, H. W., Aardenne, J. van, and Lauer, A. (2005a). "Emissions from international shipping: 1. The last 50 years". In: *Journal of Geophysical Research* 110.D17, p. D17305. ISSN: 0148-0227. DOI: 10.1029/2004JD005619. URL: http://doi.wiley.com/10.1029/2004JD005619.

Eyring, V., Köhler, H. W., Lauer, A., and Lemper, B. (2005b). "Emissions from international shipping: 2. Impact of future technologies on scenarios until 2050". In: *Journal of Geophysical Research* 110.D17, p. D17306. ISSN: 0148-0227. DOI: 10.1029/2004JD005620. URL: http://doi.wiley.com/10.1029/2004JD005620.

Eyring, V., Isaksen, I. S., Berntsen, T., Collins, W. J., Corbett, J. J., Endresen, O., Grainger, R. G., Moldanova, J., Schlager, H., and Stevenson, D. S. (2010b). "Transport impacts on atmosphere and climate: Shipping". In: *Atmospheric Environment* 44.37, pp. 4735–4771. ISSN: 13522310. DOI: 10.1016/j.atmosenv.2009.04.059. URL: http://linkinghub.elsevier.com/retrieve/pii/S1352231009003379.

[revised manuscript text omitted]